# Genome-wide association analyses highlight the role of the intestinal molecular environment in human gut microbiota variation

Despite the importance of the gut microbiome to health, the role of human genetic variation in shaping its composition remains poorly understood. Here we report genome-wide association analyses of harmonized metagenomic data from 16,017 adults in four Swedish population-based studies, with replication in 12,652 people from the Norwegian HUNT study. We identified variants in the *OR51E1–OR51E2* locus, encoding sensors for microbiome-derived fatty acids, associated with microbial richness. We further identified 15 study-wide significant genetic associations ($P < 5.4 \times 10^{-11}$) involving eight loci and 14 common bacterial species, of which 11 associations at six loci were replicated. The results confirm previously reported associations at *LCT*, *ABO* and *FUT2*, and provide evidence for new loci *MUC12*, *CORO7–HMOX2*, *SLC5A11*, *FOXP1* and *FUT3–FUT6*, with supporting data from metabolomics and gene expression analyses. Our findings link gut microbial variation genetically to gastrointestinal functions, including enteroendocrine fatty acid sensing, bile composition and mucosal layer composition.

The human gut microbiome—a complex community of microorganisms residing in the gastrointestinal tract—influences many physiological processes. Recent advances in sequencing technologies have enabled detailed characterization of this microbial community, uncovering its variability and associations with several health conditions[1,2]. Although human twin and primate multigenerational studies have demonstrated evidence for host genetic contributions to the microbiome composition[3,4], only a limited number of genome-wide association studies (GWAS) have been conducted. These include a meta-analysis of 24 studies including 18,240 participants that used 16S rRNA sequencing—a method offering limited species-level discrimination[5]. The study was further hampered by the fact that few shared bacterial taxa were detected across included studies, due partly to high variability in sample processing methods[5]—a common challenge in the field[6]. The largest high-resolution metagenomic study to date comprised 7,738 participants from the Netherlands[7]. So far, only variants in two loci,

harboring the lactase (*LCT*) and the histo-blood group ABO system transferase (*ABO*) genes, have been linked robustly and repeatedly to specific microbiome species at study-wide significance ($P < 5 \times 10^{-8}$ corrected for the number of species tested)[4,5,7–10]. A Finnish cohort of 5,959 people identified an additional study-wide significant signal near *MED13L*[9], but this signal has not been replicated in other studies. Other variants have been implicated at genome-wide significance ($P < 5 \times 10^{-8}$, no correction for the number of taxa tested), such as in the secretor status locus fucosyltransferase 2 (*FUT2*)[11].

Here we leveraged high-resolution metagenomic data from 16,017 participants across four Swedish studies, with replication in 12,652 participants from the Norwegian Trøndelag Health Study (HUNT). We identified and replicated a genetic association with microbiome alpha diversity mapping to the *OR51E1–OR51E2* locus that encodes microbial fatty acid chemosensors expressed by enteroendocrine cells (EECs). We further identified 15 single nucleotide polymorphism

✉e-mail: tove.fall@medsci.uu.se

(SNP)–species associations at study-wide significance representing eight genetic loci, of which five are new. Our findings highlight the contribution of gut physiological functions, including enteroendocrine chemosensing, bile acid metabolism and mucosal layer make-up in microbiome composition, paving the way for future studies and potential therapeutic interventions that consider both host genetics and microbiome profiles.

## Results

### GWAS of deep shotgun metagenomic data from four Swedish studies profiled with a standardized pipeline

We performed and meta-analyzed GWAS of gut microbiome composition in 16,017 participants of European ancestry from four Swedish studies sampled between 2011 and 2021 (Fig. 1 and Supplementary Table 1). Participants were aged 18 to 96 years and 51% were female. The mean study sequencing depth ranged from 25.3 to 56.1 million read pairs. To ensure comparability, stool metagenomic reads were processed using a standardized pipeline[12]. Analyses included alpha diversity (richness, Shannon, inverse Simpson), 921 species present in ≥5% of participants in all four cohorts (excluding 3,214 rarer species), 652 higher taxa and 117 functional modules. Based on simulations maximizing power and minimizing false positive findings, we applied logistic regression for 679 species present in ≤50% of participants in all four cohorts (testing 5,368,906 variants, minor allele frequency (MAF) ≥ 5%) and linear regression for 242 species with >50% prevalence (7,454,886 variants, MAF ≥ 1%). GWAS was run separately by cohort and phenotype using REGENIE v.3.3 with sex, age, age$^2$, plate and genetic principal components 1–10 as covariates; results were meta-analyzed by inverse-variance weighted fixed effects. Study-wide associations with species and diversity were replicated in HUNT ($n = 12,652$).

### A locus including genes encoding EEC receptors is implicated in gut microbial richness

Low gut microbial alpha diversity has been associated with higher risk of metabolic disorders, although causality remains uncertain[13,14]. We estimated heritability at 9% for Shannon index and 20% for richness (Supplementary Table 2), lower than the 30–37% reported in twin studies[4]. We found associations (lead variant rs10836441-T) in the locus covering *OR51E1* (mouse ortholog *Olfr558*) and *OR51E2* (*Olfr78*) genes on chromosome (Chr.) 11 (Extended Data Fig. 1a) with microbiome richness (−5.7 species per T allele, $P = 1.9 \times 10^{-9}$; Supplementary Table 3), which was replicated in the HUNT study (−2.8 species per T allele, $P = 2.1 \times 10^{-3}$). The imputation of genotypes for rs10836441 was confirmed in a subset of 148 people using Sanger sequencing with a concordance of 100% (Extended Data Fig. 2a). rs10836441 is an expression quantitative trait locus (eQTL) for *OR51E2* and *OR51E1* expression in several tissues (GTEx v.8)[15]. At the species level, rs10836441 was associated at the genome-wide level with the uncharacterized species HGM14224 sp900761905 (*Bacillota* phylum) and with SFEL01 sp004557245. The latter is reported as a predictor of response to short-chain fatty acids (SCFA) supplementation in Parkinson's disease[16]. *OR51E1* and *OR51E2* belong to the large olfactory receptor gene family encoding G protein-coupled receptors expressed primarily in the olfactory epithelium but also more broadly across the body[17]. Recently, the proteins encoded by the mouse orthologs of *OR51E1* and *OR51E2* have been identified as sensors for gut microbiome-derived short-, medium- and branched-chain fatty acids in EECs[18]. EECs are hormone-producing cells in the gastrointestinal epithelium, with important roles in the physiological response to feeding, such as gut motility and satiety. A role of EECs in microbiome composition is supported by a recent study where mice deficient in colonic EECs were shown to have lower alpha diversity compared to controls[19]. Further, knockout of the *OR51E2* receptor ortholog in a mouse model of colitis caused higher levels of intestinal inflammation[20]. EECs express several fatty acid chemosensors, such as FFAR1-FFAR4, of which FFAR2 and FFAR3 are

relevant for sensing microbiome-derived SCFA[21]. Further corroborating our findings of a potential role of fatty acid chemosensing of EECs in microbiome composition, we observed that genetic variants in the *FFAR1–FFAR2–FFAR3* locus at Chr. 19 were associated at near study-wide significance with *Pullichristensenella excrementipullorum* ($P = 5.7 \times 10^{-11}$, Supplementary Table 4; replicated in HUNT, $P = 1.5 \times 10^{-3}$). The lead variant rs75481361 at the *FFAR1–FFAR2–FFAR3* locus was also associated with the same uncharacterized species as rs10836441 (HGM14224 sp900761905, $P = 2.3 \times 10^{-9}$; Supplementary Table 4) and associated nominally with richness ($P = 5 \times 10^{-3}$). rs75481361 is reported as an eQTL for *FFAR3* in colon tissue (GTEx v.8)[15]. We assessed the expression of *OR51E1* and *OR51E2* in single-cell RNA sequencing (scRNA-seq) from three sources: human intestinal cells[22], EECs purified from human duodenal and ileal organoids[23] and in EECs of transgenic mice[24]. The scRNA-seq data from human intestinal cells showed expression of *OR51E1* in EECs along the intestinal tract, whereas *OR51E2* was expressed mainly in EECs in the colon (Extended Data Fig. 3a). *OR51E2* was expressed across most colonic immune cell types, highest in T cells and monocytes/macrophages (Extended Data Fig. 3b). *FFAR1* was restricted mainly to duodenal and ileal EECs, *FFAR2* to several cell types including EECs, whereas *FFAR3* showed overall low expression. To evaluate the expression of these olfactory and fatty acid receptor genes in different EEC types, we analyzed scRNA-seq from EECs purified from human duodenal and ileal organoids[23] (Extended Data Fig. 4) and from EECs of transgenic mice[24] (Extended Data Fig. 3c). In the human organoid-derived EECs, we observed overlap of *OR51E1* expression with tryptophan hydroxylase 1 (TPH1)—a marker of enterochromaffin cells. Enterochromaffin cells constitute less than 1% of the total intestinal epithelium cells but have important effects on modulating motility by release of serotonin. However, the lead variant rs10836441 was not associated ($P = 0.62$) with self-reported stool frequency—a proxy measurement of gastrointestinal motility—in a published GWAS[25]. The expression of *OR51E2* was considerably lower in the human duodenal and ileal organoids (Extended Data Fig. 4), consistent with the human intestinal results (Extended Data Fig. 3a). The mouse ortholog of *OR51E2* (*Olfr78*) was expressed in L-cells in the mouse lower intestinal tract, which are responsible for secretion of glucagon-like peptide 1 (GLP-1), peptide YY (PYY) and insulin-like peptide 5 (INSL5). To test whether the *OR51E1–OR51E2* locus was linked to GLP-1 or SCFA, we examined rs10836441 in relation to fasting and 2-h post-oral glucose load GLP-1 in up to 3,514 participants from the Malmö Diet and Cancer Study (MDC) and the Prevalence, Prediction and Prevention of Type 2 Diabetes–Botnia Study (PPP-Botnia) and to SCFA in 1,800 people from the Malmö Offspring Study (Supplementary Tables 5 and 6). No association could be detected in this somewhat limited sample when correcting for multiple testing. In summary, our results suggest that genetic variation affecting SCFA chemosensors that are expressed in EECs is relevant to the human gut microbiome composition; however, more research is needed to determine the causal genes and mechanism of action.

### Meta-analysis identified eight genetic loci associated with 14 microbial species at study-wide significance

After clumping of meta-analysis results, we found 149 SNP–species associations at the genome-wide significance level ($P < 5 \times 10^{-8}$; Supplementary Table 4) comprising 113 loci separated by at least 100 kb and 132 species. We used FUMA[26] to identify functional or phenotypic genesets and found 38 enrichments, including genesets previously linked to diet ($n = 10$), cancer biomarkers ($n = 3$), blood group ($n = 3$), gallstone disease ($n = 1$) and waist-to-hip ratio (WHR) adjusted for body mass index (BMI) (WHRadjBMI) ($n = 1$) (Supplementary Table 7). At the stricter study-wide threshold ($P < 5.4 \times 10^{-11}$), we identified 15 SNP–species associations across eight loci and 14 species (Figs. 2 and 3 and Table 1), and 12 SNP–higher taxa associations at five loci (*LCT*, *PLEKHG1*, *MUC12*, *ABO* and *SLC5A11*) (Supplementary Table 8).

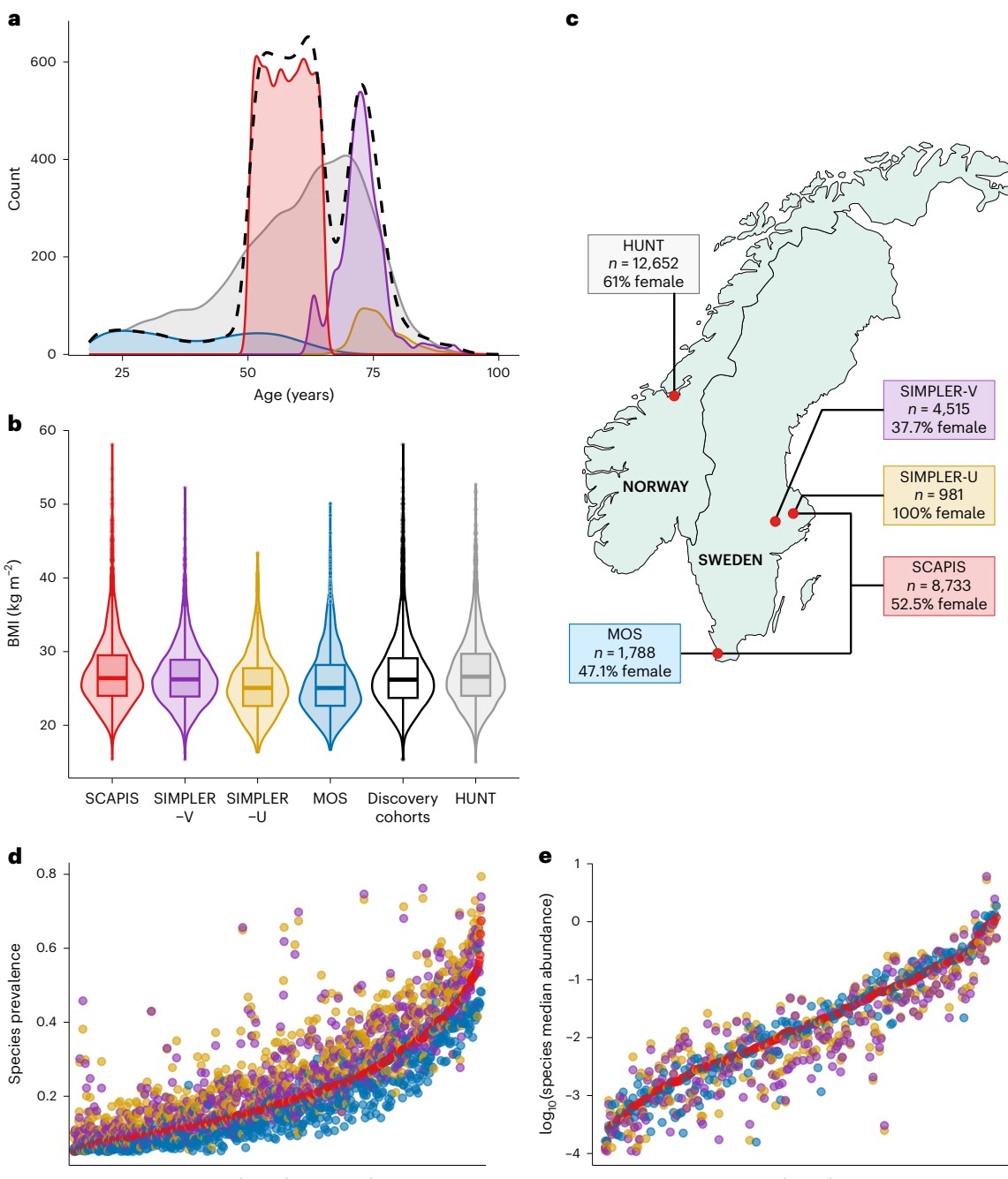

**Fig. 1 | Characteristics of participants and microbiome composition across studies. a**, Density plots of age of participants in the discovery studies (SCAPIS, $n$ = 8,733; SIMPLER-V, $n$ = 4,515; SIMPLER-U, $n$ = 981; MOS, $n$ = 1,788; total $n$ = 16,017 individuals) and in HUNT. Dashed line: combined discovery studies. **b**, Violin and boxplots of BMI of participants in the discovery studies (SCAPIS, $n$ = 8,733; SIMPLER-V, $n$ = 4,512; SIMPLER-U, $n$ = 978; MOS, $n$ = 1,788; total $n$ = 16,011) and in HUNT ($n$ = 12,652). Violin plots show the density distribution. The boxplots within the violin plots show the medians and the IQR, and whiskers extend to the values no larger than 1.5 times the IQR (upper whisker) or smaller

than 1.5 times the IQR (lower whisker). Outliers are depicted as individual points. **c**, Map with the study sites for the discovery studies in Sweden (SCAPIS, SIMPLER-V, SIMPLER-U and MOS) and the replication cohort in Norway (HUNT), including the sample size and proportion of female participants in each study. **d**, Prevalence for the species analyzed with the logistic model. **e**, The log-transformed median abundance for the species analyzed with the linear model in the discovery studies. In **d** and **e**, each dot represents one species. Species are ranked by their prevalence and median abundance in SCAPIS.

The 14 species had a median heritability of 13% (interquartile range (IQR) 5–16%; Supplementary Table 2), highest for *Clostridium saudiense* (33%). Corresponding estimates were 11% (IQR 5–19%) for species with genome-wide associations and 8% (IQR 3–16%) for those without. All 14 species were at least moderately prevalent; the least prevalent species was detected in 27% of the participants. Candidate genes based on genetic distance, eQTL data, gene expression in human intestinal

cells (Extended Data Fig. 3) and biological function were *LCT*, *ABO*, *FOXP1*, *MUC12*, *CORO7–HMOX2*, *SLC5A11*, *FUT2* and *FUT3–FUT6*—all expressed in the human intestine (Extended Data Fig. 3). We did not observe evidence of genomic inflation (mean λ = 1.03; s.d. = 0.02), and findings were consistent across studies (Supplementary Table 4 and Extended Data Fig. 5). No differences between estimates were found in the sex-stratified analysis at the 5% false discovery rate (FDR) level

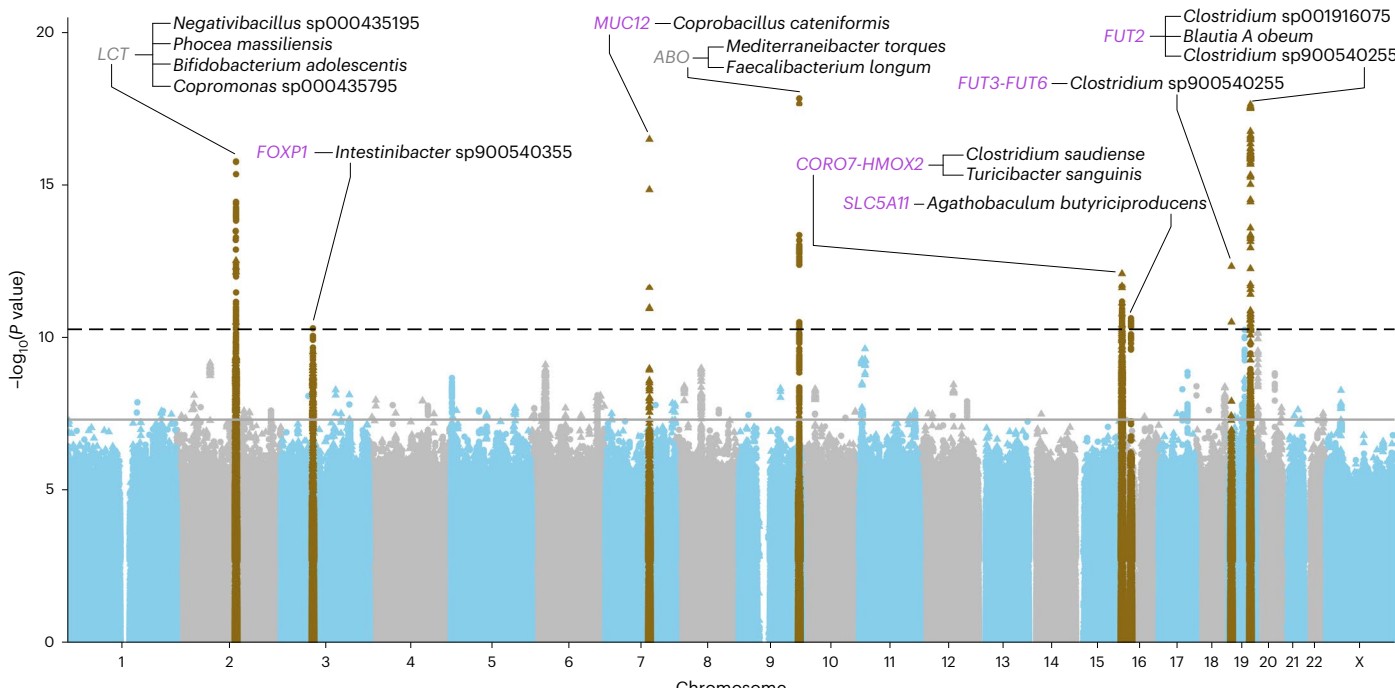

**Fig. 2 | Manhattan plot for associations between genetic variants and 921 species in the discovery studies (n = 16,017).** The dashed black line represents the study-wide ($P < 5.4 \times 10^{-11}$, after Bonferroni correction of the genome-wide threshold), and the solid gray line genome-wide ($P < 5.0 \times 10^{-8}$) significant thresholds. Filled triangles represent binary outcomes (absence/presence), which were tested using logistic regression models; circles represent continuous outcomes (relative abundance), which were tested using linear regression models. All tests were two-sided. Loci not found previously in other GWAS at study-wide significance are indicated in purple.

(Supplementary Table 9). The genome-wide significant associations were consistent in sensitivity analyses using models with centered log-ratio transformation (linear regression models), with Firth correction (logistic regression models), without age[2] as a covariate, with study sites analyzed separately, excluding all but one person per household, excluding one from each related pair, excluding recent antibiotic users, excluding self-reported inflammatory bowel disease (IBD) cases and including BMI, smoking, alcohol or fiber intake as covariates, respectively (Extended Data Fig. 6).

Of the 15 SNP–species associations, we replicated 11 at six loci in HUNT at the Bonferroni-corrected threshold ($P < 3 \times 10^{-3}$) and all 15 at $P < 0.05$ with consistent effect direction. Of these 15 SNP–species associations, seven were present in FINRISK[9] and four in the Dutch Microbiome Project[7], of which seven and two were replicated, respectively (Supplementary Table 10). Allele frequencies were comparable across studies, except for the *LCT* SNPs in FINRISK (Supplementary Table 11), which are known to vary across populations[27]. Lactase persistence alleles at *LCT* were associated with decreased levels of *Bifidobacterium adolescentis* and with increased levels of *Phocea massiliensis, Negativibacilus* sp000435195 and *Copromonas* sp000435795, and at genome-wide level with five additional species, including three *Bifidobacterium* species (Extended Data Fig. 1b–e and Supplementary Table 4). Variants in *LCT* were also associated at study-wide significance with the genera *Phocea* and *Bifidobacterium*, and the family, order and class (*Bifidobacteriaceae, Actinomycetales* and *Actinomycetia*, respectively) of the *Bifidobacterium* spp. In a nontargeted plasma metabolomics analysis in the Swedish CArdioPulmonary bioImage Study (SCAPIS), we confirmed previously reported associations of the *LCT* lead variants with the glycemic marker 1,5-anhydroglucitol[28] and found associations with vitamin B6 levels (Supplementary Table 12; FDR $q < 0.05$). Our colocalization analysis revealed a shared genetic signal in the *LCT* locus for *B. adolescentis, P. massiliensis, Negativibacilus* sp000435195 and *Copromonas* sp000435795 with plasma levels of the secondary bile acid

isoursodeoxycholate and low-density lipoprotein (LDL) cholesterol (Supplementary Table 13). Our findings thus expand the number of robustly replicated microbiome-associated loci from two (*ABO* and *LCT*) to six (*ABO, LCT, FUT2, MUC12, CORO7–HMOX2* and *SLC5A11*) and provide strong supportive evidence for two additional (*FUT3–FUT6* and *FOXP1*) loci.

## Several associations support an important role of fucosylated glycans in microbiome regulation

In this study, we found study-wide species associations with three loci linked to the phenotypical variation and secretion of histo-blood group antigens: *ABO, FUT2* and *FUT3–FUT6* (Extended Data Figs. 1f–h and 7a–c). Histo-blood group antigens are fucosylated glycans present on cell surfaces and in secretions, including the gastrointestinal mucus layer. These antigens constitute a carbon source and binding site for many gut bacteria[29]. We confirmed previous associations of *ABO* variants with *Faecalibacterium longum* and reported new associations with *Mediterraneibacter torques* and the genus *UMGS1623*. The association of *ABO* variants with specific species and strains is reported to depend on the secretor status of histo-blood group antigens determined by variations in *FUT2*–a gene encoding a fucosyltransferase[11]. Nonsecretors, who comprise about 20% of people of European ancestry, do not secrete histo-blood group antigens in bodily secretions such as saliva and mucus.

Given the association with *ABO*, an association between *FUT2* variants and the gut microbiome is expected[11] but has so far been observed only at the genome-wide significance level[5,30]. Here we identified three species associated with *FUT2* variants at a study-wide significance level: *Blautia A obeum, Clostridium* sp900540255 and *Clostridium* sp001916075, and on a genome-wide significance level with *Mediterraneibacter torques, Mediterraneibacter faecis* and *Ruminococcus B gnavus. Blautia A obeum* is a highly prevalent species that has been shown to harbor glycosyl hydrolase genes that can remove fucose from glycans[31].

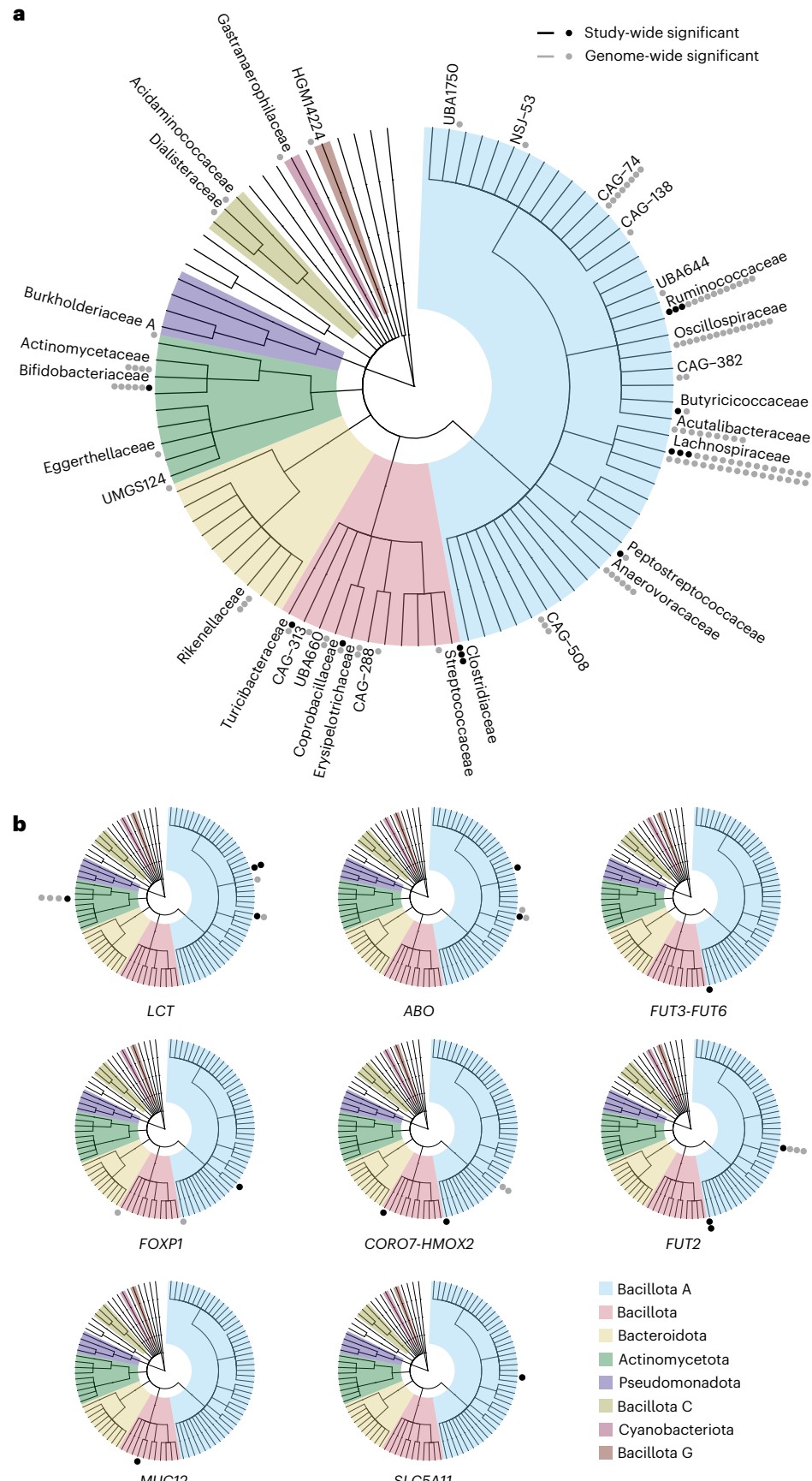

**Fig. 3 | Cladogram of genetic associations with gut bacterial species.**
**a**, The phylogenetic tree layers from center to periphery are kingdom-phyla-class-order-family, and all families captured by the 921 species are plotted. Phyla with at least one genetic association are colored. Species are placed at their family. **b**, Per-locus associations with microbiome species for loci with at least one study-wide significant association. Each dot corresponds to one species.

**Table 1 | Loci associated with gut microbiome composition at study-wide significance**

| | Variant | | | | | Microbiome feature | | Model | Swedish studies | | | HUNT | | |
|---|---|---|---|---|---|---|---|---|---|---|---|---|---|---|
| Locus | Lead variant | Chr: Pos37 | EA/OA | Effect prediction | EAF | Trait | Prev | | Beta | s.e. | P | Beta | s.e. | P |
| OR51E1–OR51E2 | rs10836441 | 11:4689742 | T/C | Intergenic | 0.52 | Richness | NA | Linear | **−0.06** | **0.01** | **1.9×10⁻⁹** | **−0.04** | **0.01** | **2.1×10⁻³** |
| LCT | rs4988235 | 2:136608646 | A/G | Intron (*MCM6*) | 0.72 | *Negativibacillus* sp000435195 (NCBIª: *Clostridium* sp. CAG:169) | 28.1 | Logistic | 0.23 | 0.03 | 2.9×10⁻¹³ | 0.09 | 0.04 | 0.04 |
| | rs4988235 | 2:136608646 | A/G | intron (*MCM6*) | 0.72 | *Phocea massiliensis* | 75.7 | Linear | **0.08** | **0.01** | **1.4×10⁻¹¹** | **0.08** | **0.02** | **5.1×10⁻⁶** |
| | rs182549 | 2:136616754 | T/C | Intron (*MCM6*) | 0.72 | *Bifidobacterium adolescentis* | 90.0 | Linear | **−0.10** | **0.01** | **1.7×10⁻¹⁶** | **−0.14** | **0.02** | **1.4×10⁻¹⁴** |
| | rs6754311 | 2:136707982 | T/C | Intron (*DARS*) | 0.72 | *Copromonas* sp000435795 (NCBIª: *Alitiscatomonas acetii*) | 67.7 | Linear | **0.08** | **0.01** | **3.2×10⁻¹¹** | **0.10** | **0.02** | **2.4×10⁻⁸** |
| FOXP1 | rs17007949 | 3:70920041 | C/G | Intergenic | 0.32 | *Intestinibacter* sp900540355 (NCBIª: *Clostridium* sp. 1001270J_160509_D11) | 55.9 | Linear | 0.07 | 0.01 | 5.1×10⁻¹¹ | 0.03 | 0.01 | 7.3×10⁻³ |
| MUC12 | rs4556017 | 7:100632790 | T/C | Intron (*MUC12*) | 0.83 | *Coprobacillus cateniformis* | 26.9 | Logistic | **0.34** | **0.04** | **3.3×10⁻¹⁷** | **0.38** | **0.06** | **1.7×10⁻¹¹** |
| ABO | rs9411378 | 9:136145425 | A/C | Intron (*ABO*) | 0.28 | *Mediterraneibacter torques* (NCBI*: [*Ruminococcus*] *torques*) | 86.9 | Linear | **0.11** | **0.01** | **1.4×10⁻¹⁸** | **0.12** | **0.01** | **1.0×10⁻¹⁶** |
| | rs550057 | 9:136146597 | T/C | Intron (*ABO*) | 0.31 | *Faecalibacterium longum* | 96.6 | Linear | **0.08** | **0.01** | **3.8×10⁻¹¹** | **0.08** | **0.02** | **1.6×10⁻⁹** |
| CORO7–HMOX2 | rs8182173 | 16:4420787 | T/C | Intron (*CORO7*) | 0.23 | *Clostridium saudiense* | 40.2 | Logistic | −0.22 | 0.03 | 7.8×10⁻¹³ | −0.11 | 0.05 | 0.02 |
| | rs4785960 | 16:4453319 | C/G | Intron (*CORO7*) | 0.26 | *Turicibacter sanguinis* | 53.9 | Linear | **−0.08** | **0.01** | **2.0×10⁻¹²** | **−0.04** | **0.01** | **1.7×10⁻³** |
| SLC5A11 | rs55808472 | 16:24931691 | A/G | Noncoding transcript exon (AC008731.1) | 0.06 | *Agathobaculum butyriciproducens* | 98.3 | Linear | **0.15** | **0.02** | **2.4×10⁻¹¹** | **0.16** | **0.03** | **4.3×10⁻⁹** |
| FUT3–FUT6 | rs708686 | 19:5840619 | T/C | Upstream gene (*FUT6*) | 0.30 | *Clostridium* sp900540255 (NCBIª: uncultured *Clostridium* sp.) | 36.3 | Logistic | −0.20 | 0.03 | 4.5×10⁻¹³ | −0.11 | 0.05 | 0.02 |
| FUT2 | rs679574 | 19:49206108 | C/G | Intron (*FUT2*) | 0.56 | *Clostridium* sp001916075 (NCBIª: *C. lentum*) | 31.2 | Logistic | **0.17** | **0.03** | **2.5×10⁻¹¹** | **0.15** | **0.05** | **1.6×10⁻³** |
| | rs492602 | 19:49206417 | A/G | Synonymous (*FUT2*) | 0.56 | *Blautia A obeum* (NCBIª: *B. obeum*) | 96.4 | Linear | **0.07** | **0.01** | **1.6×10⁻¹¹** | **0.08** | **0.01** | **7.6×10⁻¹⁰** |
| | rs681343 | 19:49206462 | T/C | Stop gained (*FUT2*) | 0.44 | *Clostridium* sp900540255 (NCBIª: uncultured *Clostridium* sp.) | 36.3 | Logistic | **−0.22** | **0.03** | **2.2×10⁻¹⁸** | **−0.19** | **0.04** | 1.1 |

Associations shown here were those at study-wide significance after Bonferroni correction of the genome-wide threshold, that is, $P<1.7×10^{-8}$ for richness and $P<5.4×10^{-11}$ for species. Bold type indicates those robustly replicated in HUNT at a Bonferroni-corrected $P=3.3×10^{-3}$. Betas are regression coefficients in standard deviation richness or species abundance per effect allele (calculated using a linear regression model) or log odds of species presence per effect allele (calculated using a logistic regression model). Tests were two-sided. Locus, manually assigned locus name based on previous GWAS assignment or function of nearby genes if new; Lead variant, reference SNP identifier of the locus lead variant (that is, the variant with the lowest $P$ value); Pos37, human genome GRCh37 position on the chromosome; EA, effect allele; OA, other allele; EAF, mean effect allele frequency across studies; Trait, microbial species richness or species name; Model, GWAS regression model; Prev, mean species prevalence across studies (based on rarefied relative abundances for logistic models and nonrarefied relative abundances for linear models); Swedish studies $P$, $P$ value in Swedish studies SCAPIS, SIMPLER-V, SIMPLER-U and MOS (discovery); HUNT $P$, $P$ value in HUNT (replication). ªNational Center for Biotechnology Information (NCBI) equivalents refer to the unfiltered NCBI taxonomy of GTDB species representative as of 2024-04-24. This was only added for species for which the name of the NCBI equivalent was different than GTDB.

The lead variant in the current study is in close linkage disequilibrium (LD) with rs601338, which introduces a stop codon resulting in the nonsecretor status. Variants in *FUT2* have been linked previously to IBD, and our colocalization results show evidence of shared causal variants of IBD with *Blautia A obeum*, *Clostridium* sp900540255 and *Clostridium* sp001916075 (Supplementary Table 13). To ascertain that our *FUT2*-associations were not due to secondary effects of IBD, we reanalyzed the results excluding IBD cases, which yielded similar results (all $P < 3.7 × 10^{-10}$; Extended Data Fig. 6). We identified associations of *ABO* and *FUT2* lead variants with plasma secondary bile acid levels—probably an effect of altered gut microbiome composition as bacteria are responsible for the conversion of primary to secondary bile acids (Supplementary Table 12; FDR $q < 0.05$). We found strong evidence for a secretor-status-dependent effect of genetically predicted expression of the ABO A antigen (blood groups A or AB) on *M. torques* abundance but not for the B antigen (blood group B) (Supplementary Table 14; interaction $P = 5.7 × 10^{-7}$). The abundance of *M. torques* was higher in secretors

(median abundance 0.06 (Q1, Q3 0.004, 0.26)) than in nonsecretors (0.03 (0.0008, 0.17)) in those presumed to express antigen A, and low (median 0.03) in those predicted to express the antigen B, irrespective of secretor status. These findings might be explained by the potential of *M. torques*, also known as *Ruminococcus torques*, to produce an α-N-acetylgalactosaminidase that removes N-acetylgalactosamine (GalNac) from the antigen A[32].

*FUT2* also determines the phenotype of the Lewis blood group antigen; those who are secretors express Le(b) instead of Le(a), provided that the person carries a functional *FUT3* gene. The Le(b) antigen is proposed to act as a binding site for bacteria such as *Helicobacter pylori*[33]. Here we found associations of the *FUT3*–*FUT6* locus with the species *Clostridium* sp900540255. The *FUT3* locus has not been associated previously with gut microbiome traits but has been linked to several other traits, such as gallstone disease[34] and LDL cholesterol[35]. Our colocalization analysis provided strong evidence for a shared genetic signal for *Clostridium* sp900540255 with LDL cholesterol, at

both the *FUT2* and the *FUT3–FUT6* loci (Supplementary Table 13). We also tested for the interaction of secretor status and the Lewis blood group (Le$^+$ versus Le$^-$) for relevant species. However, in contrast to the *ABO* findings, we did not find robust evidence that the effect of Lewis antigen is dependent on secretor status. Taken together, our observed associations of the *ABO*, *FUT2* and *FUT3–FUT6* loci with specific bacterial species underline the importance of fucosylated glycans in shaping the gut microbial landscape.

## Genes involved in the mucosal layer implicated in gut microbiome composition

We discovered and replicated an association between a variant in an intron of *MUC12* and *Coprobacillus cateniformis*, flanked by two other mucin genes, *MUC3A* and *MUC17* (Extended Data Fig. 7d). The same variant was also associated at study-wide significance with the genus *Coprobacillus*. Our genotyping array did not cover the *MUC3A* gene region well due to gaps in the human genome assemblies for the human *MUC3* cluster[36]. Imputed genotypes for the lead variant rs4556017 were confirmed in a subset of 148 people using Sanger sequencing with a concordance of 96.6% (Extended Data Fig. 2b). Mucins, including MUC3A, MUC12 and MUC17, are main components of the enterocyte glycocalyx and are heavily O-glycosylated glycoproteins. *MUC12* is expressed most strongly by enterocytes and goblet cells in the human colon, whereas *MUC3A* and *MUC17* are expressed most strongly in the duodenum and ileum (Extended Data Fig. 3). Host glycans play an important role in determining which bacteria can colonize the host, and serve as an important nutrient source for gut microbes[37]. Variants in this locus have been associated previously with stool frequency[25], and we showed through colocalization analysis evidence supporting a shared genetic signal between *C. cateniformis* and stool frequency (*P*(H4) > 0.99; Supplementary Table 13). *C. cateniformis* is a recently described Gram-positive, nonsporulating, anaerobic, rod-shaped bacterium[38]. The stool levels of *C. cateniformis* were reported to decrease in patients with irritable bowel syndrome after fecal microbiota transplantation and were correlated positively with both symptoms and fatigue[39]. Variants near mucin genes (*MUC5*, *MUC12*, *MUC13*, *MUC22*) have been suggested previously at genome-wide or near genome-wide significance with metagenomic features[9,40,41]. Our findings corroborate previous findings that genetic variations in mucin genes can shape the gut microbiome composition.

## Shared genetic background of *Turicibacter* sp., *Clostridium saudiense*, *Intestinibacter* sp900540355, adiposity traits and bile acids

We discovered new associations of variants in the *CORO7–HMOX2* locus on Chr. 16 with the strictly anaerobic, Gram-positive *Turicibacter sanguinis* (rs4785960, *P* = 2.0 × 10$^{-12}$; replication *P* = 1.7 × 10$^{-3}$), with the spore-forming, anaerobic, Gram-positive *Clostridium saudiense*, previously known as *Clostridium saudii* (*P* = 7.8 × 10$^{-13}$; replication *P* = 0.02), and at a genome-wide threshold with *Intestinibacter* sp900540355. Genes located in this locus include *CORO7*, *VASN*, *PAM16* and *HMOX2* (Extended Data Fig. 7e,f). eQTL analysis showed that the lead variants are associated with the expression of several of these genes in several tissues. We found another locus with a similar pattern of species associations near *FOXP1* on Chr. 3, which was associated with *Intestinibacter* sp900540355 (rs17007949; *P* = 5.1 × 10$^{-11}$) at study-wide significance level (Extended Data Fig. 7g), and with *C. saudiense*, *Faecalibacterium prausnitzii F* and *Turicibacter bilis* at the genome-wide significance level. Variants near *FOXP1*, which has a key role in the immune system[42,43], have been associated previously with traits such as neutrophil count, hemorrhoidal disease, Crohn's disease, dietary intake and Barrett's esophagus, and at genome-wide significance with *Leptospirales*[9]. A variant in a third locus near *PLEKHG1* was also associated at study-wide significance with the *Turicibacter* genus, family (*Turicibacteraceae*) and order (*Haloplasmatales*) of *Turicibacter* spp. (Supplementary Table 8).

A recent study has shown that some *Turicibacter* strains encode and produce bile salt hydrolases—enzymes involved in producing secondary bile acids[44]. Furthermore, mice gavaged with *Turicibacter* presented with alterations in fat mass and circulating bile acids and lipids[44]. In our metabolomics analysis, the *Turicibacter*-lowering C allele of rs4785960 in the *CORO7–HMOX2* locus was associated with higher plasma levels of several secondary bile acids (Supplementary Table 12). Consistent findings were observed when examining the associations of *T. sanguinis* and *C. saudiense* abundances with these secondary bile acid metabolites in plasma (Supplementary Table 15). The lead variant in the *FOXP1* locus was associated with stool levels of the secondary bile acid glycoursodeoxycholate (*P* = 9.8 × 10$^{-7}$; Supplementary Table 16). We observed a shared genetic signal between *Intestinibacter* sp9005540355 and LDL cholesterol in the *FOXP1* locus, but not between *T. sanguinis*, *C. saudiense* and LDL cholesterol in the *CORO7–HMOX2* locus. We performed a Mendelian randomization (MR) analysis to investigate potential bidirectional effects between LDL cholesterol and *Intestinibacter* sp9005540355. The analysis suggested a positive effect of *Intestinibacter* sp9005540355 abundance on LDL cholesterol (*P* = 4.4 × 10$^{-4}$; *q*-value = 0.001) but not in the opposite direction (Supplementary Table 17 and Extended Data Fig. 8). Creating the genetic instruments using a more liberal *P* value threshold of 5 × 10$^{-6}$ yielded concordant results (*P* = 0.006; *q*-value = 0.02); however, the MR–Egger intercept indicates the presence of horizontal pleiotropy in this liberal analysis (*P* = 0.012). The *CORO7–HMOX2* locus was reported previously to be associated with WHRadjBMI[45]. We found that WHRadjBMI shares a genetic signal with *T. sanguinis* and *C. saudiense* in colocalization analyses (*P*(H4) > 0.94) (Supplementary Table 13). The MR analysis showed evidence of an effect of *T. sanguinis* on WHRadjBMI, but not in the opposite direction. Analyses using the liberal *P* value threshold of 5 × 10$^{-6}$ to create genetic instruments did not support the effect of *T. sanguinis* on WHRadjBMI (*P* = 0.23). Although the mechanism is still unclear, it seems plausible that these two loci might affect similar or the same pathways. Our findings suggest that genetic variations at two different loci, *CORO7–HMOX2* and *FOXP1*, affect a shared set of bacteria, including *Turicibacter* sp., *C. saudiense* and an *Intestinibacter* species, as well as LDL cholesterol, bile acids and body composition.

## Variants in the *SLC5A11* locus associated with a butyrate-producing bacterium

We identified variants in the *SLC5A11* locus on Chr. 16 associated with the abundance of *Agathobaculum butyriciproducens* and its family *Butyricicoccaceae* (Extended Data Fig. 7h and Supplementary Table 8). This locus has been linked previously to the related genus *Butyricicoccus* at genome-wide significance[46]. The lead variant rs55808472 is an eQTL for *SLC5A11*. The species-increasing A allele reduces *SLC5A11* expression (also known as *SMIT2* or *SGLT6*) in the ileum[47]. This gene encodes sodium/myo-inositol cotransporter 2, which mediates apical myo-inositol absorption in the intestine. Myo-inositol plays roles in various physiological processes, including cellular signaling as a precursor for phosphatidylinositol and inositol phosphates. In SCAPIS, our metabolomics analysis confirmed previous findings[48] of an association between the A allele and lower plasma myo-inositol (*P* = 1.2 × 10$^{-6}$; Supplementary Table 12). *A. butyriciproducens* is a strictly anaerobic, butyric acid-producing bacterium and has been implicated in mouse models as a potentially beneficial agent for cognitive function, Alzheimer's disease pathology and Parkinson's disease[49]. Another gene in the locus is *ARHGAP17* encoding the RhoGTPase-activating protein 17, known to be involved in the maintenance of tight junctions and vesicle trafficking. Arhgap17-deficient mice have increased intestinal permeability and impairment of the mucosal layer compared to wild-type mice in a colitis model[50]. Our findings provide evidence for a genetic variant in the *SLC5A11* locus affecting the abundance of *A. butyriciproducens*—a bacterium with potential health-beneficial effects.

## Loci associated with microbial functions suggest genetic links to microbial carbohydrate and amino acid catabolism

We investigated associations between host genetic variation and 117 previously curated functional modules representing different aspects of microbial metabolism[51] and microbial functions implicated in the gut–brain axis[52]. No study-wide significant findings were identified. Using the genome-wide significance threshold, we found that 11 candidate genetic loci, including *CYP7A1* and *EGFR*, associated with 11 microbial functions, most related to carbohydrate and amino acid catabolism (Supplementary Table 18).

## Discussion

We have identified and replicated a human genetic variant associated with gut microbiome richness at genome-wide significance: the *OR51E1–OR51E2* locus. We further report 15 study-wide and 149 genome-wide significant associations of genetic variants with individual microbial species, where the 15 study-wide associations represent eight loci and 14 species. Of these 15, 11 were replicated in an external sample using strict criteria and the remaining four were nominally significant. The eight loci included the well-known *ABO* and *LCT* loci, the previously suggested *FUT2* and five new loci (*MUC12*, *CORO7–HMOX2*, *SLC5A11*, *FOXP1* and *FUT3–FUT6*). Our findings expand considerably our understanding of the host genetic regulation of the microbiome composition and point toward the importance of key gastrointestinal physiological mechanisms in microbiome regulation. Identified variants were located near or in genes linked to gastrointestinal physiology, such as enteroendocrine fatty acid chemosensing, bile composition, mucosal composition and presentation and secretion of cell surface glycans.

The strengths of this study include harmonized bioinformatic processing across cohorts, strict Bonferroni adjustment of the genome-wide threshold to limit false positives and consistent replication in the Norwegian HUNT study. Limitations include the focus on participants of European ancestry, mainly from Nordic countries, restricting generalizability and limited power to detect associations with rare variants or less prevalent microbial species. All study-wide associations were for species present in at least 27% of participants, whereas most gut species are less common. Another limitation was incomplete genomic coverage downstream of *MUC12* on Chr. 7 in the reference genome used for genotyping, which hindered exploration of that locus. As in most GWAS, identifying causal genes remains challenging.

Future work should address these limitations and clarify causal pathways linking host genetics and the microbiome. We expect larger GWAS to continue highlighting genes related to gastrointestinal physiology and to factors known to shape the microbiome, such as antibiotics, cardiometabolic medication and diet[53–55]. They may also uncover more species–locus associations, as suggested by our 149 genome-wide findings, where several loci were linked to several species. In conclusion, our study advances understanding of the host genetic determinants of gut microbiome composition and highlights gastrointestinal physiology as a key driver.

## Online content

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

**Koen F. Dekkers** [1,30], **Kamalita Pertiwi** [1,30], **Gabriel Baldanzi** [1,30], **Per Lundmark** [1], **Ulf Hammar**[1], **Marta Riise Moksnes** [2], **Eivind Coward** [3], **Maria Nethander** [2], **Ghassan Ali Salih** [4,5], **Mariam Miari**[6], **Diem Nguyen** [1], **Sergi Sayols-Baixeras**[1,7,8], **Aron C. Eklund** [9], **Jacob Bak Holm** [9], **H. Bjørn Nielsen** [9], **Camila Gazolla Volpiano**[10,11], **Guillaume Méric** [1,10,11,12,13], **Manonanthini Thangam** [6], **Liisa Hakaste**[14,15], **Tiinamaija Tuomi**[6,14,15,16], **Emma Ahlqvist** [6,15], **Christopher A. Smith**[17], **Marie Allen**[4], **Frank Reimann** [17], **Fiona M. Gribble** [17], **Claes Ohlsson** [2,18], **Kristian Hveem**[3,19,20], **Olle Melander**[6,21], **Peter M. Nilsson** [6], **Gunnar Engström**[6], **J. Gustav Smith**[22,23,24,25], **Karl Michaëlsson**[7], **Johan Ärnlöv**[26,27,28], **Marju Orho-Melander** [6] & **Tove Fall** [1,29] ✉

[1]Department of Medical Sciences, Molecular Epidemiology, Uppsala University, Uppsala, Sweden. [2]Department of Internal Medicine and Clinical Nutrition, Institute of Medicine, Sahlgrenska Osteoporosis Centre, Centre for Bone and Arthritis Research at the Sahlgrenska Academy, University of Gothenburg, Gothenburg, Sweden. [3]Department of Public Health and Nursing, HUNT Center for Molecular and Clinical Epidemiology, Norwegian University of Science and Technology, Trondheim, Norway. [4]Department of Immunology, Genetics and Pathology, Uppsala University, Uppsala, Sweden. [5]Department of Biology, College of Science, University of Baghdad, Baghdad, Iraq. [6]Department of Clinical Sciences in Malmö, Lund University, Malmö, Sweden. [7]Department of Surgical Sciences, Medical Epidemiology, Uppsala University, Uppsala, Sweden. [8]CIBER Cardiovascular diseases (CIBERCV), Instituto de Salud Carlos III, Madrid, Spain. [9]Cmbio, Copenhagen, Denmark. [10]Cambridge Baker Systems Genomics Initiative, Baker Heart and Diabetes Institute, Melbourne, Victoria, Australia. [11]Department of Cardiometabolic Health, University of Melbourne, Melbourne, Victoria, Australia. [12]Department of Cardiovascular Research, Translation, and Implementation, La Trobe University, Melbourne, Victoria, Australia. [13]Department of Life Sciences, University of Bath, Bath, UK. [14]Folkhälsan Research Center, Helsinki, Finland. [15]Finnish Institute for Molecular Medicine (FIMM) and Research Program for Clinical and Molecular Medicine Diabetes and Obesity, Helsinki University, Helsinki, Finland. [16]Endocrinology, Abdominal Center, Helsinki University Central Hospital, Helsinki, Finland. [17]Institute of Metabolic Science-Metabolic Research Laboratories & MRC-Metabolic Diseases Unit, University of Cambridge, Cambridge, UK. [18]Department of Drug Treatment, Sahlgrenska University Hospital, Region Västra Götaland, Gothenburg, Sweden. [19]HUNT Research Centre, Norwegian University of Science and Technology, Levanger, Norway. [20]Levanger Hospital, Nord-Trøndelag Hospital Trust, Levanger, Norway. [21]Department of Internal Medicine, Skåne University Hospital, Malmö, Sweden. [22]The Wallenberg Laboratory/Department of Molecular and Clinical Medicine, Institute of Medicine, Gothenburg University and Science for Life Laboratory, Gothenburg, Sweden. [23]Department of Cardiology, Sahlgrenska University Hospital, Gothenburg, Sweden. [24]Department of Cardiology, Clinical Sciences, Wallenberg Center for Molecular Medicine and Lund University Diabetes Center, Lund University, Lund, Sweden. [25]Department of Cardiology, Skåne University Hospital, Lund, Sweden. [26]Department of Neurobiology, Care Sciences and Society, Division of Family Medicine and Primary Care, Karolinska Institutet, Huddinge, Sweden. [27]School of Health and Social Studies, Dalarna University, Falun, Sweden. [28]Center of Clinical Research, Region Dalarna, Falun, Sweden. [29]Department of Medical Sciences, Science for Life Laboratory, Uppsala University, Uppsala, Sweden. [30]These authors contributed equally: Koen F. Dekkers, Kamalita Pertiwi, Gabriel Baldanzi. ✉e-mail: tove.fall@medsci.uu.se

## Methods

### Ethical considerations
The current study has been approved by the Swedish Ethical Review Authority (DNR 2022-06137-01, DNR 2024-01992-02). All participants in the respective studies below provided written informed consent. The Swedish Ethical Review Board approval numbers are: SCAPIS (DNR 2010-228-31M), SIMPLER (DNR 2009/2066-32, DNR 2009/1935-32, DNR 2010/0148-32, DNR 2014/892-31/3), MDC (DNR 532/2006, DNR 51-90) and MOS (DNR 2012-594). The PPP-Botnia study received approval from the Ethics Committee of Helsinki University (approval number 608/2003). The HUNT study was approved by the local ethical review board (Regional committee for medical and health research ethics, Central Norway; REK-656785).

### Discovery studies
**SCAPIS.** SCAPIS[56] is a multicenter cohort comprising 30,154 people aged 50–65 years. For this analysis, 8,733 participants of European ancestry from the Malmö and Uppsala sites with both gut microbiome and genotype data were included. At baseline, participants provided blood samples during the first visit and were asked to collect stool samples at home, storing them at −20 °C until samples were brought to the study center at the second visit for storage at −80 °C. DNA extracted from whole blood was used for genotyping. Birth year and sex were obtained from the Swedish population register. Information on dispensed antibiotics (Anatomical Therapeutic Chemical code J01) in the past 6 months was obtained from the Swedish Prescribed Drug Register. BMI was defined as weight divided by height squared (kg m$^{-2}$). Habitual alcohol and fiber intakes were estimated from a food frequency questionnaire (g day$^{-1}$)[57]. Smoking behavior was assessed using a questionnaire and defined as current, former and never smoker.

**SIMPLER-Västmanland and SIMPLER-Uppsala.** The Swedish Infrastructure for Medical Population-Based Life-Course and Environmental Research (SIMPLER; https://www.simpler4health.se/w/sh/en) includes data from two large, ongoing population-based studies: the Cohort of Swedish Men (COSM) and the Swedish Mammography Cohort (SMC)[58]. The COSM initially enrolled 48,850 men born between 1918 and 1952 living in Västmanland and Örebro counties in 1997. The SMC enrolled 66,651 women by sending invitations to all women born between 1914 and 1948 living in Uppsala and Västmanland counties between 1987 and 1990. The current analysis is based on a subsample selected randomly from these studies who were invited for clinical examination with genotype and gut microbiome data: SIMPLER-Västmanland (SIMPLER-V) and SIMPLER-Uppsala (SIMPLER-U). SIMPLER-V includes 4,515 COSM and SMC participants from Västmanland examined between 2010 and 2019. SIMPLER-U includes 981 women from the county of Uppsala, examined between 2003 and 2009 (no stool collected) and re-examined between 2015 and 2019 (stool collected). Participants were asked to collect stool samples at home and store them at −20 °C until they were brought to the test center, where samples were stored at −80 °C. For 115 SIMPLER-V participants, the examination was conducted at home. DNA for genotyping was extracted from whole-blood samples. Information on dispensed antibiotics in the past 6 months was obtained from the Swedish Prescribed Drug Register.

**Malmö offspring study.** The Malmö offspring study (MOS) includes participants aged ≥18 years who are children or grandchildren of participants from the Malmö Diet and Cancer Study (MDC)−cardiovascular cohort, a subset of the larger MDC[59]. Data collection in MOS began in 2013 and included 4,721 participants by 2020. The current study included 1,788 participants with genotype and gut microbiome data who attended baseline measurements between 2013 and 2017. Stool samples were collected and stored in home freezers (−20 °C) until they were brought to the study sites, where they were stored at −80 °C

in the biobank. DNA for genotyping was extracted from whole-blood samples. Demographic information was collected using a questionnaire. Antibiotic use was self-reported and was also derived from the Swedish Prescribed Drug Register. Participants who were also part of SCAPIS were excluded from the MOS data.

### Replication cohort
**Norwegian Trøndelag Health Study.** The Trøndelag Health (HUNT) study is a long-term population-based health investigation conducted in the Trøndelag county, Norway[60,61]. Four surveys have been used to collect data and biological samples from participants between 1984 and 2019. Approximately 230,000 people have participated in at least one survey. Of these, around 88,000 participants have undergone genotyping[62]. Among the 56,042 participants in the HUNT4 survey, 13,268 submitted stool samples for gut microbiome analysis on a filter paper. We included data from 12,652 HUNT4 participants of European descent having both genetic and gut microbiome data available. Sequencing and bioinformatic processing were performed analogously to SCAPIS and MOS at Cmbio (Copenhagen, Denmark).

BMI and age distribution were compared between studies with density plots. A map depicting the study sites was generated with the maps v.3.4.2.1 R package. Other studies (MDC, PPP-Botnia) are described in the Supplementary Note.

### Genetic analysis
**Genotyping and imputation.** DNA extraction, genotyping, pre-imputation quality control and imputation were performed separately in each cohort (SCAPIS, SIMPLER, MOS and HUNT) using high-density Illumina genotyping arrays and standard pipelines for variant calling and quality filtering. Quality control steps removed samples with poor genotyping quality, sex discrepancies, non-European ancestry and markers with high missingness or implausible allele frequencies. Imputation was performed using standard algorithms (EAGLE, minimac, PBWT) at established imputation servers against the Haplotype Reference Consortium (HRC) r1.1 panel. Detailed protocols for each cohort are provided in the Supplementary Note.

**Validation of genotypes using Sanger sequencing.** Direct genotyping using Sanger sequencing was performed to confirm the variants in rs10836441 (OR51E1−OR51E2 locus) and rs4556017 (MUC12 locus). Details are given in the Supplementary Note.

### Stool DNA extraction and metagenomic sequencing
**SCAPIS, MOS and HUNT.** Stool DNA extraction and quality control for SCAPIS and MOS were performed by Cmbio and described in Sayols-Baixeras et al.[63]. In brief, samples were randomized on the box level, and DNA was extracted using the NucleoSpin 96 Soil extraction kit (Macherey−Nagel). DNA extraction quality was evaluated using agarose gel electrophoresis. One negative and one positive (mock) control were added to each batch. DNA was quantified with fluorometric techniques both after DNA extraction and after library preparation. DNA extraction and quality control in samples from HUNT have been described in detail in Grahnemo et al.[64]. In brief, three 6-mm disks were punched out from each filter card into a well. DNA was isolated using the Microbiome MagMAX Ultra kit (Thermo Fisher Scientific) after bead-beating. For all three studies, genomic DNA was fragmented and used for library construction using the NEBNext Ultra Library Prep Kit from Illumina. The prepared DNA libraries were purified and evaluated for fragment size distribution. Libraries from stool DNA were sequenced using the Illumina Novaseq 6000 instrument using 2 × 150-base-pair paired-end reads, generating on average 26.0, 25.3 and 22.9 million read pairs, respectively, in SCAPIS, MOS and HUNT, with 97.8% of the sequenced bases having Phred quality score >20 in SCAPIS and MOS, and more than 85% had a Phred quality score ≥30 in HUNT.

**SIMPLER study.** SIMPLER stool samples were thawed, a pea-size amount was aliquoted, and 800 µl of DNA/RNA Shield (Zymo Research) was added. These aliquots were refrozen and sent to the Centre for Translational Microbiome Research at the Karolinska Institute in Stockholm, Sweden for DNA extraction and metagenomic sequencing. DNA was extracted with the MagPure Stool kit (Magen Biotechnology). Each batch had one negative (DNA/RNA Shield) and one positive control (Zymo mock). Stool DNA was fragmented and used for library construction using the MGI Easy FS DNA Library Prep Set kit. The prepared DNA libraries were evaluated with a TapeStation D1000 kit (Agilent), and the quantity was determined by QuantIT HighSensitivity dsDNA Assay on a Tecan Spark (Tecan). Equimolarly pooled libraries were circularized using the MGI Easy Circularization kit (MGI Tech) and sequenced using 2 × 150 bp paired-end reads on the DNBSEQ G400 or T7 sequencing instrument (MGI) with an average yield of 51 million reads/sample.

## Microbial taxonomic profiling

Read pairs mapped to the human reference genome GRCh38.p14 were removed using Bowtie2 (v.2.4.2)[65] in SCAPIS, MOS and HUNT, and against GRCh38 using Kraken 2 (ref. [66]) in SIMPLER. Remaining bioinformatic processing, calculation of relative abundances and microbial taxonomic annotation were performed for all studies, including HUNT, at Cmbio using the CHAMP profiler based on the Human Microbiome Reference HMR05 catalog[12] (Supplementary Note). The taxonomic annotation was based on the Genome Taxonomy Database (GTDB) release 214 (release date: 28 April 2023). A rarefied species abundance table was produced by random sampling, without replacement, of 190,977 gene counts per sample in SCAPIS and MOS, and 641,964 gene counts per sample in SIMPLER. In total, 4,248 species were detected in the rarefied data in SCAPIS, 3,430 in MOS and 4,192 in SIMPLER-V, and 3,523 in SIMPLER-U. The alpha diversity measures—Shannon index, inverse Simpson index and richness—were calculated using rarefied data with the diversity function of the vegan R package (R v.4.3.1). Only the 921 species with prevalence >5% in all four studies were kept for the species-level analyses. Those detected in fewer than 50% of samples in at least one cohort based on nonrarefied data were converted into a binary present/absent variable. Those detected in more than 50% of samples in all four studies were rank-based inverse normal (RIN) transformed. Alpha diversity measures were also RIN-transformed, and, for significant findings, were also analyzed on a nontransformed scale for increased interpretability. The RIN transformation was performed separately for each cohort.

## Analysis of scRNA-seq data

Gene expression data in cells derived from human duodenum, ileum and colon were obtained from Hickey et al.[22], and mean gene expression was generated per their annotated clusters. The expression in EECs from human duodenal and ileal organoids was assessed as described[23]. Briefly, a yellow fluorescent protein was inserted downstream of the Chromogranin A promoter by CRISPR–Cas9 to label EECs. Fluorescent EECs were then isolated using flow cytometry and analyzed by 10× scRNA-seq. Gene expression in EECs from the murine gastrointestinal tract was analyzed with scRNA-seq, as described in Smith et al.[24].

## Statistical analysis

### GWAS of microbiome composition.
GWAS was performed separately for microbial alpha diversity and 921 species using REGENIE[67] v.3.3 for each cohort (SCAPIS, SIMPLER-V, SIMPLER-U, MOS). A subset of the genotype datasets was created for the first REGENIE step to fit whole-genome regression models including only quality-controlled directly genotyped SNPs with MAF > 1% and Hardy-Weinberg equilibrium $P < 1 \times 10^{-15}$. For the second step, all variants with an information score >0.7 were included in association analyses performed using logistic regression for binary variables and genetic variants with MAF > 5% in all four cohorts, and linear regression for RIN-transformed variables and genetic variants with MAF > 1% in all four cohorts. Covariates were sex, age, age2, plate and genetic principal components (PC) 1–10. The PCs were calculated in unrelated samples, separately for each cohort, with PLINK[68] using an LD-pruned dataset, and all samples were then projected onto these components. In SCAPIS and MOS, plate represents metagenomics DNA extraction plate, whereas in SIMPLER it means the metagenomic aliquoting plate. Plate, age and sex were included to increase precision and power. For SCAPIS, the site was accounted for by the plate variable because plates were nested into the site variable. Based on previous nonlinear associations between age and microbiome[69] and our results from a naive linear model for the association between age and microbial species, we opted to include age also as age[2]. REGENIE accounts for population stratification, but to account for any residual bias, we also included genetic PCs 1–10 in the model[70]. Cohort-specific results were meta-analyzed using the inverse-variance weighted fixed-effects method in METAL[71] v.2011-03-25. Independent loci were determined using LD clumping ($r^2$ 0.001, window 10 Mb) in PLINK[68] v.2.00-alpha-5-20230923 with SCAPIS dosages used to determine the correlation structure. Variant-alpha diversity associations with $P < 1.7 \times 10^{-8}$ and variant-species associations with $P < 5.4 \times 10^{-11}$ were considered study-wide-significant. This threshold was based on a Bonferroni correction of the conventional genome-wide threshold of $5 \times 10^{-8}$ for three alpha diversity metrics and 921 species tested. Confidence intervals for the $I^2$ statistic were calculated using the metagen function of the meta v.6.5-0 R package. The loci were annotated using the Open Targets Genetics[72] v.22.10 database (variant index, variant to gene and variant to trait annotations). Heritability was determined using SumHer[73] v.6 according to the GCTA heritability model, with SCAPIS dosages used to determine the correlation structure.

### Sensitivity analyses.
Sensitivity analyses were performed for the 149 genome-wide locus-species associations by (1) excluding participants with antibiotic use in the 6 months before sampling, (2) excluding participants with self-reported IBD, (3) retaining an unrelated subset where no participant had third degree relatedness or closer with any other participant using a KING-robust kinship estimator threshold of 0.0442, (4) retaining one random spouse in SIMPLER and one random participant living at the same address in MOS to assess cohabitation (SCAPIS was removed for this analysis), (5) using centered log ratio plus RIN transformation for species analyzed using linear regression, (6) using Firth correction for species analyzed using logistic regression, (7) removing age[2] from the covariates, (8) analyzing SCAPIS-Uppsala and SCAPIS-Malmö as two separate cohorts in the meta-analysis and (9–12) adding BMI, alcohol intake, smoking or fiber intake, respectively, as covariates. The analyses adding alcohol, smoking and fiber were performed in SCAPIS only, where data on these variables were nearly complete.

### External replication.
Associations passing the study-wide threshold were assessed in HUNT by applying the same models as in the Swedish cohorts and using REGENIE with the same model specifications. We further assessed the validity of our findings using summary statistics from the published FINRISK[9] and Dutch Microbiome Project[7] studies. Details are given in the Supplementary Note.

### GWAS of higher taxa.
We also performed GWAS of 455 genera, 106 families, 50 orders, 21 classes, 17 phyla and 3 superkingdoms. Relative abundances were created for these higher-level taxa by summation of their respective species-level relative abundances. The 364 taxa detected in 5–50% of samples in each cohort were analyzed using logistic regression (absence/presence), and 288 taxa with prevalence >50% were analyzed using RIN-transformed relative abundances and linear regression. Study-wide significance was considered at $P < 5.4 \times 10^{-11}$, the same level as for species.

**GWAS of functional modules.** Functional gut metabolic and gut–brain modules were attributed to species that contained at least two-thirds of the genes needed for the functionality of that module. If an alternative reaction pathway within a module existed, only one such pathway was required. All reaction pathways were required for modules with fewer than four steps. Module abundances were defined as the sum of the relative abundances of all species in a module. Similar to the GWAS of the species, two modules detected in 5–50% of samples in each cohort were analyzed using logistic regression (absence/presence) and 115 modules with prevalence >50% were analyzed using RIN-transformed relative abundances and linear regression. Study-wide significance was considered at $P < 4.3 \times 10^{-10}$.

**Interaction analysis for ABO, secretor status and Lewis blood groups.** Blood groups A, B, AB and O were determined based on allele combinations of *ABO* genetic variants rs505922 and rs8176746 (ref. 74), secretor status based on *FUT2* genetic variant rs601338 (ref. 75) and Lewis status (positive, negative) based on allele combinations of *FUT3* variants rs812936, rs28362459 and rs3894326 (ref. 75). Blood groups A and AB were combined into antigen A, and blood groups B and AB into antigen B. Mixed models were run for each cohort with species associated with *ABO*, *FUT2* or *FUT3–FUT6* at the study-wide significance level as outcome using the lmer (for species assessed with linear regression in the GWAS) and glmer (for species assessed with logistic regression in the GWAS) functions of the lmerTest v.3.1-3 R package. The interaction between antigen (ABO A, B or Lewis) and secretor status was estimated with covariates sex, age, age², plate and genetic PCs 1–10. First-degree relatedness, determined by KING[76] kinship coefficient ≥0.177, was used as a random effect. For the logistic mixed models, random and fixed effects coefficients were optimized in the penalized iteratively reweighted least squares step (setting nAGQ = 0). Cohort-specific results were meta-analyzed with the rma function of the metafor v.4.4-0 R package using the fixed-effect inverse-variance weighted method. Study-wide significance was considered at Bonferroni-corrected $P < 3.3 \times 10^{-3}$.

**GWAS of GLP-1.** After overnight fasting, GLP-1 levels were measured in MDC and PPP-Botnia study participants (Supplementary Note) before and 2 h after a 75-g oral glucose load. GWAS of GLP-1 was performed in 2,588 people with fasting and 2,613 with 2-h GLP-1 in MDC, and in 926 people with fasting and 898 with 2-h GLP-1 in PPP-Botnia. GLP-1 levels were log-transformed before analysis. SNPTEST[77] v.2.5.6 was used for genome-wide association analyses, using the frequentist score method adjusted for age, sex and the genetic PC1-4. Results were filtered based on MAF > 0.01, Hardy-Weinberg equilibrium $P > 5 \times 10^{-7}$, and imputation info scores >0.4. A fixed-effect meta-analysis was performed using GWAMA[78].

**Functional mapping.** Genetic variants associated with microbial alpha diversity or species at the genome-wide significant level were mapped to functional pathways using FUMA[26] v.1.5.2. One (out of 2,353) variant without an rsID was removed. If a genetic variant was associated with several traits or was multiallelic, the trait or allele pair with the lowest $P$ was used as input.

**Colocalization.** Pairwise colocalization analyses were performed to investigate whether microbial richness and the eight study-wide significant species colocalized in the identified study-wide significant loci and with sex hormone binding globulin, WHRadjBMI, LDL cholesterol, IBD, glucose and stool frequency. Details are provided in the Supplementary Note.

**Mendelian randomization.** We performed two-sample MR analyses to investigate bidirectional effects between specific species (*C. saudiense*, *T. sanguinis*, *Intestinibacter* sp9005540355) and BMI, WHR and LDL cholesterol. Details are provided in the Supplementary Note.

## Plasma metabolomics
The plasma metabolomics analysis in SCAPIS has been described elsewhere[79] and in the Supplementary Note. Associations of genetic variants with plasma metabolites were analyzed using the same REGENIE pipeline as for the microbiome, adjusting for age, age², sex, delivery batch and genetic PCs 1–10. Metabolites detected in fewer than 100 samples were excluded. Those detected in 5–50% of samples were analyzed by logistic regression, and those in ≥50% of samples were RIN-transformed and analyzed by linear regression. We report one lead SNP per study-wide locus; when several species were associated, we selected the lead SNP among those replicated in HUNT, prioritizing the lowest $P$ value in Swedish cohorts. FDR correction (Benjamini–Hochberg) of 5% was applied.

## Stool metabolomics
To find stool metabolites associated with the study-wide significant loci, we downloaded GWAS of stool metabolites summary statistics (only $P < 10^{-5}$ available) from Zierer et al.[80] (Supplementary Table 16) and lifted the genomic coordinates over to GRCh37 using Ensembl Variation 112 for variants with an rsID and https://genome.ucsc.edu/cgi-bin/hgLiftOver for variants without an rsID. Genetic variants that could not be lifted over were removed (247 out of 46,765). We assessed the same lead variants per study-wide locus as described for the genetic association with plasma metabolites. A lookup was performed for genetic variants within 100 kb of the locus region corresponding to the study-wide significant lead variant.

## Short-chain fatty acids
In MOS, a panel of nine plasma SCFAs was measured[81]. Laboratory method for SCFA measurement is described in the Supplementary Note. The association of genetic variants with SCFAs was assessed with the same REGENIE pipeline as described above for the microbiome, with age, age², sex, SCFA measurement batch and genetic PCs 1–10 as covariates. SCFAs were RIN-transformed and assessed using linear regression. We assessed the same lead SNPs per study-wide locus as described for the genetic association with plasma metabolites. FDR correction (Benjamini–Hochberg) of 5% was applied.

## Reporting summary
Further information on research design is available in the Nature Portfolio Reporting Summary linked to this article.

## Data availability
Complete GWAS summary statistics are available in the GWAS catalog with accession numbers GCST90670368 to GCST90671939. De-hosted anonymized metagenomic sequencing data from SCAPIS used in this study can be found at the European Nucleotide Archive under accession number PRJEB51353. scRNA-seq data are available in the GEO repository with accession numbers GSE284419 and GSE269778, and on Dryad (https://doi.org/10.5061/dryad.8pk0p2ns8). The metagenomics, metabolomics and genetic data supporting the conclusions of this article were provided by the SCAPIS, SIMPLER and MOS central data offices, and are not shared publicly due to confidentiality and ethical restrictions. Data will be shared by the respective data offices only after permission from the Swedish Ethical Review Authority (https://etikprovningsmyndigheten.se) and from the respective boards (https://www.scapis.org/data-access, https://www.simpler4health.se and https://www.malmo-kohorter.lu.se/malmo-offspring-study-mos).

## Code availability
We used publicly available software for the analysis, as described in Methods. The code for the analyses presented in this paper is available via GitHub at https://github.com/MolEpicUU/GWAS_scripts for the GWAS pipeline via Zenodo at https://doi.org/10.5281/zenodo.16947117

(ref. 82), and https://github.com/MolEpicUU/GWAS_microbiome for the meta-analysis and post-GWAS analyses scripts available via Zenodo at https://doi.org/10.5281/zenodo.16925644 (ref. 83).

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

## Acknowledgements

We would like to acknowledge the help of Biobank Sweden and the local biobank facilities for their services in handling of biological samples and biobanking related to SCAPIS. We would like to acknowledge the Karolinska Institute Biobank for their services regarding DNA extraction. Genotyping of SCAPIS samples was performed by the SNP&SEQ Technology Platform in Uppsala. The facility is part of NGI Sweden and Science for Life Laboratory. The SNP&SEQ Platform is also supported by the Swedish Research Council and the Knut and Alice Wallenberg Foundation. We acknowledge the SIMPLER national research infrastructure (https://www.simpler4health.se/w/sh/en) for generating and making available data, computational facilities and resources. With regards to the Botnia Study, the role of Leif. Groop in designing the Botnia Study and the skillful assistance of the Botnia Study Group is gratefully acknowledged. The scRNA-seq results are in part based upon data generated by the NIH Human BioMolecular Atlas Program (HuBMAP). The computations and data handling in the Swedish studies were enabled by resources in project sens2019512 and simp2023007 provided by the National Academic Infrastructure for Supercomputing in Sweden (NAISS) and the Swedish National Infrastructure for Computing (SNIC) at Uppsala Multidisciplinary Center for Advanced Computational Science (UPPMAX), funded by the Swedish Research Council through grant nos. 2022-06725 and 2018-05973. Financial support was obtained in the form of grants from the European Research Council (ERC-STG-2018-801965 (T.F.); ERC-STG-2015-679242 (J.G.S.); ERC-COG-2014-649021 (M.O.-M.); ERC-ADG-2022-101096347 (C.O.)), the Swedish Heart and Lung Foundation (Hjärt-Lungfonden, 2023-0687 (T.F.); 2018-0343 (J.Ä.); 2024-0486 (J.Ä.); 2020-0711 (M.O.-M.); 2020-0173 (G.E.); 2022-0606 (E.A.); 2022-0344 (J.G.S.)), the Swedish Research Council (VR, 2019-01471 (T.F.); 2025-02673 (T.F.); 2018-02784 (M.O.-M.); 2018-02837 (M.O.-M.); 2019-01015 (J.Ä.); 2020-00243 (J.Ä.); 2020-02191 (E.A.); 2020-01392 (C.O.); 2019-01291 (K.M.) and Strategic Research Area Exodiab 2009-1039 (M.O.-M.); 2019-01236 (G.E.); 2021-02273 (J.G.S.)), The Swedish Foundation for Strategic Research, LUDC-IRC; IRC15-0067 (M.O.-M.), the Göran Gustafsson Foundation for Research in Natural Sciences and Medicine (T.F.), the Swedish state under the agreement between the Swedish government and the county councils, the ALF agreement (2018-0148 and 2022-0258 (M.O.-M.); ALFGBG-720331, ALFGBG-965235 and ALFGBG-965744 (C.O.)), the Novo Nordisk Foundation (NNF, 20OC0063886 (M.O.-M.); 19OC0055250 and 22OC0078421 (C.O.)), the Knut and Alice Wallenberg Foundation (KAW 2015.0317, C.O.), the Lundberg Foundation (LU2021-0096, C.O.), the Clinical Research Center, Region Dalarna (CKFUU-1025348, 987986, 976460, 963488, 936407, 695401 and 797891, J.Ä.), the Swedish Diabetes Foundation (DIA 2018-375, M.O.-M.), Wellcome (220271/Z/20/Z (F.M.G., F.R.)) and UK Medical Research Council (MRC_MC_UU_12012/3 (F.M.G., F.R.)). G.M. and C.G.V. are supported by Australian NHMRC grant

GNT2013468. The Malmö Offspring Study (MOS) was funded by the Swedish Research Council (VR, 521-2013-2756 (P.M.N.)), the Swedish Heart and Lung Foundation (Hjärt-Lungfonden 20150427 (P.M.N.)) and by the Swedish state under the agreement between the Swedish government and the county councils, the ALF agreement (P.M.N). This research has been conducted using the Swedish CArdioPulmonary bioImage Study (SCAPIS) Resource, under Petition Number PETITION-507. The main funding body of SCAPIS is the Swedish Heart and Lung Foundation. The study is also funded by the Knut and Alice Wallenberg Foundation, the Swedish Research Council, VINNOVA (Sweden's Innovation agency), the University of Gothenburg and Sahlgrenska University Hospital, Karolinska Institutet and Region Stockholm, Linköping University and University Hospital, Lund University and Skåne University Hospital, Umeå University and University Hospital, Uppsala University and University Hospital. SIMPLER receives funding through the Swedish Research Council under grant nos. 2017-00644, 2017-06100 and 2021-00160 (to Uppsala University and K.M.), from ALF through the local Region Uppsala County Council and from the local Region Västmanland County Council. The PPP-Botnia Study has been supported financially by grants from Folkhälsan Research Foundation, the Sigrid Juselius Foundation, The Academy of Finland (grant nos. 263401, 267882, 312063, 336822, 312072 and 336826), University of Helsinki, Nordic Center of Excellence in Disease Genetics, EU (EXGENESIS, MOSAIC FP7-600914), Ollqvist Foundation, Swedish Cultural Foundation in Finland, Finnish Diabetes Research Foundation, Foundation for Life and Health in Finland, Finnish Medical Society, State Research Funding through the Helsinki University Hospital, Perklén Foundation, Närpes Health Care Foundation and Ahokas Foundation. The study has also been supported by the Ministry of Education in Finland, the Municipal Heath Care Center and Hospital in Jakobstad and Health Care Centers in Vasa, Närpes and Korsholm. The PPP-Botnia Study was genotyped by the FinnGen project. The Trøndelag Health Study (HUNT) is a collaboration between HUNT Research Centre (Faculty of Medicine and Health Sciences, Norwegian University of Science and Technology NTNU), Trøndelag County Council, Central Norway Regional Health Authority and the Norwegian Institute of Public Health.

## Author contributions

K.F.D., G.B., K.P., M.O.-M. and T.F. planned and designed the study. O.M., P.M.N., G.E., K.M., J.Ä., J.G.S., M.O.-M. and T.F. collected data from Swedish studies. U.H. performed simulation study for model specifications. K.F.D., P.L. and G.B. performed the analysis of Swedish studies concerning genetics, microbiome and metabolome. K.H. and C.O. provided replication data from the HUNT study. M.R.M., E.C. and M.N. analyzed replication data. C.A.S., F.R. and F.M.G. performed expression analysis. G.A.S. and M.A. performed Sanger sequencing. A.C.E., J.B.H., H.B.N., M.M. and S.S.-B. performed metagenomic bioinformatic analyses. C.G.V. and G.M. provided specialist microbiological expertise and contributed to the interpretation of results. M.T., L.H., T.T. and E.A. analyzed GWAS of GLP-1. T.F., K.F.D., K.P., D.N. and G.B. wrote the first draft of the manuscript. All authors provided intellectual input, reviewed and approved the final version of the manuscript. T.F. is the guarantor of the study.

## Funding

## Competing interests

The funders had no role in study design, data collection and analysis, decision to publish or preparation of the manuscript. A.C.E., J.B.H. and H.B.N. are employees of Cmbio. J.Ä. has served on the advisory boards for Astella, AstraZeneca and Boehringer Ingelheim and has received lecturing fees from AstraZeneca, Boehringer Ingelheim and Novartis, all unrelated to the present work. P.M.N. has received lecture fees from Novartis, Novo Nordisk, Amgen and Boehringer Ingelheim. F.M.G. and F.R. received grant funding from Eli Lilly and AstraZeneca for other projects. C.O. is an applicant on filed patent applications on the effect of probiotics on bone metabolism. The other authors declare no competing interests.

## Additional information

**Extended data** is available for this paper at https://doi.org/10.1038/s41588-026-02512-2.

**Correspondence and requests for materials** should be addressed to Tove Fall.

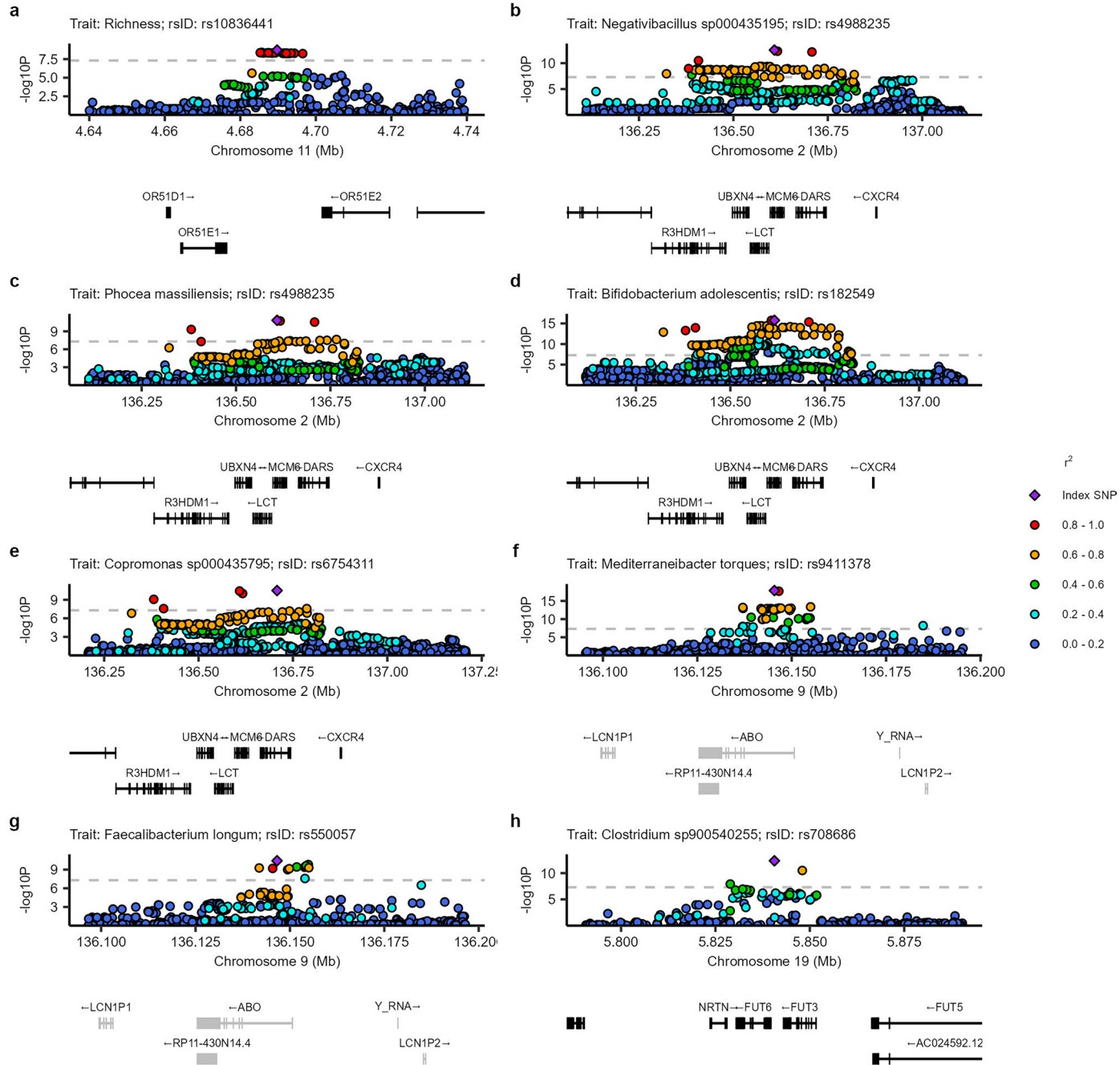

**Extended Data Fig. 1 | Regional association plots of variants in the *OR51E1-OR51E2*, *LCT*, *ABO*, and *FUT3-FUT6* gene loci. a-h**, Regional association plots of (**a**) richness with variants in the *OR51E1-OR51E2* gene locus (within a 100-kb window); (**b**) *Negativibacillus* sp000435195, (**c**) *Phocea massiliensis*, (**d**) *Bifidobacterium adolescentis*, and (**e**) *Copromonas* sp000435795 with variants in the *LCT* gene locus (within a 1-Mb window); (**f**) *Mediterraneibacter torques*, and (**g**) *Faecalibacterium longum* with variants in the *ABO* gene locus (within a 100-kb window); (**h**) *Clostridium* sp900540255 with variants in the *FUT3-FUT6*

gene locus (within a 100-kb window). The lead variant is indicated as the purple diamond. Other variants are indicated by dots colored according to the linkage disequilibrium (*r²*) values with the lead variant calculated using SCAPIS dosages. *P* values in **a**, **c**, **d**, **e**, and **f** were calculated using linear, and in **b** and **h** using logistic regression (two-sided tests). The horizontal dashed gray line indicates the genome-wide significance threshold ($-\log_{10}(5 \times 10^{-8})$). For the *ABO* locus, the non-coding genes are also plotted (in gray).

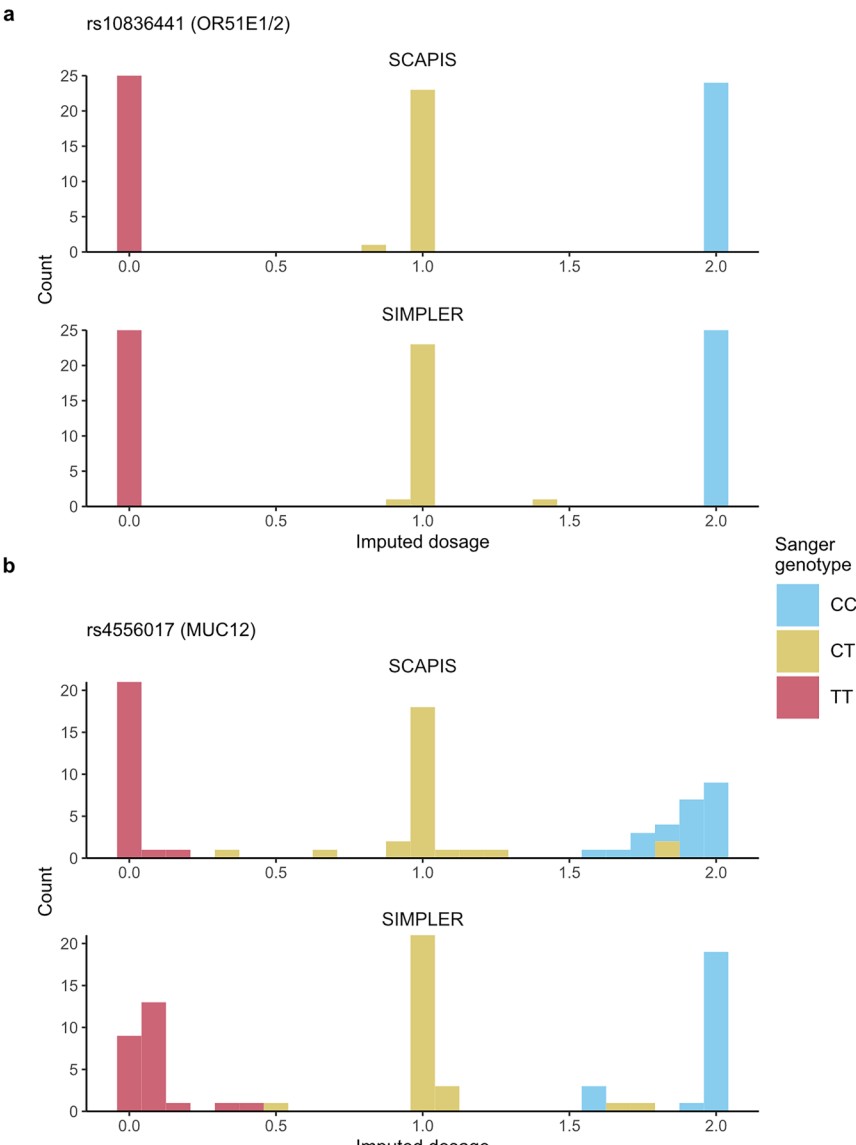

**a** rs10836441 (OR51E1/2)

**b** rs4556017 (MUC12)

**Sanger genotype**
- CC
- CT
- TT

**Extended Data Fig. 2 | Validation of imputed genotypes by Sanger sequencing.** **a**, Results from Sanger sequencing of 73 samples from the SCAPIS cohort and 75 samples from the SIMPLER-V cohort compared to the imputed genotype dosages for rs10836441. **b**, Results from Sanger sequencing of 73 samples from the SCAPIS cohort and 75 samples from the SIMPLER-V cohort compared to the imputed genotype dosages for rs4556017.

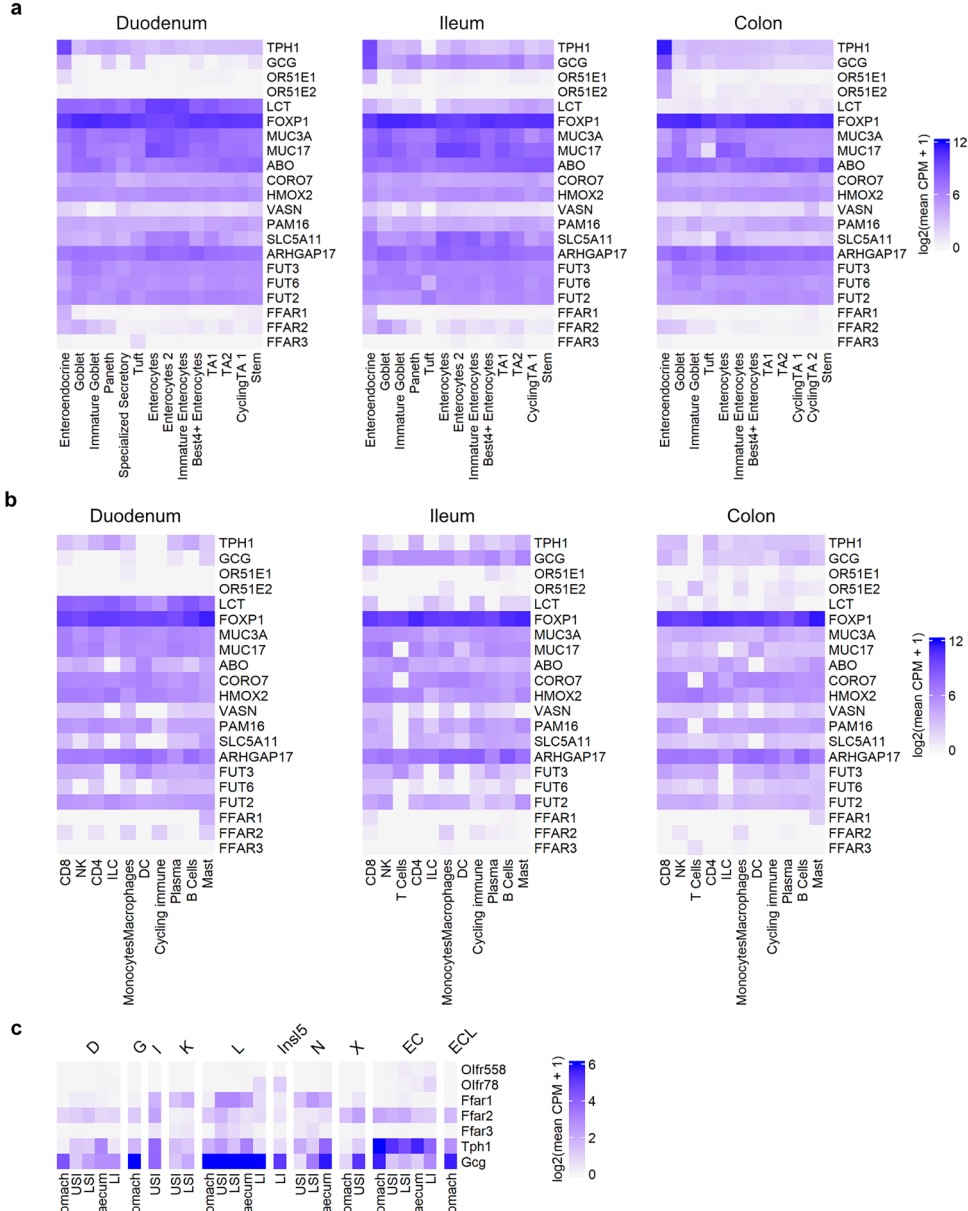

**Extended Data Fig. 3 | Single-cell expression analysis of candidate genes in human and mouse cells in intestinal tissues. a,b**, Single-cell expression analysis of candidate genes in human duodenal, ileal, and colonic epithelial (**a**) and immune (**b**) cells from donors. Heatmaps of mean gene expression were generated from the different intestinal epithelial and immune cell clusters in the dataset from Hickey et al.[22]. T-cells data were absent for duodenum. **c**, Single-cell expression in mouse enteroendocrine cells (EECs) from different regions of the gut[24]. Mean gene expression in different EEC clusters per gastrointestinal (GI) region. EECs were purified by flow cytometry from NeuroD1-Cre/YFP mice, and

analyzed by 10× single-cell RNA sequencing. Cells were clustered by *k*-means and annotated according to their expression of gut hormone genes: D (somatostatin), G (gastrin), I (cholecystokinin), K (glucose-dependent insulinotropic polypeptide), L (glucagon-like peptide 1 and peptide YY), Insl5 (insulin-like peptide 5), N (neurotensin), X (ghrelin), EC (enterochromaffin cells expressing Tph1 as a marker for serotonin biosynthesis), ECL (enterochromaffin-like cells expressing histidine decarboxylase as a marker for histamine biosynthesis). USI, upper small intestine; LSI, lower small intestine; LI, large intestine.

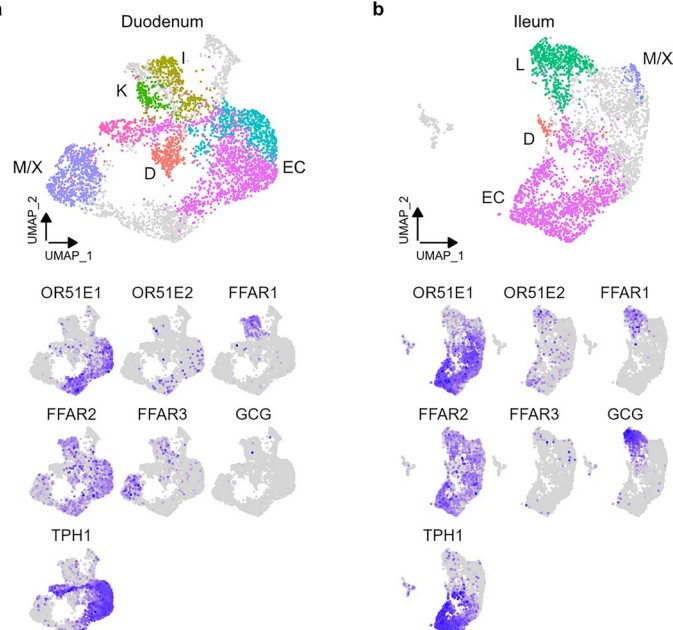

**Extended Data Fig. 4 | Single-cell expression in enteroendocrine cells from duodenal and ileal human-derived organoids. a,b,** Feature maps represent gene expression in human enteroendocrine cells (EEC) clusters from the duodenum (**a**) and ileum (**b**)[23]. EECs were labeled by inserting a yellow fluorescent protein downstream of the Chromogranin A promoter in organoids, by CRISPR-Cas9. Fluorescent cells were purified by flow cytometry and analyzed by 10× single-cell RNA sequencing. Cells were clustered by *k*-means and annotated according to their expression of gut hormone genes: D (somatostatin), I (cholecystokinin), K (glucose-dependent insulinotropic polypeptide), L (glucagon-like peptide 1 and peptide YY), M/X (motilin and ghrelin), EC (enterochromaffin cells expressing TPH1 as a marker for serotonin biosynthesis).

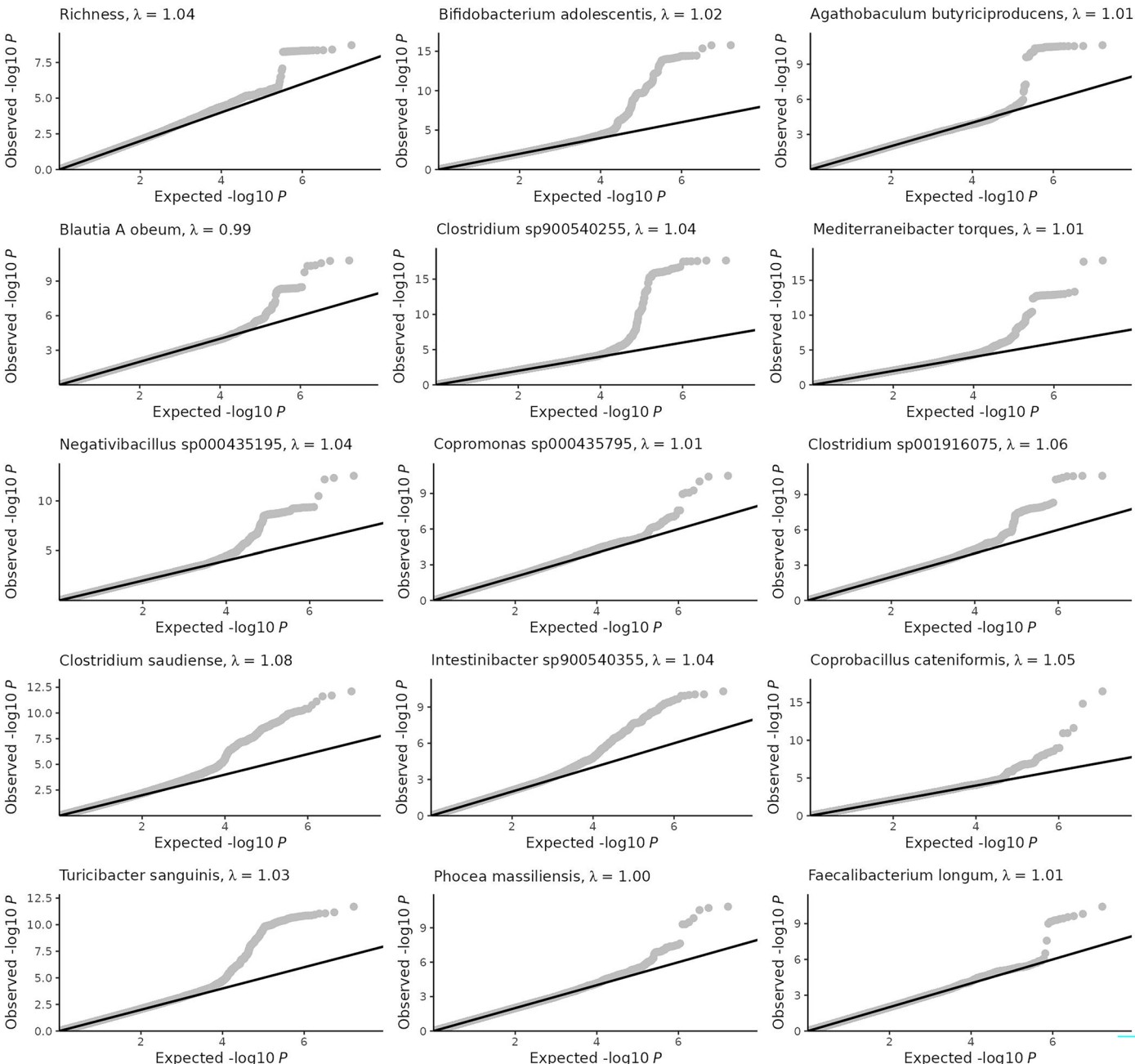

**Extended Data Fig. 5 | Quantile-quantile plots.** Quantile-quantile plots and genomic inflation factors for the GWAS of microbiome features (richness and species) with study-wide significant findings. Observed *P* values were from linear or logistic models (two-sided tests).

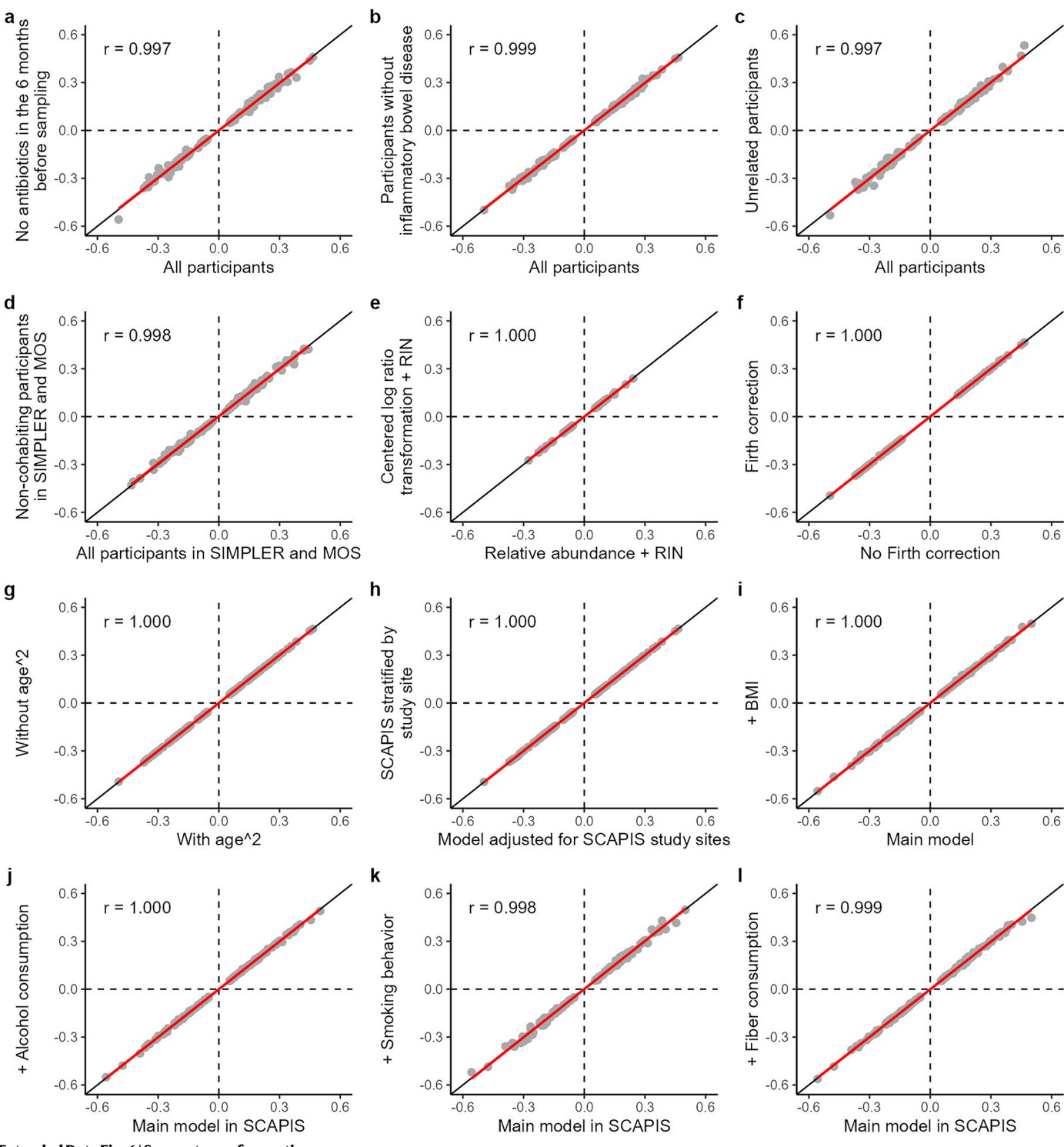

**Extended Data Fig. 6 | See next page for caption.**

**Extended Data Fig. 6 | Robustness of genetic microbiome associations across subgroups and model specifications. a-c**, Sensitivity analysis of 149 genome-wide associations restricted to (**a**) individuals without antibiotic use in the past 6 months ($n = 14,171$), (**b**) individuals without inflammatory bowel disease ($n = 15,260$), or (**c**) unrelated participants ($n = 14,229$) compared to using data from all participants ($n = 16,017$). Related participants were identified based on kinship coefficients, and individuals were excluded until there were no pairs remaining with 3rd degree relatedness or closer. **d**, Sensitivity analysis of 149 genome-wide associations restricted to one participant from each household in SIMPLER and MOS ($n = 6,983$) compared to using data from all participants in those cohorts ($n = 7,284$). **e-i**, In the full dataset ($n = 16,017$), sensitivity analyses were also performed for (**e**) the 56 genome-wide linear regression associations with centered log-ratio (CLR) transformation before the rank-based inverse normal transformation compared to rank-based inverse normal transformation only, (**f**) the 93 genome-wide logistic regression associations using Firth correction compared to not using Firth correction, and (**g**) the 149 genome-wide associations without age[2] as a covariate compared to including it, (**h**) analyzing SCAPIS-Malmö and SCAPIS-Uppsala separately compared to models pooling them with site adjustment (original analysis), and (**i**) including vs. not including body mass index (BMI) as an additional covariate. **j-l**, Finally, sensitivity analyses were performed for the 149 genome-wide associations comparing the original model in SCAPIS ($n = 8,733$) with models including (**j**) alcohol intake ($n = 8,707$), (**k**) smoking behavior ($n = 8,452$), or (**l**) fiber consumption ($n = 8,624$) as an additional covariate. Smoking behavior (3% missing) was categorized into current smokers (12%), former smokers (35%), and never smokers (49%). Mean ± s.d. for fiber consumption and median (25th-75th percentile) for alcohol consumption in SCAPIS were 12.0 ± 4.2 g/day and 5.9 (2.0-10.6) g/day, respectively. The diagonal black line indicates where values of $y = x$, the red line a slope from linear regressions of beta coefficients from the sensitivity analysis and the original analysis, and in the upper left corner the Pearson correlation coefficient $r$ is shown.

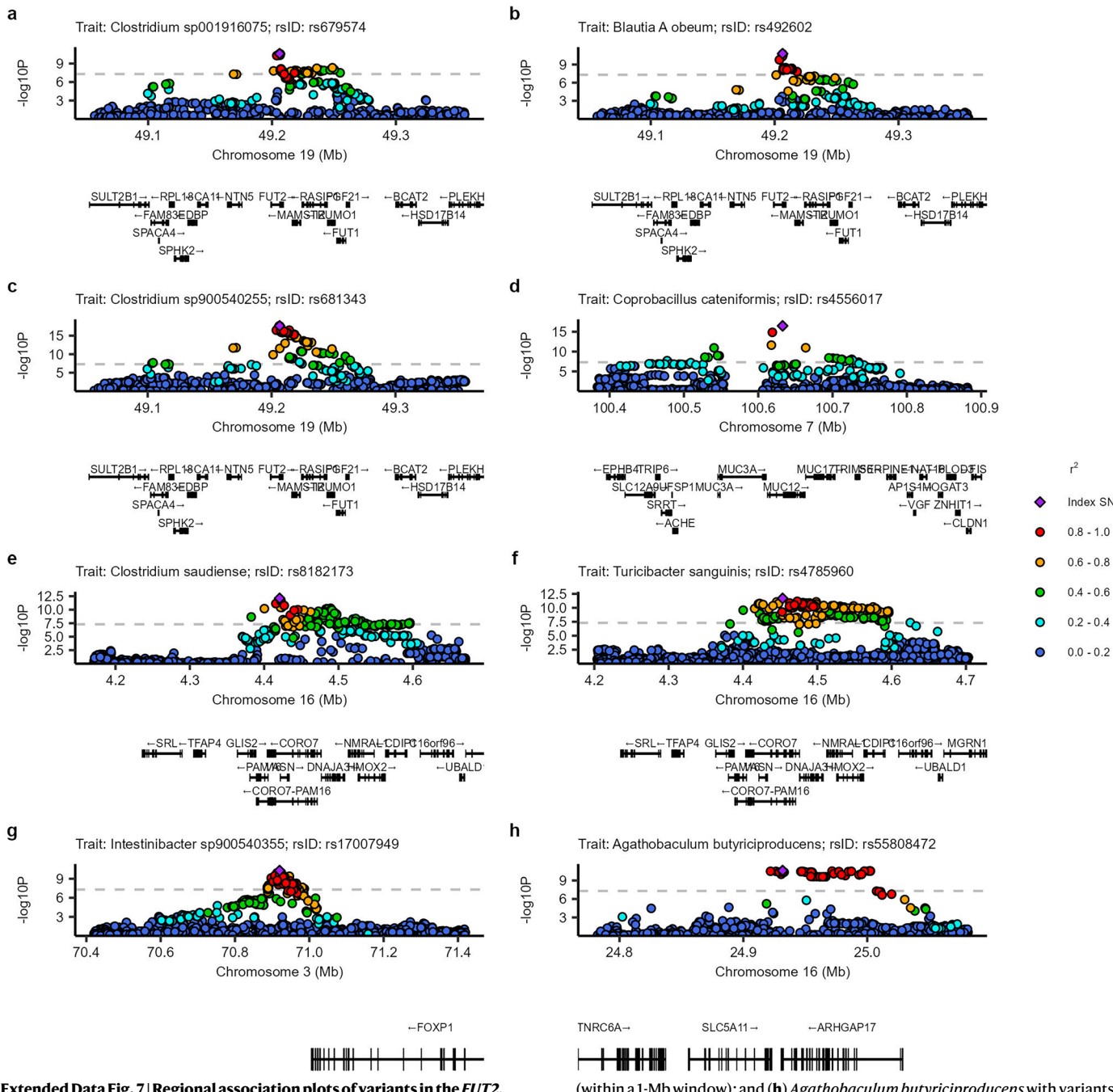

**Extended Data Fig. 7 | Regional association plots of variants in the *FUT2*, *MUC12*, *CORO7*–*HMOX2*, *FOXP1*, and *SLC5A11* gene loci. a-h**, Regional association plots of (**a**) *Clostridium* sp001916075, (**b**) *Blautia A obeum*, and (**c**) *Clostridium* sp900540255 with variants in the *FUT2* gene locus (within a 300-kb window); (**d**) *Coprobacillus cateniformis* with variants in the *MUC12* gene locus (within a 500-kb window); (**e**) *Clostridium saudiense*, and (**f**) *Turicibacter sanguinis* with variants in the *CORO7*-*HMOX2* gene locus (within a 500-kb window); (**g**) *Intestinibacter* sp900540355 with variants in the *FOXP1* gene locus

(within a 1-Mb window); and (**h**) *Agathobaculum butyriciproducens* with variants in the *SLC5A11* gene locus (within a 300-kb window). The lead variant is indicated as the purple diamond. Other SNPs are indicated by dots colored according to the linkage disequilibrium ($r^2$) values with the lead variant calculated using SCAPIS dosages. *P* values in **a**, **c**, **d**, **e**, **g**, and **h** were calculated using logistic and in **b** and **f** using linear regression (two-sided tests). The horizontal dashed gray line indicates genome-wide significance threshold ($-\log_{10}(5 \times 10^{-8})$).

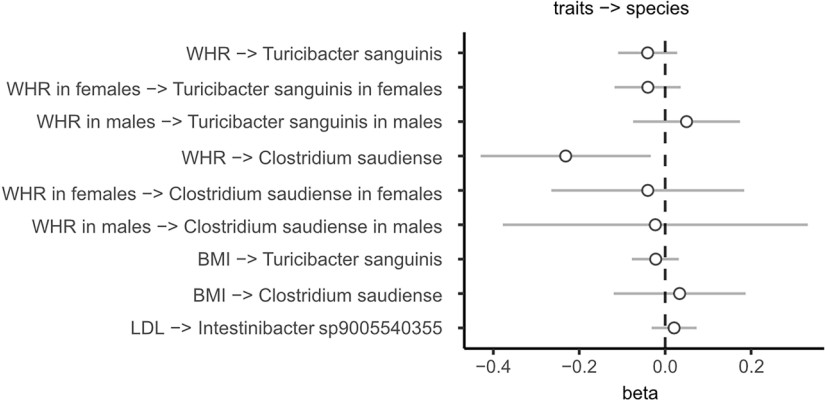

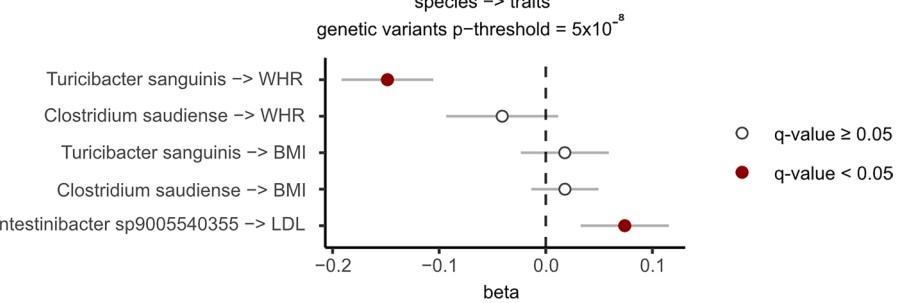

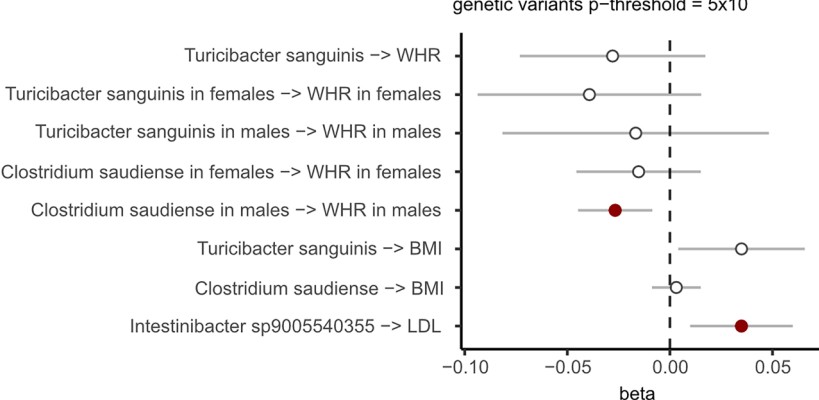

**Extended Data Fig. 8 | Bidirectional Mendelian randomization of gut species and metabolic traits.** Forest plot showing results from bidirectional inverse variance weighted Mendelian randomization (MR) analyses (two-sided tests) of specific gut species ($n = 16{,}017$) with adiposity traits ($n = 694{,}649$) and LDL ($n = 1.32$ M). Data are presented as inverse variance weighted regression betas representing genetically proxied increase in either species abundance or trait, with estimates passing false discovery rate threshold ($q < 0.05$) depicted as filled circles and those with $q \geq 0.05$ as empty circles. Gray lines show the 95% confidence intervals. Genetic variants $P$-threshold, variant inclusion threshold used to create the genetic instruments; LDL, low-density lipoprotein cholesterol; WHR, waist-hip ratio adjusted for body mass index (BMI).

# Reporting Summary

## Statistics

For all statistical analyses, confirm that the following items are present in the figure legend, table legend, main text, or Methods section.

| n/a | Confirmed | |
|---|---|---|
| ☐ | ☒ | The exact sample size (*n*) for each experimental group/condition, given as a discrete number and unit of measurement |
| ☐ | ☒ | A statement on whether measurements were taken from distinct samples or whether the same sample was measured repeatedly |
| ☐ | ☒ | The statistical test(s) used AND whether they are one- or two-sided *Only common tests should be described solely by name; describe more complex techniques in the Methods section.* |
| ☐ | ☒ | A description of all covariates tested |
| ☐ | ☒ | A description of any assumptions or corrections, such as tests of normality and adjustment for multiple comparisons |
| ☐ | ☒ | A full description of the statistical parameters including central tendency (e.g. means) or other basic estimates (e.g. regression coefficient) AND variation (e.g. standard deviation) or associated estimates of uncertainty (e.g. confidence intervals) |
| ☐ | ☒ | For null hypothesis testing, the test statistic (e.g. *F*, *t*, *r*) with confidence intervals, effect sizes, degrees of freedom and *P* value noted *Give P values as exact values whenever suitable.* |
| ☒ | ☐ | For Bayesian analysis, information on the choice of priors and Markov chain Monte Carlo settings |
| ☒ | ☐ | For hierarchical and complex designs, identification of the appropriate level for tests and full reporting of outcomes |
| ☐ | ☒ | Estimates of effect sizes (e.g. Cohen's *d*, Pearson's *r*), indicating how they were calculated |

*Our web collection on statistics for biologists contains articles on many of the points above.*

## Software and code

Policy information about availability of computer code

| Data collection | No software was used for data collection. |
|---|---|
| Data analysis | All statistical analyses used R version 4.3.1, unless stated otherwise.<br><br>1. Bioinformatic processing, calculation of relative abundances, and microbial taxonomic annotation were conducted at Cmbio using the CHAMP profiler based on the Human Microbiome Reference HMR05 catalog.<br>2. Calculation of alpha diversity measures using vegan package v2.5-7<br>3. For simulations to assess the type I error of logistic and linear models, identifying the species prevalence cut-off where a linear model becomes unreliable (R v.4.1.1).<br>4. Principal components were calculated in the unrelated samples set with PLINK 1.9.<br>5. For imputation of the SCAPIS and MOS genotype data to the HRC r1.1, we used the Sanger Imputation Service with the pipeline "Pre-phasing and imputation with EAGLE2+PBWT". For the imputation of the SIMPLER -V and SIMPLER-U genotype data  to the HRC r1.1 panel, the Michigan Imputation Server was used (EAGLE v.2.4 + minimac v4).<br>6. For genome-wide association analyses GWAS of microbiome composition (species and higher taxonomic levels) and function, metabolites, short-chain fatty acids, we used the software REGENIE v3.3, METAL v2011-03-25, PLINK v2.00-alpha-5-20230923, SumHer v6 and the meta v6.5-0 R package. SNPTEST v.2.5.6 was used for genome-wide association analyses of GLP-1.<br>7. For functional pathways mapping, we used FUMA v1.5.2.<br>8. For causal inferences analyses (Mendelian randomization), we used MendelianRandomization v0.9.0 R package (R v4.2.2).<br>9. For the stool metabolomics analyses coordinates were lifted over using Ensembl Variation 112 and https://genome.ucsc.edu/cgi-bin/hgLiftOver.<br>10. Colocalization was performed using TwoSampleMR v0.5.7 and coloc v5.2.2 R packages. |

11. Antigen - secretor status interactions were performed using the lmerTest v3.1-3 and metafor v4.4-0 R packages.
12. For the species - metabolite Spearman correlations we used the ppcor v1.1 R package.
13. For calculation of kinship estimator, we used KING as implemented in PLINK v.2.0.
14. For analyzing data from Sanger sequencing we used Sequencher v5.4.6
15. For functional annotation, EggNOG-mapper v.2.0.1 was used

Code related to the analyses in this study are available at https://github.com/MolEpicUU/GWAS_scripts and https://github.com/MolEpicUU/GWAS_microbiome and in Zenodo: https://doi.org/10.5281/zenodo.16947117 and https://doi.org/10.5281/zenodo.16925644

For manuscripts utilizing custom algorithms or software that are central to the research but not yet described in published literature, software must be made available to editors and reviewers. We strongly encourage code deposition in a community repository (e.g. GitHub). See the Nature Portfolio guidelines for submitting code & software for further information.

# Data

Policy information about availability of data

All manuscripts must include a data availability statement. This statement should provide the following information, where applicable:
- Accession codes, unique identifiers, or web links for publicly available datasets
- A description of any restrictions on data availability
- For clinical datasets or third party data, please ensure that the statement adheres to our policy

Complete GWAS summary statistics are available in the GWAS catalog with accession numbers GCST90670368 to GCST90671939. De-hosted anonymized metagenomic sequencing data from SCAPIS used in this study can be found at the European Nucleotide Archive under accession number PRJEB51353. Single-cell RNA-seq data is available in the GEO repository with accession number GSE284419 and GSE269778, and on Dryad (https://doi.org/10.5061/dryad.8pk0p2ns8). The metagenomics, metabolomics and genetic data supporting the conclusions of this article were provided by the SCAPIS, SIMPLER, and MOS central data offices, and are not shared publicly due to confidentiality and ethical restrictions. Data will be shared by the respective data offices only after permission from the Swedish Ethical Review Authority (https://etikprovningsmyndigheten.se) and from the respective boards (https://www.scapis.org/data-access, https://www.simpler4health.se, and https://www.malmo-kohorter.lu.se/malmo-offspring-study-mos).

# Human research participants

Policy information about studies involving human research participants and Sex and Gender in Research.

| Reporting on sex and gender | The genome-wide association study performed here utilized genetic information and fecal metagenomic data from both sexes from 16,017 adults of European ancestry from four Swedish cohorts. The sex balance in the dataset is SCAPIS (52.5% female), SIMPLER-Västmanland (37.7% female) and SIMPLER-Uppsala (100% female), MOS (47.1% female). |
| --- | --- |
| | SCAPIS and MOS: sex was obtained from the Swedish population register.<br>SIMPLER-Västmanland and SIMPLER-Uppsala: Invitation sent to all women for mammography screening (identified from the Swedish population register); men were identified from the Swedish population register. |
| | Sex-stratified analyses were conducted for associations of study-wide significant loci-species combinations, and Mendelian randomization analyses of specific gut microbial species with several adiposity traits and LDL cholesterol. |
| Population characteristics | SCAPIS<br>The Swedish CArdioPulmonary BioImage Study (SCAPIS) is a multi-center cohort comprising 30,154 individuals aged 50-65. For this analysis, 8,733 participants of European ancestry from the Malmö and Uppsala sites with both gut microbiome and genotype data were included. At baseline, participants provided blood samples during the first visit and were asked to collect fecal samples at home, storing them at -20°C until samples were brought to the study center for the second visit. DNA extracted from whole blood was used for genotyping. Birth year and sex were obtained from the Swedish population register.<br><br>SIMPLER-Västmanland and SIMPLER-Uppsala<br>The Swedish Infrastructure for Medical Population-Based Life-Course and Environmental Research (SIMPLER; https://www.simpler4health.se/w/sh/en) includes data from two large, ongoing population-based studies: the Cohort of Swedish Men (COSM) and the Swedish Mammography Cohort (SMC).58 The COSM initially enrolled 48,850 men born between 1918 and 1952 living in Västmanland and Örebro counties in 1997. The SMC enrolled 66,651 women by sending invitations to all women born between 1914 and 1948 living in Uppsala and Västmanland counties between 1987 and 1990. The current analysis is based on a randomly selected subsample from these studies who were invited for clinical examination with genotype and gut microbiome data: SIMPLER-Västmanland (SIMPLER-V) and SIMPLER-Uppsala (SIMPLER-U). SIMPLER-V includes 4,515 COSM and SMC participants from Västmanland examined between 2010 and 2019. SIMPLER-U includes 981 women from the county of Uppsala, examined between 2003 and 2009 (no stool collected) and re-examined between 2015 and 2019 (stool collected).<br><br>MOS<br>The Malmö Offspring Study (MOS) includes participants aged ≥18 who are children or grandchildren of participants from the Malmö Diet and Cancer Study (MDC)-Cardiovascular Cohort, a subset of the larger MDC.59 Data collection in MOS began in 2013 and included 4,721 participants by 2020. The current study included 1,788 participants with genotype and gut microbiome data who attended baseline measurements between 2013 and 2017. |
| Recruitment | SCAPIS: 30,154 participants aged 50-64 years invited from a random selection from the Swedish population register in areas adjacent to study sites. |

SIMPLER-Västmanland and SIMPLER-Uppsala: (SMC) From March 1987 to December 1990, all women living in Uppsala County of central Sweden and who were born in 1914 through 1948 (n = 48,517) and all women living in the adjacent Västmanland County (n = 41,786) who were born in 1917 through 1948 received an invitation by mail to participate in a population-based mammography screening program, along with a questionnaire. Returning of the questionnaire was their informed consent. The SMC population is comparable to the general Swedish population with regards to age distribution, education level and body mass index (BMI). (COSM) In the fall of 1997, all men born in 1918 through 1952 living in Västmanland and Örebro counties in central Sweden (n = 100,303) received an invitation to participate in the study, along with a self-administered questionnaire. The COSM population is comparable to the general Swedish population with regards to age distribution, education level and BMI.

MOS: Participants were children and grandchildren of index individuals in Malmö Diet and Cancer Study—Cardiovascular Cohort, which was a random, subpopulation of the Malmö Diet and Cancer Study. The participants were 18 years or older and living in Malmö or the nearby catchment area.

| Ethics oversight | The current association study has been approved by the Swedish Ethical Review Authority (DNR 2022-06137-01 and DNR 2024-01992-02). All participants in the respective cohorts below have provided written informed consent to participate in the studies and have their samples and data collected, stored, and processed. The Swedish Ethical Review Board has approved the data collection, and the approval numbers are provided: SCAPIS (DNR 2010-228-31M), SIMPLER (DNR 2009/2066-32, DNR 2009/1935-32, DNR 2010/0148-32, DNR 2014/892-31/3), MDC (DNR 532/2006, DNR 51-90), and MOS (DNR 2012-594). The PPP-Botnia study received approval from the Ethics Committee of Helsinki University (approval number 608/2003). The HUNT study was approved by the local ethical review board (Regionale kommitter for medicinsk og helsefaglig forskningsetik Midt-Norge; REK-656785). |
|---|---|

Note that full information on the approval of the study protocol must also be provided in the manuscript.

# Field-specific reporting

Please select the one below that is the best fit for your research. If you are not sure, read the appropriate sections before making your selection.

☒ Life sciences ☐ Behavioural & social sciences ☐ Ecological, evolutionary & environmental sciences

For a reference copy of the document with all sections, see nature.com/documents/nr-reporting-summary-flat.pdf

# Life sciences study design

All studies must disclose on these points even when the disclosure is negative.

| Sample size | The sample size was based on the number of participants in the respective cohorts who have both high-quality data of gut microbiome and genotype data, resulting in 16,017 participants across 4 Swedish cohorts. The need for larger size for a microbiome GWAS is a recognized issue in the field (Sanna et al, Nat Genet 2022, 54:100-106). However it is challenging to combine data from multiple cohorts to increase power due to biological and technical variation, including in metagenomic data processing, among cohorts. This study is the largest multi-cohort analysis with microbiome data processed with a harmonized bioinformatics pipeline in each of the cohorts. The sample size of each cohort included is outlined below.<br>SCAPIS: For this analysis, 8,733 participants of European ancestry from the Malmö and Uppsala sites data were included.<br>SIMPLER-Västmanland and SIMPLER-Uppsala: SIMPLER-V includes 4,515 COSM and SMC participants from Västmanland examined between 2010 and 2019. SIMPLER-U includes 981 women from the county of Uppsala, examined between 2003 and 2009 (no stool collected) and re-examined between 2015 and 2019 (stool collected).<br>MOS: The current analysis included 1,788 participants. |
|---|---|
| Data exclusions | For genotyping data, samples from individuals of non-European ancestry, failure in sex check, excess heterozygosity, and other quality control criteria including Hardy-Weinberg equilibrium, and minor allele frequency or count, were excluded. For gut microbiome data, only data that passed quality control was included.<br><br>Sensitivity analysis excluding individuals with antibiotic use in the past 6 months or self-reported inflammatory bowel disease;<br>exclusion of individuals who used antibiotics in the last six months or self-reported inflammatory bowel disease did not impact the genome-wide significant associations;<br>exclusion of persons in the same household resulting in only one person per household from SIMPLER and MOS did not impact results from SIMPLER and MOS cohorts;<br>excluding related participants resulting in only one participant from each related pair (meaning, no more related participant up to 3rd degree) did not affect the genome-wide findings;<br>MOS: Participants who were also part of the SCAPIS cohort were excluded from the MOS data. |
| Replication | Replication was conducted in the large Norwegian HUNT cohort of 12,652 individuals. We also used published summary statistics from two previous studies in FINRISK (n=5,959) and Dutch Microbiome Project (n=7,738) to validate the present findings. Best matching species were identified and our results were consistent with all 7 available associations in FINRISK, and 2 out of 4 in the Dutch cohort. The study in FINRISK used an earlier GTDB version (R89) for taxonomic annotations compared to our study (R214) while the Dutch study annotated their taxa by using MetaPhlAn2, which uses NCBI nomenclature. |
| Randomization | This is a population-cohort study and not an intervention study. Thus randomization is not applicable. |
| Blinding | This is a population-cohort study and not an intervention study. Thus blinding is not applicable. |

# Reporting for specific materials, systems and methods

We require information from authors about some types of materials, experimental systems and methods used in many studies. Here, indicate whether each material, system or method listed is relevant to your study. If you are not sure if a list item applies to your research, read the appropriate section before selecting a response.

## Materials & experimental systems

| n/a | Involved in the study |
|-----|------------------------|
| ☒ | Antibodies |
| ☒ | Eukaryotic cell lines |
| ☒ | Palaeontology and archaeology |
| ☒ | Animals and other organisms |
| ☒ | Clinical data |
| ☒ | Dual use research of concern |

## Methods

| n/a | Involved in the study |
|-----|------------------------|
| ☒ | ChIP-seq |
| ☒ | Flow cytometry |
| ☒ | MRI-based neuroimaging |

