## [Peer Review File · Nature Genetics]

GENOME-WIDE ASSOCIATION ANALYSES HIGHLIGHT THE ROLE OF THE INTESTINAL MOLECULAR ENVIRONMENT IN HUMAN GUT MICROBIOTA VARIATION

Corresponding Author: Professor Tove Fall

Version 0:

Decision Letter:

7th November 2024

Dear Professor Fall,

Your Article "Genome-wide association study highlights the role of the intestinal molecular environment in human gut microbiota variation" has been seen by two referees. You will see from their comments below that, while they find your work of potential interest, they have raised substantial concerns that must be addressed. In light of these comments, we cannot accept the manuscript for publication at this time, but we would be interested in considering a suitably revised version that addresses the referees' concerns.

We hope you will find the referees' comments useful as you decide how to proceed. If you wish to submit a substantially revised manuscript, please bear in mind that we will be reluctant to approach the referees again in the absence of major revisions.

To guide the scope of the revisions, the editors discuss the referee reports in detail within the team with a view to identifying key priorities that should be addressed in revision, and sometimes overruling referee requests that are deemed beyond the scope of the current study. In this case, we ask that you thoroughly address all technical points related to the association analyses and their interpretation, performing additional analyses (including further adjustments for potential confounders such as cohabitation and kinship) and genotyping as requested, and revise the presentation where needed. We hope you will find this prioritized set of referee points to be useful when revising your study. Please do not hesitate to get in touch if you would like to discuss these issues further.

If you choose to revise your manuscript taking into account all reviewer and editor comments, please highlight all changes in the manuscript text file. At this stage, we will need you to upload a copy of the manuscript in MS Word .docx or similar editable format.

*2) If you have not done so already, please begin to revise your manuscript so that it conforms to our Article format instructions, available here. Refer also to any guidelines provided in this letter.

*3) Include a revised version of any required Reporting Summary: <https://www.nature.com/documents/nr-reporting-summary.pdf>

Please be aware of our [guidelines](https://www.nature.com/nature-research/editorial-policies/image-integrity) on digital image standards.

Link Redacted

If you wish to submit a suitably revised manuscript, we hope to receive it within 3-6 months. If you cannot send it within this time, please let us know. We will be happy to consider your revision so long as nothing similar has been accepted for publication at Nature Genetics or published elsewhere. Should your manuscript be substantially delayed without notifying us in advance and your article is eventually published, the received date would be that of the revised, not the original, version.

Nature Genetics is committed to improving transparency in authorship. As part of our efforts in this direction, we are now requesting that all authors identified as 'corresponding author' on published papers create and link their Open Researcher and Contributor Identifier (ORCID) with their account on the Manuscript Tracking System (MTS), prior to acceptance. ORCID helps the scientific community achieve unambiguous attribution of all scholarly contributions. You can create and link your ORCID from the home page of the MTS by clicking on 'Modify my Springer Nature account'. For more information, please visit www.springernature.com/orcid.

Thank you for the opportunity to review your work.

Sincerely,
Kyle

Kyle Vogan, PhD
Senior Editor
Nature Genetics
<https://orcid.org/0000-0001-9565-9665>

Referee expertise:

Referee #1: Genetics, metagenomics, inflammatory bowel diseases

Referee #2: Genetics, metagenomics, bioinformatics

Reviewers' Comments:

Reviewer #1 (Remarks to the Author):

The mQTL study by Dekkers and colleagues studied 16,040 Swedish paired gut metagenome-SNP array data sets (from four independent studies) and performed a validation in the Norwegian HUNT data set (n=12,652 individuals). Fifteen study-wide significant signals from eight distinct loci were identified, of which 11 associations were replicated including the known ABO and LCT associations. Novel associations comprised MUC12, CORO7-HMOX2, SLC5A11, FUT2, FOXP1, and FUT3-FUT6 and were corroborated by additional omics data. The mentioned loci genetically link gut microbial variation to enteroendocrine chemosensing and the molecular composition of the mucosal layer. OR51E1/E2 was associated with richness, number of species detected.

Strengths

The authors use high-resolution metagenomics instead of 16S rRNA amplicon data and a reasonable sized discovery study of one ethnicity. However, they are not the first to do so in an mQTL study (at least four previous studies as mentioned in introduction). One of the first metagenome-based mQTL studies was performed by the Groningen group and comprised 7,738 individuals.

The section on the validation of the ABO and FUT2 associations is strong and the FUT3-FUT6 association is intriguing, further highlighting the importance of fucosylated glycans in host-microbe interactions.

Major points

1. How did the authors identify confounders/covariates in their data? Why did they choose to use the listed ones? Why did they add age2, too? I miss these details on page 7, top paragraph.
2. Why did the authors not attempt a validation of their signals in the other metagenome mQTL data sets, e.g. from Finland or the Netherlands (Groningen group)? These data sets have been published and should be readily available, at least upon request. I strongly recommend to further replicate their signals in at least one of these cohorts and to check what may be "Nordic" association or if all are consistent.
3. The OR51E1/2 association is intriguing, and the authors attempt to further support this association through genome-wide significant SNP-species associations and (single cell) gene expression data. However, polymorphism at olfactory genes is generally high and these variants are often excluded from whole exome sequencing studies as they are considered false positives and likely the result of erroneous genotype calling. I strongly recommend to genotype rs10836441 in a larger subset of samples (if not all) using e.g. TaqMan technology to validate that the genotypes are 100% concordant. Has this SNP been imputed or is on the used arrays? I cannot find the regional association plot for this locus in the Supplement. Is there multi-SNP support for this signal?
4. The authors included individuals with IBD and antibiotics in their discovery analysis. While they show that the main hits are not influenced when excluding these individuals, I strongly recommend running the discovery analysis without these individuals. It is good practice in microbiome studies to exclude individuals that took antibiotics up front (at least 6 weeks before sampling, 6 months more conservative).
5. MUC12 association: Very intriguing signal! I would like to see though again that actually genotyped SNPs (with good clustering and quality) support the signal. The regional association plot in SFigure 1g shows a gap upstream of the main peak. What is the reason here? Can the authors genotype a subset of samples to confirm the genotype concordance (see also #2 above)? MUCs are normally excluded in WES studies, similarly as ORs.
6. Please better highlight hits in Table 1 that are replicated robustly (after Bonferroni-correction), e.g. by bold type.
7. I suggest moving Figure 3 to the Supplement, it shows expression of the candidate genes in the intestinal tract and is from a previously published data set. If the authors want to highlight OR51E1/2 expression, they should also do this in the Figure. What about immune cells that are also present in the gut and that interact with the microbial components?

Minor points

1. Line numbers would have helped to point to specific sections.
2. The authors should properly reference and acknowledge the first (!) descriptions of the LCT (probably Blekhman, R. et al. Host genetic variation impacts microbiome composition across human body sites. *Genome Biol.* 16, 191 (2015).; not listed among references) and ABO (in their paper #32, however, already mentioned in the beginning) associations.
3. Can the authors comment on the known MED13L association in their data? Do they replicate the signal?
4. Table 1: are the allele frequencies for all candidate variants comparable for the 4 individual Swedish studies, and also vs. HUNT? Or any significant deviations?
5. Figure 1a: please add age distribution of replication cohort HUNT in grey. And same for BMI. I also suggest adding a dashed line for the age distribution of the overall discovery cohort.
6. Figure 2a: please better highlight the novel loci here.
7. Figure 2b/c: make clear that one dot corresponds to one SNP.
8. Please provide references for the used tools, e.g. REGENIE, in Methods section.

Reviewer #2 (Remarks to the Author):

This is a largest to date metagenomics-based mbGWAS. It is performed in more or less standard way. However, I have some general comments that might make the study a bit better.

1. The signal for richness is borderline significant. I see no reason to skip microbiome-wise multiple testing for this specific association: it's still a part of mbGWAS, and the trait itself is still more or less independent from the species abundance. And with a study-wise cutoff it's not significant ($1.9e-9$). Having a replication is nice, but even meta-analysis p-value from discovery/replication cohorts only gives $P \sim 3e-10$, which is still below study-wise cutoff. It does not necessarily mean that this result should be removed from the paper, but some claims should be calmed down, like the beginning of discussion: "For the first time, we could identify and replicate a human genetic variant associated with gut 380 microbiome richness, the OR51E1/E2 locus". It's a suggestive locus at best.
2. I'm surprised not to see any kind of composition-aware transformation. Instead of 'validation' with non-transformed scale, it

seems logical to make a round of validation with CLR instead. Compositionality might be a problem for any kind of MWAS, and mbGWAS in no exception

3. The choice of using specific toolkit for microbiome data processing (CHAMP). While showing a theoretical performance comparable or higher than other toolkits, there is one problem which creates problems for further use: the taxonomy. It seems that the pipeline uses GTDB taxonomy, even more: it's outdated version. For example, *Clostridium* sp001916075 (associated to FUT2) in the last release is already called *Clostridium lentium* (https://gtdb.ecogenomic.org/genome?gid=GCF_014287815.1). I would recommend not to use taxonomy that is very tricky to find and that requires reader to go deep into the history of releases of databases. Ideally, in the text it's better to use NCBI taxonomy and give GTDB names somewhere in supplementary material (if possible). I absolutely agree that there is a huge mess with multiple taxonomies at the moment, yet still it is appreciated to make readers' life easier.

4. Why didn't authors perform GWAS on higher taxonomic levels? It would make a lot of sense to do it on at least genera level, as the members of some genera carry a lot of similarities in their metabolic potential. I see no excuse for skipping it at least for genera and families (but also worth a try for other levels as well). The only possible reason why authors didn't do it is due to more strict necessity for multiple testing correction, but it's not a holy grail: I would even stick the cutoff to $5e-8 * N_{species}$, even if more taxonomic levels are involved (at the end of the day, from composition point of view, higher level taxa are just arithmetic sums of their species members).

5. Figure 3 is quite uninformative. By itself it doesn't bring a clear message. It could be better to rethink it: maybe to add more panels to point to exact parts of it that are referred to in the text.

MR results could be supported with the figures. Also, its description in the text is unclear: what were the instrument p-value cutoffs for exposures/outcomes? How was reverse causality controlled for? Were the results adjusted for multiple testing? The latter has to be done for sure.

6. FUT2 is not novel.

7. MUC genes (MUC22 in particular) were also previously picked up in mbGWAS. Worth to mention in the discussion.

8. Why there's no functional mbGWAS performed? KEGG or metacyc

9. MR part is rather weak. No reverse causality tests, no multiple testing adjustment for exposure-outcome pairs tested. It might be good idea to decrease the inclusion cutoff to get more instruments as a validation.

10. Last but not least, the adjustment for covariates. There are multiple problems here that might potentially affect the results: (1) some of the cohorts like SCAPIS and SIMPLER are actually multi-centre cohorts, thus the SNPs might proxy differences in LD between counties/cities, rather than be causal. This should be taken into account, and a few genetic components might not be enough. (2) most clear in SCAPIS but also can affect other: the total population of Malmo and Uppsala is 533 thousand people, and many of those are not from European ancestry. With only selecting 50-65 year-old people, this gives extremely small general population to sample from. It is very, very likely that many of those people are relatives, even close relatives, or cohabitants. Cohabitation is an extremely important source of microbiome variation and has to be taken into account.

So, to conclude, both adjustment for cohabitation (if possible) and more robust adjustment for genetic kinship should be used in the study, as both of those problems likely to occur in the study cohorts and might significantly affect the results.

Version 1:

Decision Letter:

13th February 2025

Dear Professor Fall,

Your revised Article "Genome-wide association study highlights the role of the intestinal molecular environment in human gut microbiota variation" has been seen by the original referees. You will see from their comments below that, while they consider the study improved, Reviewer #1 had a few ongoing technical concerns. We remain interested in the possibility of publishing your study in Nature Genetics, but we would like to consider your response to these ongoing concerns in the form of a further revision before we make a final decision on publication.

We therefore invite you to further revise your manuscript taking into account these comments. Please highlight all changes in the manuscript text file. At this stage, we will need you to upload a copy of the manuscript in MS Word .docx or similar editable format.

*2) If you have not done so already, please begin to revise your manuscript so that it conforms to our Article format instructions, available

[here](http://www.nature.com/ng/authors/article_types/index.html).

*3) Include a revised version of any required Reporting Summary: <https://www.nature.com/documents/nr-reporting-summary.pdf>

EXTENDED DATA FIGURES

Link Redacted

We hope to receive your revised manuscript within 4-8 weeks. If you cannot send it within this time, please let us know.

Nature Genetics is committed to improving transparency in authorship. As part of our efforts in this direction, we are now requesting that all authors identified as 'corresponding author' on published papers create and link their Open Researcher and Contributor Identifier (ORCID) with their account on the Manuscript Tracking System (MTS), prior to acceptance. ORCID helps the scientific community achieve unambiguous attribution of all scholarly contributions. You can create and link your ORCID from the home page of the MTS by clicking on 'Modify my Springer Nature account'. For more information, please visit www.springernature.com/orcid.

Sincerely,
Kyle

Kyle Vogan, PhD
Senior Editor
Nature Genetics
<https://orcid.org/0000-0001-9565-9665>

Referee expertise:

Referee #1: Genetics, metagenomics, inflammatory bowel diseases

Referee #2: Genetics, metagenomics, bioinformatics

Reviewers' Comments:

Reviewer #1 (Remarks to the Author):

The authors have done a very good job in addressing my comments. Extra efforts were put into validating some of their lead SNPs with an independent genotyping technology (here Sanger resequencing). Several of my concerns have diminished, only few questions remain:

(1) Can the authors mention the genotype concordance in % rather than showing the plots? Is the concordance close to 100%? Supplementary Figure 7 (MUC12 Locus) does not look convincing!

(2) Along the same line, Response letter figure 2a highlights 3 genotyped SNPs. However, these genotyped SNP are neither passing study-wide significance nor are they in high LD with the peak SNPs. This is concerning. In addition, the authors have not tested known variants in the gap and just write "Assemblies based on long-read sequencing such as the T2T-CHM13, are now filling these gaps." While I understand that the authors worked only with the array content and imputed genotypes thereof, they should have meanwhile genotyped SNPs in the gap region (given that informative and testable variants exist). This would help to understand the signal and LD structure better. Maybe the authors will also get a stronger signal. It is not satisfactory to have this gap in the data while it could potentially have been closed.

(3) Co-variables: For the genetic data, I am happy with the reply, but I was more thinking of other factors that are known to shape the microbiome: BMI, alcohol consumption, smoking, ... and also stool composition is strongly associated to gut microbiome composition. Can the authors exclude that accounting for these factors won't change the association results of the main hits?

Reviewer #2 (Remarks to the Author):

The authors have thoroughly taken into account the concerns from me and another reviewer. My concerns were mostly about:

The complex genetic architecture of studied populations. It seems that the adjustment for that was done properly.

GWAS on higher taxonomic ranks and functional pathways. This is now also done.

Mess in taxonomic names and taxonomic databases. This is now better.

Weak MR part, especially FDR and test for horizontal pleiotropy. Now is also done.

Overall, the authors did a good job in addressing my concerns. This paper is a great addition to a field of microbiome GWAS overall, and I was very glad for having the opportunity to review it.

Version 2:

Decision Letter:

Our ref: NG-A66450R1

18th June 2025

Dear Dr. Fall,

Thank you for submitting your revised manuscript "GENOME-WIDE ASSOCIATION STUDY HIGHLIGHTS THE ROLE OF THE INTESTINAL MOLECULAR ENVIRONMENT IN HUMAN GUT MICROBIOTA VARIATION" (NG-A66450R1). In light of the revisions implemented in response to Reviewer #1, we will be happy in principle to publish your study in Nature Genetics as an Article pending final revisions to comply with our editorial and formatting guidelines.

We are now performing detailed checks on your paper, and we will send you a checklist detailing our editorial and formatting requirements soon. Please do not upload the final materials or make any revisions until you receive this additional information from us.

Thank you again for your interest in Nature Genetics. Please do not hesitate to contact me if you have any questions.

Sincerely,
Kyle

Kyle Vogan, PhD
Senior Editor
Nature Genetics
<https://orcid.org/0000-0001-9565-9665>

Referee expertise:

Referee #1: Genetics, metagenomics, inflammatory bowel diseases

Referee #2: Genetics, metagenomics, bioinformatics

Reviewers' Comments:

Reviewer #1 (Remarks to the Author):

The mQTL study by Dekkers and colleagues studied 16,040 Swedish paired gut metagenome-SNP array data sets (from four independent studies) and performed a validation in the Norwegian HUNT data set (n=12,652 individuals). Fifteen study-wide significant signals from eight distinct loci were identified, of which 11 associations were replicated including the known ABO and LCT associations. Novel associations comprised MUC12, CORO7-HMOX2, SLC5A11, FUT2, FOXP1, and FUT3-FUT6 and were corroborated by additional omics data. The mentioned loci genetically link gut microbial variation to enteroendocrine chemosensing and the molecular composition of the mucosal layer. OR51E1/E2 was associated with richness, number of species detected.

Strengths

The authors use high-resolution metagenomics instead of 16S rRNA amplicon data and a reasonable sized discovery study of one ethnicity. However, they are not the first to do so in an mQTL study (at least four previous studies as mentioned in introduction). One of the first metagenome-based mQTL studies was performed by the Groningen group and comprised 7,738 individuals.

The section on the validation of the ABO and FUT2 associations is strong and the FUT3-FUT6 association is intriguing, further highlighting the importance of fucosylated glycans in host-microbe interactions.

Major points

1. How did the authors identify confounders/covariates in their data? Why did they choose to use the listed ones? Why did they add age², too? I miss these details on page 7, top paragraph.

:

Thank you for this question. Genetic confounding may cause spurious results if there are subpopulations in the data with different allele frequencies and microbiome compositions. Our primary way to mitigate this issue was restrict the study population to those of European ancestry. Additionally, we used the REGENIE software designed to account for population structure and relatedness (Mbatchou et al., *Nat Genet* 2021) Based on recommendations from REGENIE authors, we also included PC1-PC10 in the model to further account for any remaining bias (<https://github.com/rcggithub/regenie/issues/61>). Aside from genetic confounding, we do not see any other possible confounders.

To increase precision and power, we added age, sex, and technical covariates (metagenomics sequencing plate), as these factors can affect the microbiome composition. Specifically, we included age² to account for previously observed non-linear age-microbiome relationships (Odamaki et al., 2016). Additionally, the inclusion of age² slightly increased the model fit in linear regression analyses performed before the GWAS, where the average adjusted R² was 0.000417 for the “age only” model and 0.000444 for the “age + age²” model. We have now added a sensitivity analysis

excluding age² for the main associations showing nearly identical results (See new Supplementary Figure 5g below).

Site/cohort was not included as a separate covariate because cohorts were analyzed separately, and all cohorts included only one site except SCAPIS. SCAPIS included two sites, which contained non-overlapping plates, meaning that the plate adjustment covered the site adjustment. We have now performed a sensitivity analysis where we analyzed SCAPIS-Uppsala and SCAPIS-Malmö separately. After meta-analysis, the main associations were nearly identical (See new Supplementary Figure 5h below).

Revised text:

Results, page 7

Based on simulations aiming to maximize power and minimize false positive findings, we used logistic regression for the presence of 679 species detected in 50% or fewer of the samples, testing 5,368,906 variants with minor allele frequency (MAF) $\geq 5\%$. For 242 species with a prevalence $>50\%$ in all four studies, we used linear regression for the relative abundance of those species, testing 7,454,886 variants with MAF $\geq 1\%$. **GWAS was performed separately for each microbial phenotype using REGENIE v3.3 in each cohort separately with sex, age, age², metagenomics sequencing plate, and genetic principal components 1-10 included as covariates and was followed by inverse-variance weighted fixed-effects meta-analysis.** We replicated associations study-wide associations of host genetic variants with species and species diversity in HUNT ($n=12,652$), including stool metagenomic data profiled with the same pipeline employed in the Swedish studies.

Results, page 10

The genome-wide significant associations were similar in sensitivity analysis including models using centered log-ratio transformation for species analyzed using linear regression and Firth correction for species analyzed using logistic regression, in models not including age² as a covariate, in models where SCAPIS-Malmö and SCAPIS-Uppsala were analyzed separately, after excluding all but one individual per household and after excluding all but one individual from each related pair, or when individuals who used antibiotics in the last six months, or who self-reported inflammatory bowel disease (IBD) were excluded (Supplementary Fig. 5 and 6).

Methods, page 32-33

GWAS was performed separately for microbial alpha diversity and 921 species using REGENIE v3.3 for each cohort (**SCAPIS, SIMPLER-V, SIMPLER-U, MOS**). **A subset of the genotype** datasets was created for the first REGENIE step to fit whole genome regression models including only quality-controlled directly genotyped SNPs with MAF $>1\%$ and HWE $P < 1 \times 10^{-15}$. For the second step all variants with an information score >0.7 was included in association analyses performed using logistic regression for binary variables and genetic variants with MAF $>5\%$, and linear regression for RIN-transformed variables and genetic variants with MAF $>1\%$. Covariates were sex, age, age², metagenomics sequencing plate, and genetic principal components 1-10. **Metagenomics sequencing plate, age, and sex were included to increase precision and power. For SCAPIS, the site was accounted for by the plate variable because plate was nested into the site variable, while all other cohorts were single-site. Based on previous non-linear associations between age and microbiome⁸⁶ and our results from a naive linear model for**

the association between age and microbial species, we opted to include age also as age^2 . REGENIE accounts for population stratification, and to further account for any residual bias, we additionally included PC1-PC10 in the model.⁸⁷ Cohort-specific results were meta-analyzed using the inverse-variance weighted **fixed-effects** method in METAL v2011-03-25.

Supplementary Figure 5g. Sensitivity analysis of 149 genome-wide associations in the full dataset (n=16,017) excluding age^2 compared to including age^2 as a covariate.

Supplementary Figure 5h. Sensitivity analysis of 149 genome-wide associations in the full dataset (n=16,017) where SCAPIS-Malmö and SCAPIS-Uppsala were analyzed separately compared to models adjusted for the SCAPIS sites (original analysis).

2. Why did the authors not attempt a validation of their signals in the other metagenome mQTL data sets, e.g. from Finland or the Netherlands (Groningen group)? These data sets

have been published and should be readily available, at least upon request. I strongly recommend to further replicate their signals in at least one of these cohorts and to check what may be “Nordic” association or if all are consistent

Authors’ reply:

Thank you for this suggestion. We have now added data from two additional studies from FINRISK (Qin et al., 2022) and the Dutch Microbiome Project (Lopera-Maya et al., 2022). The study in FINRISK used an earlier GTDB version (R89) than our study (R214). The study in the Dutch Microbiome Project used NCBI nomenclature. We identified the best available matching species using Taxon History in the GTDB database (<https://gtdb.ecogenomic.org/taxon-history>). In summary, we could match species for 7/15 of our study-wide associations with the FINRISK study and 4/15 of our associations with the Dutch Microbiome Project study. Our results showed consistent results using a Bonferroni-corrected p-value for 7/7 associations in FINRISK and 2/4 associations in the Dutch cohorts. We believe these results strengthen our current findings and have included these in Supplementary Table 9.

Revised text:

Methods, page 33-34

We assessed all associations passing the study-wide threshold in HUNT applying the same models as in the Swedish cohorts using REGENIE with the same model specifications. We further assessed the validity of our findings using summary statistics from the previously published FINRISK¹⁴ and Dutch Microbiome Project¹² studies. The FINRISK study included species with >25% prevalence, applied centered log-ratio transformation to relative abundances and analyzed genetic associations in a linear mixed model (BOLT-LMM v.2.3.2). The Dutch Microbiome Project study focused on taxa with mean relative abundance >0.001% across all samples and present in $\geq 1,000$ of the 7,738 individuals (~13% of population), treating all zero values as missing data, log-transformed relative abundances followed by a RIN transformation and analyzed genetic associations using a linear mixed model implemented in SAIGE v.0.38. We identified the best matching species to our study-wide associated species using the Taxon History in the GTDB database (<https://gtdb.ecogenomic.org/taxon-history>). The study in FINRISK used an earlier GTDB version (R89) for taxonomic annotations as compared to our study (R214), while the Dutch study annotated their taxa by using MetaPhlan2, which uses NCBI nomenclature. Public GWAS results for the matched species were accessed, and beta, standard error, p-value, and species prevalence were extracted for the relevant SNPs. If the SNPs were unavailable, we used proxies with $r^2 > 0.8$ and aligned the alleles.

Results, page 10

From the 15 SNP-species associations, we replicated 11 associations at six loci in HUNT at a Bonferroni-corrected threshold ($P < 3 \times 10^{-3}$) with concordant direction and all 15 at $P < 0.05$ with concordant direction. **Of these 15 SNP-species associations, 7 could be assessed in FINRISK¹⁴ and 4 in the Dutch Microbiome Project¹² (Supplementary Table 9). Allele frequencies for all study-wide significant variants were comparable across studies, except for the LCT SNPs in FINRISK (Supplementary Table 10). LCT variants are known to vary across countries.³⁰ Altogether, we found support for replication in at least one external cohort for 13 associations at 6 loci.** These 6 loci included the well-known *ABO* and *LCT* loci.^{8,15} Lactase non-persistence alleles of variants in the *LCT* locus were associated

with ~~in~~decreased levels of *Bifidobacterium adolescentis* while ~~de~~increased levels of *Phoceamassiliensis*, *Negativibacillus* sp000435195, and *Copromonas* sp000435795 and at genome-wide level with five additional species including three *Bifidobacterium* species.

Discussion, page 17

All associations were consistent in the Norwegian HUNT study in terms of directionality and reached at least nominal significance supporting the robustness.

We further found strong support for all 7 available associations in FINRISK and 2 associations in the Dutch Microbiome Project. Our study only included participants of European ancestry, **mainly** from Nordic countries, which limits generalizability to other ethnic groups and regions.

3. The OR51E1/2 association is intriguing, and the authors attempt to further support this association through genome-wide significant SNP-species associations and (single cell) gene expression data. However, polymorphism at olfactory genes is generally high and these variants are often excluded from whole exome sequencing studies as they are considered false positives and likely the result of erroneous genotype calling. I strongly recommend to genotype rs10836441 in a larger subset of samples (if not all) using e.g. TaqMan technology to validate that the genotypes are 100% concordant. Has this SNP been imputed or is on the used arrays? I cannot find the regional association plot for this locus in the Supplement. Is there multi-SNP support for this signal?

Authors' reply:

Thank you for this question. The variant (rs10836441) was imputed with a high imputation info score of 0.996. We went back to the raw data and extracted the signal intensity cluster plots with the software GenomeStudio from one batch of SCAPIS genotyping of the three neighboring SNPs with the highest correlation with rs10836441. No issues were observed in the cluster plots for these three markers (response letter figure 1a). Moreover, one of the directly genotyped markers, rs7124760 were associated with alpha diversity at the genome-wide threshold (response letter figure 1b).

Response letter figure 1: a) GenomeStudio cluster plots for the top 3 genotyped markers, with r^2 values with our lead SNP rs10836441 in SCAPIS of 0.98, 0.59, and 0.46, and p-values in the GWAS of 4.5×10^{-9} , 1.3×10^{-5} and 1.3×10^{-4} for panel 1-3, respectively. Numbers under each cluster indicate the number of individuals in each cluster. Samples marked with grey dots were excluded. b) Region plot depicting the genetic variants in the OR51E1/2 locus associated with microbial species richness. The three genotyped SNPs mentioned above are shown. This region plot without the highlighted SNPs is also available at Supplementary Fig.1a.

Per the Reviewer's request, we performed direct genotyping of the SNP to confirm imputation accuracy. We performed analysis using Sanger sequencing in 73 DNA samples from SCAPIS Uppsala and Malmö and 75 DNA samples from SIMPLER-V. Samples were picked based on the values of imputed dosages of the variant, with 25 samples each from 0-0.5, 0.5-1.5, and 1.5-2.0 dosage ranges. We found excellent concordance between Sanger sequencing and imputed dosages (Supplementary Figure 2).

Supplementary Figure 2. Results from Sanger sequencing of 73 samples from the SCAPIS cohort and 75 samples from the SIMPLER-V cohort compared to the imputed genotype dosages used in the association analysis for rs10836441.

Revised text:

Results, page 8

Imputed genotypes for rs10836441 were confirmed in a subset of 148 individuals using Sanger sequencing (Supplementary Fig. 2).

Methods, page 28-29

Validation by Sanger sequencing

Direct genotyping using Sanger sequencing was performed to confirm the variants for rs10836441 (OR51E1/2 locus) and rs4556017 (MUC12 locus) using samples from SCAPIS and SIMPLER-V cohorts. For each of the two SNPs, 75 samples were selected from each of the two cohorts (25 samples with imputed dosages of 0-0.5, 25 with 0.5-1.5, and 25 with 1.5-2.0). In total 300 samples were selected for genotyping (2 SNPs x 2 cohorts x 75 samples). Of these, four SCAPIS samples (two from each SNP) were unavailable in the biobank and therefore could not be sequenced. Sanger sequencing was performed in forward and reverse mode using the following primers: F 5'-CTCTGGGTCCTCTTTTCATCC-3' and R 5'-TGAAATATCCCTATTTGTACA 3' for rs10836441, and F 5'-TGATGTGTAAACCAGACAATA-3' and R 5'-TCAAGGTGCTCAGGCATCAA-3' for rs4556017. PCR was performed at an annealing temperature of 50 and 60°C for rs10836441 and rs4556017, respectively. Cleanup and Sanger sequencing were performed using ExoSAP-IT™ and the BigDye™ Terminator v1.1 Cycle Sequencing Kit, reducing the reaction volumes (Applied Biosystem Waltham, MA, USA). Dye terminator removal was performed using an EdgeBio column plate (Edge Biosystem Inc., CA, USA), and the capillary electrophoresis was performed on a SeqStudio™ Genetic analyzer (Applied Biosystem).

Raw data from Sanger sequencing was analyzed in Sequencher⁷⁴ v5.4.6 (Gene Codes) by an analyst blinded to the imputed data results. Reads were trimmed, reverse reads reversed and complemented, and finally aligned using default settings in Sequencher. This software was also used to call the two SNPs. For

rs10836441, two reverse reads were discarded due to unsuccessful sequencing. Eleven samples had an initial mismatch of the SNP calls from the forward and reverse reads, however, manual inspection of chromatograms revealed that the forward reads signal quality was excellent, while the reverse reads contained capillary migration artefacts with stretched peaks, and hence, these were all called according to the forward read (all heterozygotes). For the remaining 135 samples, the forward and reverse SNP calls agreed. For rs4556017, two forward reads and two reverse reads were discarded due to unsuccessful sequencing. The complementary read held high quality and for the remaining 144 samples, forward and reverse SNP calls agreed.

4. The authors included individuals with IBD and antibiotics in their discovery analysis. While they show that the main hits are not influenced when excluding these individuals, I strongly recommend running the discovery analysis without these individuals. It is good practice in microbiome studies to exclude individuals that took antibiotics up front (at least 6 weeks before sampling, 6 months more conservative).

Authors' reply:

While designing the analytical plan of our study, our decision to not exclude individuals with IBD and recent antibiotics in the discovery analyses was based on our ambition to identify genetic variants impacting the microbiome composition regardless of mechanism, in line with previous metagenomic microbiome-GWAS published in this journal (Qin et al., 2022, Lopera-Maya et al., 2022). GWASes in other fields have employed similar strategies e.g., GWAS studies of myocardial infarction not excluding individuals with high LDL-cholesterol (Tcheandjieu et al., 2022) or GWAS studies of childhood asthma not excluding individuals with allergies (Pividori et al., 2019). Indeed, in those studies, researchers observed that many of the associated genetic variants were linked to lipid traits and allergies, respectively, which improved their understanding of the underlying genetic architecture of these phenotypes. Moreover, restricting the study sample to healthier participants, i.e., by excluding individuals with IBD or recent infections, could lead to biased associations due to collider bias (Schoeler et al., 2023, Munafò et al., 2018). We would, therefore, prefer not to change the main analysis in the present study but instead continue to provide the results excluding IBD and recent antibiotic users as a sensitivity analysis, which for our associated variants provide nearly identical results.

Supplementary Figure 5a. Sensitivity analysis of 149 genome-wide associations restricted to individuals without antibiotic use in the past 6 months (n=14,171) compared to using data from all participants (n=16,017).

Supplementary Figure 5b. Sensitivity analysis of 149 genome-wide associations restricted to individuals without inflammatory bowel disease (n=15,260) compared to using data from all participants (n=16,017).

5. MUC12 association: Very intriguing signal! I would like to see though again that actually genotyped SNPs (with good clustering and quality) support the signal. The regional association plot in SFigure 1g shows a gap upstream of the main peak. What is the reason here? Can the authors genotype a subset of samples to confirm the genotype concordance (see also #2 above)? MUCs are normally excluded in WES studies, similarly as ORs.

Authors' reply:

Thank you for this question. The variant (rs4556017) was imputed with a high imputation info score of 0.90. Such uncertainties are dealt with in our study by using genotype dosages providing a probabilistic representation of imputed genotypes with continuous value between 0 and 2. To assess the quality of the underlying data for the imputation we went back to the raw data and extracted the signal intensity cluster plots with the software GenomeStudio from one batch of SCAPIS genotyping of the three neighboring SNPs with the highest correlation with rs4556017. No issues were observed in the cluster plots for these three markers (Response letter figure 2). Direct genotyping for the lead SNP rs4556017 was performed with Sanger sequencing, with the same methods described in the reply to Reviewer's comment number 3 (Supplementary Figure 7).

The gap in the regional plots most likely relates to unsolved gaps in the human genome assemblies for human *MUC* genes in general and the *MUC3* cluster in particular (Lang and Pelaseyed, 2022). Assemblies based on long-read sequencing such as the T2T-CHM13, are now filling these gaps. However, this was after the genotyping arrays of this study were designed. We have now added information to our revised manuscript about this region's neighboring gap in SNPs.

Response letter figure 2: a) GenomeStudio cluster plots for the top 3 genotyped markers, with r^2 values with our lead SNP rs4556017 in SCAPIS of 0.41, 0.32, and 0.32, and P -values in the GWAS of 9.4×10^{-9} , 2.4×10^{-7} and 1.2×10^{-7} for panel 1-3, respectively. Numbers under each cluster indicate the number of individuals. Samples marked with grey dots were excluded. b) Regional plot highlighting directly genotyped SNPs.

Supplementary Figure 7. Results from Sanger sequencing of 73 samples from the SCAPIS cohort and 75 samples from the SIMPLER-V cohort compared to the imputed genotype dosages used in the association analysis for rs4556017.

Revised text:

Results, page 13

Genes involved in the mucosal layer implicated in gut microbiome composition

We found and replicated an association between a variant in an intron of *MUC12* and *Coprobacillus cateniformis*, flanked by two other mucin genes, *MUC3A* and *MUC17* (Supplementary Fig. 1g). The same variant was also associated at study-wide significance with the genus *Coprobacillus*. **Our genotyping array did not cover the MUC3A gene well due to gaps in the human genome assemblies for the human MUC3 cluster.⁴⁰ Imputed genotypes for the lead SNP (rs4556017) were confirmed in a subset of 148 individuals using Sanger sequencing (Supplementary Fig. 7).**

Methods

See Reviewer 1, Comment 3.

6. Please better highlight hits in Table 1 that are replicated robustly (after Bonferroni-correction), e.g. by bold type.

Authors' reply: Bonferroni-replicated hits are now marked in bold.

7. I suggest moving Figure 3 to the Supplement, it shows expression of the candidate genes in the intestinal tract and is from a previously published data set. If the authors want to highlight OR51E1/2 expression, they should also do this in the Figure. What about immune cells that are also present in the gut and that interact with the microbial components?

Authors' reply:

We thank the Reviewer for the suggestion. We have moved Figure 3 to the Supplement (Supplementary Figure 3a in the revised manuscript) and included the expression in immune cells (Supplementary Figure 3b in the revised manuscript).

Additionally, *OR51E2* was expressed at some level in most immune cell types in the colon, highest in T-cells, followed by monocytes/macrophages. Expression of *OR51E2* was also detected in ileal monocytes/macrophages (Supplementary Fig. 3b).

New Supplementary Fig. 3

Supplementary Fig. 3. Single-cell expression analysis of candidate genes in (a) human duodenal, ileal, and colonic epithelial and (b) immune cells from donors. Heatmaps of mean gene expression were generated from the different intestinal epithelial and immune cell clusters in the dataset from Hickey et al.⁸⁴ T-cells data were absent for duodenum. (c) single-cell expression in mouse enteroendocrine cells (EECs) from different regions of the gut. Mean gene expression in different EEC clusters per gastrointestinal (GI) region. EECs were purified by flow cytometry from NeuroD1-Cre/YFP mice, and analyzed by 10X single-cell RNA sequencing. Cells were clustered by k-means and annotated according to their expression of gut

hormone genes: D (somatostatin), G (gastrin), I (cholecystokinin), K (glucose-dependent insulinotropic polypeptide), L (glucagon-like peptide 1 and peptide YY), InsI5 (insulin-like peptide 5), N (neurotensin), X (ghrelin), EC (enterochromaffin cells expressing Tph1 as a marker for serotonin biosynthesis), ECL (enterochromaffin-like cells expressing histidine decarboxylase as a marker for histamine biosynthesis). USI (upper small intestine), LSI (lower small intestine), LI (large intestine).

Minor points

1. Line numbers would have helped to point to specific sections.

Authors' reply: Line numbers have been added.

2. The authors should properly reference and acknowledge the first (!) descriptions of the LCT (probably Blekhman, R. et al. Host genetic variation impacts microbiome composition across human body sites. *Genome Biol.* 16, 191 (2015).; not listed among references) and ABO (in their paper #32, however, already mentioned in the beginning) associations.

Authors' reply:

We apologize for this omission and have now referred to the study by Blekhman et al. (reference number 15 in the revised manuscript) in the Main and Results sections of the revised manuscript where relevant.

Revised text:

Main, page 5

Despite these efforts, only the variants in the lactase (*LCT*) and histo-blood group ABO system transferase (*ABO*) loci have been identified at study-wide significance ($P < 5 \times 10^{-8}$ corrected for multiple species tested) and replicated across studies.^{7,11-15}

Results, page 10

These 6 loci included the well-known *ABO* and *LCT* loci.^{8,15}

3. Can the authors comment on the known MED13L association in their data? Do they replicate the signal?

Authors' reply:

The 2 SNPs (rs187309577 and rs143507801) in the *MED13L* locus previously associated with *Enterococcus* genus and *Enterococcus faecalis* in FINRISK (Qin et al., 2022) were not part of our GWAS, because the MAF for those 2 SNPs was <1% in our study (MAF = 0.001 for both SNPs in SCAPIS). Shown below (Response letter figure 3) is the region plot depicting the included common genetic variants in the *MED13L* locus and their association with *Enterococcus faecalis* in our GWAS, with the lowest p-value of 0.0006. The vertical dashed lines are the chromosomal positions of rs187309577 (left vertical dashed line) and rs143507801 (right vertical dashed line).

Response letter figure 3. Regional association plot of *Enterococcus faecalis* with variants in the MED13L locus.

We additionally performed a GWAS for *Enterococcus faecalis* without an MAF filter. However, we could not replicate the associations with the previously reported MED13L SNPs (rs187309577 p-value = 0.91 and rs143507801 p-value = 0.88).

4. Table 1: are the allele frequencies for all candidate variants comparable for the 4 individual Swedish studies, and also vs. HUNT? Or any significant deviations?

Authors' reply:

Allele frequencies for all study-wide significant variants are now shown per study in Supplementary Table 10. They are all comparable with no significant deviations, except for the LCT SNPs in FINRISK. LCT variants are known to vary across countries (Travis et al., 2013).

Revised text:

Results, page 10-11

Allele frequencies for all study-wide significant variants were comparable across studies, except for the LCT SNPs in FINRISK (Supplementary Table 10). LCT variants are known to vary across countries.³⁰

5. Figure 1a: please add age distribution of replication cohort HUNT in grey. And same for BMI. I also suggest adding a dashed line for the age distribution of the overall discovery cohort.

Authors' reply:

Thank you for the suggestion. We have now edited Figure 1a to include age and BMI in HUNT.

New figure legend:

Fig. 1. Characteristics of participants and microbiome composition in the discovery studies. a, Density plots of age and box-plots of BMI of the participants in the discovery studies **and in HUNT**. **The dashed line in the density plot of age shows the discovery cohorts combined.**

6. Figure 2a: please better highlight the novel loci here.

Authors' reply:

Thank you. We have highlighted those not previously found in GWAS with study-wide significance in purple.

New Figure 2a

Revised legend (Figure 2):

Fig. 2. Manhattan plot and cladogram for associations between genetic variants and species in the discovery studies. a, Combined Manhattan plot of the GWAS of 921 species in 16,017 individuals. The dashed black line represents the study-wide (top, $P < 5.4 \times 10^{-11}$) and the solid gray line genome-wide significant thresholds (bottom, $P < 5.0 \times 10^{-8}$). Filled triangles represent binary outcomes (absence/presence) and circles represent continuous outcomes (relative abundance). **Loci not previously found in other GWAS at study-wide significance are indicated with purple color.** b, Cladogram of genetic associations with gut bacterial species. The phylogenetic tree layers from center to periphery are kingdom-phylo-class-order-family, and all families captured by the 921 species are plotted. Phyla with at least one genetic association are colored. Species are placed at their family. c, Per-locus associations with microbiome species. In b and c, each dot corresponds to one species.

7. Figure 2b/c: make clear that one dot corresponds to one SNP.

Authors' reply:

We thank the Reviewer for the opportunity to clarify the legends for Figure 2.

Revised text:

Figure 2, page 23

b, Cladogram of genetic associations with gut bacterial species. The phylogenetic tree layers from center to periphery are kingdom-phylo-class-order-family, and all families captured by the 921 species are plotted. Phyla with at least one genetic

association are colored. Species are placed at their family. c, Per-locus associations with microbiome species. **In b and c, each dot corresponds to one species.**

8. Please provide references for the used tools, e.g. REGENIE, in Methods section.

Authors' reply: We have now provided references for all tools used in the Methods section.

Reviewer #2 (Remarks to the Author):

This is a largest to date metagenomics-based mbGWAS. It is performed in more or less standard way. However, I have some general comments that might make the study a bit better.

1. The signal for richness is borderline significant. I see no reason to skip microbiome-wise multiple testing for this specific association: it's still a part of mbGWAS, and the trait itself is still more or less independent from the species abundance. And with a study-wise cutoff it's not significant ($1.9e-9$). Having a replication is nice, but even meta-analysis p-value from discovery/replication cohorts only gives $P \sim 3e-10$, which is still below study-wise cutoff. It does not necessarily mean that this result should be removed from the paper, but some claims should be calmed down, like the beginning of discussion: "For the first time, we could identify and replicate a human genetic variant associated with gut 380 microbiome richness, the *OR51E1/E2* locus". It's a suggestive locus at best.

Authors' reply:

We thank the Reviewer for the comment. We respectfully disagree and would prefer to keep SNP-diversity analyses separately and not include them in the multiple testing correction together with the SNP-species analyses as we consider alpha diversity a different trait with a different hypothesis. However, we agree with the Reviewer that the interpretation of this association should be more cautious. Thus, we have toned down the claims around this association.

Revised text:

Results, page 7

Our analysis **indicates that** ~~revealed~~ variants (lead variant rs10836441-T) in one locus covering *OR51E1* (mouse orthologue *Olfir558*) and *OR51E2* (*Olfir78*) genes on chromosome 11 (Supplementary Fig. 1a) **are** associated with microbiome richness (-5.7 species per T allele, $P=1.9 \times 10^{-9}$, Supplementary Table 3).

Discussion, page 16

~~For the first time, we could identify and replicate~~ **We identified and replicated** a human genetic variant associated with gut microbiome richness **at genome-wide significance**, the *OR51E1/E2* locus.

2. I'm surprised not to see any kind of composition-aware transformation. Instead of 'validation' with non-transformed scale, it seems logical to make a round of validation with CLR instead. Compositionality might be a problem for any kind of MWAS, and mbGWAS in no exception

Authors' reply:

We thank the Reviewer for the opportunity to clarify our choice. We have now performed a round of sensitivity analysis for those 47 associations that were genome-wide significant in the linear model. Results were highly correlated and similar. We opted not to apply centered log-ratio (CLR) transformation to the microbiome data as our primary analysis, based on concerns from Boshuizen and te Beest (2022) about the ability of CLR to resolve the compositionality problem (Section 2.10 "Using the centred log-ratio to "deal" with compositionality.").

Supplementary Figure 5e. Sensitivity analysis of 56 genome-wide linear regression associations in the full dataset (n=16,017) using centered log-ratio (CLR) transformation + rank-based inverse normal transformation of relative abundances compared to rank-based inverse normal transformation.

Revised text (Results, page 10)

The genome-wide significant associations were similar in sensitivity analysis including models using centered log-ratio transformation for species analyzed using linear regression and Firth correction for species analyzed using logistic regression, in models not including age² as a covariate, in models where SCAPIS-Malmö and SCAPIS-Uppsala were analyzed separately, after excluding all but one individual per household and after excluding all but one individual from each related pair, or when individuals who used antibiotics in the last six months, or who self-reported inflammatory bowel disease (IBD) were excluded (Supplementary Fig. 5 and 6).

3. The choice of using specific toolkit for microbiome data processing (CHAMP). While showing a theoretical performance comparable or higher than other toolkits, there is one problem which creates problems for further use: the taxonomy. It seems that the pipeline uses GTDB taxonomy, even more: its outdated version. For example, *Clostridium* sp001916075 (associated to FUT2) in the last release is already called *Clostridium lentium* (https://gtdb.ecogenomic.org/genome?qid=GCF_014287815.1). I would recommend not to use taxonomy that is very tricky to find and that requires reader to go deep into the history of releases of databases. Ideally, in the text it's better to use NCBI taxonomy and give GTDB names somewhere in supplementary material (if possible). I absolutely agree that there is a huge mess with multiple taxonomies at the moment, yet still it is appreciated to make readers' life easier.

Authors' reply:

We fully agree with the Reviewer that the multiple taxonomies currently being used in the microbiology literature make comparing studies difficult in this quickly moving field with continuous evolution as more high-quality full genome sequences become available. We apologize for being unclear in our submitted version of the manuscript and thank for this possibility to clarify and improve the clarity of this issue in the revised manuscript. We consider GTDB as a well-established taxonomic system due to several reasons: It has been extensively used in studies of shotgun metagenomics data, including in the previous large GWAS of the gut microbiome

(Qin et al., 2022) published in *Nature Genetics*. Further, GTDB obtains genomes from NCBI but has more strict quality control criteria to include them in the database (<https://gtdb.ecogenomic.org/methods>). Additionally, not all genome-based species from GTDB have a corresponding NCBI annotation yet. For instance, *Clostridium* sp900540255 associated with *FUT2* is named “uncultured *Clostridium* sp.” in NCBI. Additionally, GTDB contains several *Faecalibacterium* spp. including *F. prausnitzii* subspecies (*F. prausnitzii_E*, *F. prausnitzii_F*, etc.) and *F. longum*, which contain genomes that correspond to *F. prausnitzii* in NCBI. Both NCBI and GTDB are constantly updating their taxonomy names. For example, NCBI is currently updating the phyla names and changing Firmicutes to Bacillota in their taxonomy (https://www.nlm.nih.gov/pubs/techbull/ma23/ma23_ncbi_taxonomy_names.html).

As the referee points out, we are not using the latest GTDB version. The most recent version (R220) was released on 24 April 2024, at that time the current manuscript was already soon to be submitted. Updating taxonomic annotations for species detected with shotgun metagenomics is not merely a matter of changing the names. Certain genomes initially assigned to one single taxon may become two taxa as the phylogenetic relationships are updated. *Actinomyces graevenitzii* in R214 became *Actinomyces graevenitzii* and *Actinomyces* sp915063485 in R220. We argue that, given the current metagenomics profiling available, it is reasonable to retain the R214 names in the manuscript. To facilitate for the reader, we added NCBI equivalents in Table 1 for the study-wide significant species based on the current GTDB species representative in <https://gtdb.ecogenomic.org/>.

Revised text:

Updated Table 1

Table 1: Loci associated with gut microbiome composition at study-wide significance, i.e., $P < 1.7 \times 10^{-8}$ for richness and $P < 5.4 \times 10^{-11}$ for species. Robust replication was considered in HUNT at a Bonferroni-corrected P of 3.3×10^{-3} .

Variant						Microbiome feature		Model	Swedish studies			HUNT		
Locus	Lead variant	Chr: Pos37	EA/OA	Effect prediction	EAF	Trait	Prev		Beta	SE	p	Beta	SE	p
OR51E1-OR51E2	rs10836441	11:4689742	T/C	intergenic	0.52	Richness	N.A.	Linear	-0.06	0.01	1.90E-09	-0.04	0.01	2.1E-03
LCT	rs4988235	2:136608646	A/G	intron (MCM6)	0.72	Negativibacillus sp000435195 (NCBI*: Clostridium sp. CAG:169)	29.2	Logistic	0.23	0.03	2.90E-13	0.09	0.04	0.04
	rs4988235	2:136608646	A/G	intron (MCM6)	0.72	Phocaea massiliensis	75.7	Linear	0.08	0.01	1.40E-11	0.08	0.02	5.1E-06
	rs182549	2:136616754	T/C	intron (MCM6)	0.72	Bifidobacterium adolescentis	90.0	Linear	-0.10	0.01	1.70E-16	-0.14	0.02	1.4E-14
	rs6754311	2:136707982	T/C	intron (DARS)	0.72	Copromonas sp000435795 (NCBI*: Alliscatomonas gsgli)	67.7	Linear	0.08	0.01	3.20E-11	0.10	0.02	2.4E-08
FOXP1	rs17007949	3:70920041	C/G	intergenic	0.32	Intestinibacter sp900540355 (NCBI*: Clostridium sp. 1001270J_160509_D11)	55.9	Linear	0.07	0.01	5.10E-11	0.03	0.01	7.3E-03
MUC12	rs4556017	7:100632790	T/C	intron (MUC12)	0.83	Coprobacillus cateniformis	31.4	Logistic	0.34	0.04	3.30E-17	0.38	0.06	1.7E-11
ABO	rs9411378	9:136145425	A/C	intron (ABO)	0.28	Mediterraneibacter torques (NCBI*: [Ruminococcus] torques)	86.9	Linear	0.11	0.01	1.40E-18	0.12	0.01	1.0E-16
	rs550057	9:136146597	T/C	intron (ABO)	0.31	Faecalibacterium longum	96.6	Linear	0.08	0.01	3.80E-11	0.08	0.02	1.6E-09
CORO7-HMOX2	rs8182173	16:4420787	T/C	intron (CORO7)	0.23	Clostridium saudiense	46.4	Logistic	-0.22	0.03	7.80E-13	-0.11	0.05	0.02
	rs4785960	16:4453319	C/G	intron (CORO7)	0.26	Turicibacter sanguinis	53.9	Linear	-0.08	0.01	2.00E-12	-0.04	0.01	1.7E-03
SLC5A11	rs55808472	16:24931691	A/G	non coding transcript exon (AC008731.1)	0.06	Agathobaculum butyrificiproducens	98.3	Linear	0.15	0.02	2.40E-11	0.16	0.03	4.3E-09
FUT3-FUT6	rs708686	19:5840619	T/C	upstream gene (FUT6)	0.3	Clostridium sp900540255 (NCBI*: uncultured Clostridium sp.)	41.6	Logistic	-0.20	0.03	4.50E-13	-0.11	0.05	0.02
	rs679574	19:49206108	C/G	intron (FUT2)	0.56	Clostridium sp001916075 (NCBI*: C. lentum)	36.4	Logistic	0.17	0.03	2.50E-11	0.15	0.05	1.6E-03
FUT2	rs492602	19:49206417	A/G	synonymous (FUT2)	0.56	Blautia A obscurum (NCBI*: B. obscurum)	96.4	Linear	0.07	0.01	1.60E-11	0.08	0.01	7.6E-10
	rs681343	19:49206462	T/C	stop gained (FUT2)	0.44	Clostridium sp900540255 (NCBI*: uncultured Clostridium sp.)	41.6	Logistic	-0.22	0.03	2.20E-18	-0.19	0.04	1.1E-05

*NCBI equivalents refer to the unfiltered NCBI taxonomy of GTDB species representative as of 2024-04-24. This was only added for species for which the name of the NCBI equivalent was different than GTDB.

Methods, page 31

The MAGs were taxonomically annotated using the GTDB database release 214 (date release: 28 April 2023). **GTDB nomenclature can be matched to other releases and to the National Center for Biotechnology Information (NCBI) nomenclature by using GTDB taxonomy history:**
https://qtodb.ecogenomic.org/taxon_history/.

4. Why didn't authors perform GWAS on higher taxonomic levels? It would make a lot of sense to do it on at least genera level, as the members of some genera carry a lot of similarities in their metabolic potential. I see no excuse for skipping it at least for genera and families (but also worth a try for other levels as well). The only possible reason why authors didn't do it is due to more strict necessity for multiple testing correction, but it's not a holy grail: I would even stick the cutoff to $5e-8 * N_{species}$, even if more taxonomic levels are involved (at the end of the day, from composition point of view, higher level taxa are just arithmetic sums of their species members).

Authors' reply:

We thank the Reviewer for this important comment. We have now performed GWAS of higher taxonomic levels. The results are summarized in the manuscript and included in the Supplementary Table 8.

Revised text:

Results, page 10

In the analyses of higher taxonomic ranks, 12 SNP-taxa associations were identified at study-wide significance in five loci (*LCT*, *PLEKHG1*, *MUC12*, *ABO*, *SLC5A11*) (Supplementary Table 8).

Results, page 10

Variants in *LCT* were also associated at study-wide significance with the genus *Phoceae* and *Bifidobacterium*, and the family, order, and class (*Bifidobacteriaceae*, *Actinomycetales*, and *Actinomycetia*, respectively) of the *Bifidobacterium* spp.

Results, page 11

Firstly, we confirmed previous associations of *ABO* variants with *Faecalibacterium longum* and reported a novel association with *Mediterraneibacter torques* and the genus **UMGS1623**.

Results, page 13

The same variant was also associated at study-wide significance with the genus *Coprobacillus*.

Results, page 14

A variant near *PLEKHG1* was also associated at study-wide significance with the genus, family (*Turicibacteraceae*), and order (*Haloplasmales*) of the *Turicibacter* spp.

Results, page 15

Further, variants in the SLC5A11 locus on chr 16 were associated with the abundance of *Agathobaculum butyriciproducens* and its family *Butyricicoccaceae* (Supplementary Fig. 1I and Supplementary Table 8).

Methods, page 34

GWAS of higher taxa

We also performed GWAS of 455 genera, 106 families, 50 orders, 21 classes, 17 phyla, and 3 superkingdoms. Similar to the GWAS of the species, 364 taxa detected in 5-50% of samples in each cohort were analyzed using logistic regression (absence/presence) and 288 taxa with prevalence >50% were estimated using linear regression of the RIN-transformed relative abundances. Study-wide significance was considered at $P < 5.4 \times 10^{-11}$, the same level as for species.

5. Figure 3 is quite uninformative. By itself it doesn't bring a clear message. It could be better to rethink it: maybe to add more panels to point to exact parts of it that are referred to in the text. MR results could be supported with the figures. Also, its description in the text is unclear: what were the instrument p-value cutoffs for exposures/outcomes? How was reverse causality controlled for? Were the results adjusted for multiple testing? The latter has to be done for sure.

Authors' reply:

Thank you. Figure 3 is now moved to the Supplementary Material (Supplementary Figure 3a in the revised manuscript) as suggested by Reviewer 1.

We thank the Reviewer for the questions about the MR analysis. The p-value cut-off used to create the genetic instruments was 5×10^{-8} . We have clarified this point in the Methods. We had not applied a multiple testing adjustment in the initial manuscript as we regarded these analyses as hypothesis-driven. We have now applied a Benjamini-Hochberg with False Discovery Rate of 5% to the MR-IVW p-values and focused on species and traits supported by the colocalization analysis.

About reverse causation, we present the results of a bidirectional MR in Supplementary Table 17 and in the new Supplementary Figure 8. We clarify the results of these analyses in the Results section.

Revised text:

Results, page 14-15

We observed a shared genetic signal between *Intestinibacter* sp9005540355 and LDL-C in the *FOXP1* locus, but not between *T. sanguinis*, *C. saudiense*, and LDL-C in the *CORO7-HMOX2* locus. We performed a Mendelian randomization (MR) analysis to investigate potential bidirectional effects between LDL-C and *Intestinibacter* sp9005540355. The analyses suggested a positive effect of *Intestinibacter* sp9005540355 abundance on LDL-C ($P = 4.4 \times 10^{-4}$, q-value = 0.001) but not in the opposite direction (Supplementary Table 17 and

Supplementary Fig. 8). Creating the genetic instruments using a more liberal P threshold of 5×10^{-6} yielded concordant results ($P=0.006$, q-value=0.02); however, the MR-Egger intercept indicates the presence of horizontal pleiotropy in this analysis ($P=0.012$). The *CORO7-HMOX2* locus was previously associated with the WHRadjBMI.⁵⁰ We found that WHRadjBMI shares a genetic signal with *T. sanguinis* and *C. saudiense* in colocalization analyses ($P(H4)>0.94$) (Supplementary Table 12). **The MR analyses showed evidence of an effect of *T. sanguinis* on WHRadjBMI, but not in the opposite direction. Analyses using the liberal P threshold of 5×10^{-6} to create genetic instruments did not detect an effect of *T. sanguinis* on WHRadjBMI ($P=0.23$). Still, these analyses could be underpowered. While the mechanism is still unclear, it seems plausible that these 2 loci might affect similar or the same pathways. Our findings suggest that genetic variations at 2 different loci, *CORO7-HMOX2* and *FOXP1*, affect a set of bacteria, including *Turicibacter* sp., *C. saudiense*, and an *Intestinibacter* species as well as **LDL-C**, bile acids, and body composition.**

Methods, page 36-37

We performed two-sample Mendelian randomization (MR) analyses to investigate bidirectional effects between specific gut microbiome species (*Clostridium saudiense*, *Turicibacter sanguinis*, *Intestinibacter* sp9005540355) and **BMI, WHR, and LDL-C**. The genetic associations with BMI and BMI-adjusted WHR were obtained from a meta-analysis of GWAS in GIANT and UK Biobank data (max $n=694,649$).^{101,102} Genetic associations with SHBG levels were obtained from UK Biobank.⁸⁶ **Genetic associations with LDL-C were obtained from a meta-analysis of GWAS of European ancestry ($n=1.32M$).**³⁹ Genetic instruments were created by **LD-clumping (LD R2 <0.001 within 10,000kb) of genetic variants associated with a species or trait with a $P < 5 \times 10^{-8}$.** LD-clumping was performed using the European 1000 Genomes reference.¹⁰³ **We only performed MR analysis when genetic instruments comprised >1 genetic variant. Sex-stratified analyses were performed when WHR was the exposure or the outcome.** For *Clostridium saudiense* and *Turicibacter sanguinis*, the instruments consisted of 2 loci, and for *Intestinibacter* sp9005540355 of 4 loci. **Sensitivity analyses were performed when the species were the exposure by using $P < 5 \times 10^{-6}$ to create the genetic instruments.** Inverse-weighted MR, weighted-median MR, and MR-Egger were conducted using the R package MendelianRandomization v0.9.0 (R v4.2.2). **As per default in this package, inverse-weighted MR with fixed-effects was used for genetic instruments with 2 variants and multiplicative random-effects with >2 variants. Multiple testing was addressed using Benjamini-Hochberg method considering a false discovery rate of 5% and the significance was expressed as q-values.**

6. FUT2 is not novel.

Authors' reply:

We thank the Reviewer for the opportunity to clarify why we considered FUT2 finding as novel. To the best of our knowledge, FUT2 was previously identified only at genome-wide significance, not at study-wide significance. To clarify this, we revised the text in the Abstract and Results section.

Revised text:

Abstract, page 4

Of these, we replicated 11 associations at six loci, confirming previous associations at *LCT*, *ABO* and *FUT2*, and identifying novel candidate genes *MUC12*, *CORO7-HMOX2*, *SLC5A11*, *FUT2*, *FOXP1*, and *FUT3-FUT6*, supported by stool and plasma metabolomic analyses and gene expression data.

Results, page 11

Our findings thus expand the number of robustly replicated loci **study-wide significantly** associated with the human microbiome composition from two (*ABO* and *LCT* loci) to six (*ABO*, *LCT*, *FUT2*, *MUC12*, *CORO7-HMOX2*, *SLC5A11*) and provide strong supportive evidence for two additional (*FUT3-FUT6* and *FOXP1*) loci.

Results, page 12

Variants in the *FUT2-FUT1* locus have been previously associated with the abundance of the *Ruminococcus torques* genus group¹¹ at the genome-wide significance level.

7. MUC genes (*MUC22* in particular) were also previously picked up in mbGWAS. Worth to mention in the discussion.

Authors' reply:

We thank the Reviewer for the comment. We have now added a mention of *MUC* genes detected in previous GWAS.

Added text:

Results, pages 14

Variants near *MUC* genes (*MUC5*, 12, 13, 22) have been previously suggested at genome-wide or near genome-wide significance with metagenomic features.^{9,14,44}

8. Why there's no functional mbGWAS performed? KEGG or metacyc

Authors' reply:

We thank the Reviewer for this suggestion. We have now performed a GWAS on 117 gut metabolic modules (Vieira-Silva et al., 2016) and the gut-brain modules (Valles-Colomer et al., 2019). No study-wide significant findings were identified. Using a genome-wide significance threshold, we found 12 candidate genetic loci, including *CYP7A1*, *EGFR*, and *SLC37A3*, associated with a total of 13 microbial functions, respectively, most of them related to carbohydrate and amino acid catabolism.

We have now updated the manuscript:

Methods, page 34

GWAS of functional modules

For functional annotation, catalogue genes were annotated to the gut modules (gut metabolic modules v1.07 and gut-brain modules v1.0) using EggNOG-mapper⁹² v2.0.1. Potential functional profiles were determined for species that contained at least 2/3 of the enzymes/protein genes needed for the functionality of a particular module. If an alternative reaction pathway within a module existed, only one such pathway was required. All reaction pathways

were required for modules with fewer than four steps. Module abundances were defined as the sum of the relative abundances of all species in a module. We removed 34 gut-brain modules with $r > 0.95$ with gut metabolic modules. Similar to the GWAS of the species, 2 modules detected in 5-50% of samples in each cohort were analyzed using logistic regression (absence/presence), and 115 modules with prevalence $> 50\%$ were estimated using linear regression of the RIN-transformed relative abundances. Study-wide significance was considered at $P < 4.3 \times 10^{-10}$.

Results, page 7

Our GWAS meta-analysis included alpha diversity, measured as microbiome richness (number of different species), Shannon index, and inverse Simpson index, as well as 921 species present in $\geq 5\%$ of the participants in each of the four Swedish studies, while excluding 3,214 detected species with $< 5\%$ prevalence. **Similar filtering was applied for taxonomic levels above species and functional modules, resulting in 652 higher taxa and 117 modules. ...**

...GWAS was performed separately for **each microbial phenotype using REGENIE v3.3 in each cohort separately, with sex, age, age², metagenomics sequencing plate, and genetic principal components 1-10 included as covariates, and was followed by inverse-variance weighted fixed-effects meta-analysis.**

Results, page 16

Loci associated with microbial functions suggest a genetic link to microbial carbohydrate and amino acid catabolism

We investigated associations between host genetic variation and 117 previously curated functional modules representing different aspects of microbial metabolism⁵⁶ and microbial functions implicated in the gut-brain axis.⁵⁷ No study-wide significant findings were identified. Using the genome-wide significance threshold, we found 12 candidate genetic loci, including *CYP7A1*, *EGFR*, and *SLC37A3*, associated with 13 microbial functions, most related to carbohydrate and amino acid catabolism (Supplementary Table 18).

9. MR part is rather weak. No reverse causality tests, no multiple testing adjustment for exposure-outcome pairs tested. It might be good idea to decrease the inclusion cutoff to get more instruments as a validation.

Authors' reply:

We thank the Reviewer for the comments. Please find our answer about reverse causality, multiple testing, and changes made to the manuscript in our answer to question 5.

We have also performed MR analyses of species on traits using a p-value threshold of 5×10^{-6} . These results are now presented in Supplemental Table 17 and Supplemental Figure 8.

New text:

Results, page 15

Creating the genetic instruments using a more liberal P threshold of 5×10^{-6} yielded concordant results ($P = 0.006$, q-value = 0.02); however, the MR-Egger

intercept indicates the presence of horizontal pleiotropy in this analysis ($P=0.012$).

The MR analyses showed evidence of an effect of *T. sanguinis* on WHRadjBMI, but not in the opposite direction. Analyses using the liberal p-threshold of 5×10^{-6} to create genetic instruments did not detect an effect of *T. sanguinis* on WHRadjBMI ($P=0.23$).

Methods, page 37

Sensitivity analyses were performed when the species were the exposure by using $P < 5 \times 10^{-6}$ to create the genetic instruments.

10. Last but not least, the adjustment for covariates. There are multiple problems here that might potentially affect the results: (1) some of the cohorts like SCAPIS and SIMPLER are actually multi-centre cohorts, thus the SNPs might as a proxy differences in LD between counties/cities, rather than be causal. This should be taken into account, and a few genetic components might not be enough. (2) most clear in SCAPIS but also can affect others: the total population of Malmö and Uppsala is 533 thousand people, and many of those are not from European ancestry. With only selecting 50-65 year-old people, this gives an extremely small general population to sample from. It is very, very likely that many of those people are relatives, even close relatives, or cohabitants. Cohabitation is an extremely important source of microbiome variation and has to be taken into account.

So, to conclude, both adjustment for cohabitation (if possible) and more robust adjustment for genetic kinship should be used in the study, as both of those problems are likely to occur in the study cohorts and might significantly affect the results.

Authors' reply:

Thank you for raising these important questions.

- 1) **Site adjustment:** We agree that including several sites in the same analysis might bias the results due to genetic confounding and produce too small standard errors and inflated p-values. We have mitigated this and investigated the potential impact by using the following strategies:
 - a) Clarified that we accounted for the site in SCAPIS by adjusting for metagenomics extraction plate, as the plate was nested in the variable site. MOS only had one test site, SIMPLER-V and SIMPLER-U were already analyzed separately.
 - b) We have also now added a sensitivity analysis where we analyze SCAPIS-Uppsala and SCAPIS-Malmö separately. After meta-analysis, the main associations were nearly identical (Supplementary Figure 5h).
 - c) application of the software REGENIE, which is designed to account for substructure in the study population
 - d) inclusion of 10 principal components in the regression analysis.
 - e) assessing genomic inflation of p-values and presenting QQ plots
 - f) replication in external studies

Supplementary Figure 5h. Sensitivity analysis of 149 genome-wide associations in the full dataset (n=16,017) where SCAPIS-Malmö and SCAPIS-Uppsala were analyzed separately compared to models adjusted for the SCAPIS sites (original analysis).

2) **Relatedness:** The reviewer is correct that our study samples contain individuals who are relatives. The proportion of individuals with relatedness 3rd degree or closer amounts to 14% in SCAPIS, 16% in SIMPLER-V, 7% in SIMPLER-U, and 71% in MOS. Including related subjects in the analysis without proper control would produce too small standard errors and inflated p-values. We have mitigated this and investigated the potential impact by using the following strategies:

- a) application of the software REGENIE, which is designed to account for relatedness. REGENIE has been evaluated in Mbatchou et al. (2021) under various scenarios of relatedness, showing that REGENIE retained good type 1-error control, even when half of the samples were related if the cohorts are sufficiently large, as in our study. REGENIE may be overly conservative when the cohorts are smaller in size and the phenotype is binary (Gurinovich et al., 2022).
- b) assessing genomic inflation of p-values and added QQ plots (Supplementary Figure 6)
- c) replication in external studies
- d) addition of a new sensitivity analysis for our genome-wide significant species by excluding individuals (N = 1794; 11%) until there were no pairs with 3rd degree relatedness or closer. The sensitivity analysis estimates are highly similar to the main analysis (Supplementary Figure 5c):

Supplementary Figure 5c. Sensitivity analysis of 149 genome-wide associations restricted to unrelated participants (n=14,229) compared to using data from all participants (n=16,017). Related participants were identified based on kinship coefficients, and individuals were excluded until there were no pairs remaining with 3rd degree relatedness or closer.

3) **Cohabitation:** We agree that cohabitation (e.g., spouses) is an important source of microbiome variation. Our study samples contain cohabiting individuals; in SIMPLER, this amounts to 7%, and in MOS, to 11%. However, cohabitation should not be associated with the genetic setup. With no within-cluster correlation of the exposure, adjusting standard errors is not necessary according to several sources (Abadie et al., 2022; Cameron and Miller, 2015). In the case of assortative mating, there could be exceptions; however, SNPs deviating strongly from the Hardy-Weinberg assumption were filtered out before analysis. We have further mitigated bias from cohabitation using the following steps:

- a) application of the software REGENIE that is designed to account for substructure in the study population and relatedness (Mbatchou et al., 2021)
- b) assessing genomic inflation of p-values and presenting QQ plots
- c) replication in external studies
- d) A new sensitivity analysis was added, excluding 301 of 7,284 (4%) individuals until there was only one person per household from SIMPLER and MOS, and estimates were compared with the original analysis. The sensitivity analysis estimates are highly similar to the main analysis restricted to SIMPLER and MOS participants (Supplementary Figure 5d):

Supplementary Figure 5d. Sensitivity analysis of 149 genome-wide associations restricted to one participant from each household in SIMPLER and MOS (n=6,983) compared to using data from all participants in those cohorts (n=7,284).

Supplementary Figure 6. Quantile-quantile plots and genomic inflation factors for study-wide significant findings (richness and species)

In summary, including related or otherwise clustered subjects in the analysis without proper control would likely produce false positive findings due to small standard errors and inflated p-values. However, we do not observe any evidence of genomic inflation, the added sensitivity analyses show consistent results, and our additional external replication supports our results. We, therefore, conclude that we do not find any evidence to believe that relatedness or population stratification bias is the cause of the observed results. We thank the Reviewer for raising this important question and have made the following additions to the manuscript.

Revised text:

Results, page 7

Based on simulations aiming to maximize power and minimize false positive findings, we used logistic regression for the presence of 679 species detected in 50% or fewer of the samples, testing 5,368,906 variants with minor allele frequency (MAF) $\geq 5\%$. For 242 species with a prevalence $>50\%$ in all four studies, we used linear regression for the relative abundance of those species, testing 7,454,886 variants with $MAF \geq 1\%$. **GWAS was performed separately for each microbial phenotype using REGENIE v3.3 in each cohort separately, with sex, age, age², metagenomics sequencing plate, and genetic principal components 1-10 included as covariates, and was followed by inverse-variance weighted fixed-effects meta-analysis.** We replicated study-wide associations of host genetic variants with species and species diversity in HUNT ($n=12,652$), including stool metagenomic data profiled with the same pipeline employed in the Swedish studies.

Results, page 10

The genome-wide significant associations were similar in sensitivity analysis including models using centered log-ratio transformation for species analyzed using linear regression and Firth correction for species analyzed using logistic regression, in models not including age² as a covariate, in models where SCAPIS-Malmö and SCAPIS-Uppsala were analyzed separately, after excluding all but one individual per household and after excluding all but one individual from each related pair, or when individuals who used antibiotics in the last six months, or who self-reported inflammatory bowel disease (IBD) were excluded (Supplementary Fig. 5 and 6).

Methods, page 32-33

GWAS was performed separately for microbial alpha diversity and 921 species using REGENIE⁸⁵ v3.3 for each cohort (**SCAPIS, SIMPLER-V, SIMPLER-U, MOS**). A subset of the genotype datasets was created for the first REGENIE step to fit whole genome regression models including only quality-controlled directly genotyped SNPs with $MAF > 1\%$ and $HWE P < 1 \times 10^{-15}$. For the second step all variants with an information score > 0.7 was included in association analyses performed using logistic regression for binary variables and genetic variants with $MAF > 5\%$, and linear regression for RIN-transformed variables and genetic variants with $MAF > 1\%$. Covariates were sex, age, age², metagenomics sequencing plate, and genetic principal components 1-10. **Metagenomics sequencing plate, age, and sex were included to increase precision and power. For SCAPIS, the site was accounted for by the plate variable because plate was nested into the site variable, while all other cohorts were single-site. Based on previous non-linear associations between age and microbiome⁸⁶ and our results from a naive linear model for the association between age and microbial species, we opted to include age also as age². REGENIE accounts for population stratification, and to further account for any residual bias we further included PC1-PC10 in the model.⁸⁷** Cohort-specific results were meta-analyzed using the inverse-variance weighted method in METAL⁸⁸ v2011-03-25.

Methods, page 33

Sensitivity analyses

Sensitivity analyses were performed for the 149 genome-wide locus-species associations by 1) excluding individuals with antibiotic use in the 6 months before sampling, 2) excluding individuals with self-reported inflammatory bowel disease, 3) retaining an unrelated subset where no individuals had 3rd degree relatedness or closer with any other individual using a KING-robust kinship estimator threshold of 0.0442, 4) retaining one random spouse in SIMPLER and one random participant living at the same address in MOS to assess cohabitation (SCAPIS was removed for this analysis), 5) using centered log ratio plus rank inverse normal transformation for species analyzed using linear regression, 6) using Firth correction for species analyzed using logistic regression, 7) removing age² from the covariates, and 8) analyzing SCAPIS-Uppsala and SCAPIS-Malmö as two separate cohorts in the meta-analysis.

References

- Abadie, A., et al. (2022). When should you adjust standard errors for clustering?. *The Quarterly Journal of Economics* 138(1): 1-35.
- Bonder, M. J., et al. (2016). The effect of host genetics on the gut microbiome. *Nature Genetics* 48(11): 1407-1412.
- Boshuizen, H. C. and D. E. Te Beest (2023). Pitfalls in the statistical analysis of microbiome amplicon sequencing data. *Molecular Ecology Resources* 23(3): 539-548.
- Boulund, U., et al. (2022). Gut microbiome associations with host genotype vary across ethnicities and potentially influence cardiometabolic traits. *Cell Host & Microbe* 30(10): 1464-1480.e1466.
- Colin Cameron, A. and D. L. Miller (2015). A practitioner's guide to cluster-robust inference. *Journal of Human Resources* 50(2): 317-372.
- Gurinovich, A., et al. (2022). Evaluation of GENESIS, SAIGE, REGENIE and fastGWA-GLMM for genome-wide association studies of binary traits in correlated data. *Frontiers in Genetics* 13: 897210.
- Lang, T. and T. Pelaseyed (2022). Discovery of a MUC3B gene reconstructs the membrane mucin gene cluster on human chromosome 7. *PLoS One* 17(10): e0275671.
- Lopera-Maya, E. A., et al. (2022). Effect of host genetics on the gut microbiome in 7,738 participants of the Dutch Microbiome Project. *Nature Genetics* 54(2): 143-151.
- Mattar, R., D. F. de Campos Mazo and F. J. Carrilho (2012). Lactose intolerance: diagnosis, genetic, and clinical factors. *Clinical and Experimental Gastroenterology* 5: 113-121.
- Mbatchou, J., et al. (2021). Computationally efficient whole-genome regression for quantitative and binary traits. *Nature Genetics* 53(7): 1097-1103.
- Munafò, M. R., et al. (2018). Collider scope: when selection bias can substantially influence observed associations. *International Journal of Epidemiology* 47(1): 226-235.
- Odamaki, T., et al. (2016). Age-related changes in gut microbiota composition from newborn to centenarian: a cross-sectional study. *BMC Microbiology* 16: 90.
- Pividori, M., et al. (2019). Shared and distinct genetic risk factors for childhood-onset and adult-onset asthma: genome-wide and transcriptome-wide studies. *The Lancet Respiratory Medicine* 7(6): 509-522.
- Qin, Y., et al. (2022). Combined effects of host genetics and diet on human gut microbiota and incident disease in a single population cohort. *Nature Genetics* 54(2): 134-142.
- Schoeler, T., et al. (2023). Participation bias in the UK Biobank distorts genetic associations and downstream analyses. *Nature Human Behaviour* 7(7): 1216-1227.

- Tcheandjieu, C., et al. (2022). Large-scale genome-wide association study of coronary artery disease in genetically diverse populations. *Nature Medicine* 28(8): 1679-1692.
- Travis, R. C., et al. (2013). Genetic variation in the lactase gene, dairy product intake and risk for prostate cancer in the European prospective investigation into cancer and nutrition. *International Journal of Cancer* 132(8): 1901-1910.
- Valles-Colomer, M., et al. (2019). The neuroactive potential of the human gut microbiota in quality of life and depression. *Nature Microbiology* 4(4): 623-632.
- Vieira-Silva, S., et al. (2016). Species-function relationships shape ecological properties of the human gut microbiome. *Nature Microbiology* 1(8): 16088.

Reviewer #1 (Remarks to the Author):

The authors have done a very good job in addressing my comments. Extra efforts were put into validating some of their lead SNPs with an independent genotyping technology (here Sanger resequencing). Several of my concerns have diminished, only few questions remain:

(1) Can the authors mention the genotype concordance in % rather than showing the plots? Is the concordance close to 100%? Supplementary Figure 7 (MUC12 Locus) does not look convincing!

Authors' reply:

We would like to thank the Reviewer for the positive feedback and for the additional opportunity to clarify our results. Imputed genotype dosages were modelled on a continuous scale from 0-2 to account for uncertainties in the imputation, which varies by the individual sample. To directly compare the dosage results (0-2) with the discrete variable (0,1,2) from Sanger sequencing we provided the results in the mentioned histogram (Extended Data Figure 8). To be able to provide the requested genotype concordance in %, we have now created a "predicted-genotype" for the imputed data i.e., assigning 0 to dosages 0-0.49, 1 to dosages 0.5-1.49 and 2 to dosages 1.5-2. For the *MUC12* SNP, the resulting concordance was 96.6%. We have now included this result together with the results for the *OR51E1* SNP (100%).

Results, page 13

Genes involved in the mucosal layer implicated in gut microbiome composition

We found and replicated an association between a variant in an intron of *MUC12* and *Coprobacillus cateniformis*, flanked by two other mucin genes, *MUC3A* and *MUC17* (Extended Data Fig. 1h). The same variant was also associated at study-wide significance with the genus *Coprobacillus*. Our genotyping array did not cover the *MUC3A* gene well due to gaps in the human genome assemblies for the human *MUC3* cluster.⁴⁰ Imputed genotypes for the lead SNP (rs4556017) were confirmed in a subset of 148 individuals using Sanger sequencing **with a concordance of 96.6%** (Extended Data Fig. 8).

Results, page 7-8

Our analysis indicates that variants (lead variant rs10836441-T) in one locus covering *OR51E1* (mouse orthologue *Olfir558*) and *OR51E2* (*Olfir78*) genes on chromosome 11 (Extended Data Fig. 1a) are associated with microbiome richness (-5.7 species per T allele, $P=1.9 \times 10^{-9}$, Supplementary Table 3).

Imputed genotypes for rs10836441 were confirmed in a subset of 148 individuals using Sanger sequencing **with a concordance of 100%** (Extended Data Fig. 2).

Methods, page 29

To evaluate the concordance between imputed genotype dosages and the Sanger sequencing genotypes, we assigned a predicted genotype with value 0 for dosages 0-0.49, 1 to dosages 0.5-1.49, and 2 to dosages 1.5-2.

(2) Along the same line, Response letter figure 2a highlights 3 genotyped SNPs. However, these genotyped SNP are neither passing study-wide significance nor are they in high LD with the peak SNPs. This is concerning. In addition, the authors have not tested known variants in the gap and just write "Assemblies based on long-read sequencing such as the T2T-CHM13, are now filling these gaps." While I understand that the authors worked only with the array content and imputed genotypes thereof, they should have meanwhile genotyped SNPs in the gap region (given that informative and testable variants exist). This would help to understand the signal and LD structure better. Maybe the authors will also get a stronger signal. It is not satisfactory to have this gap in the data while it could potentially have been closed.

Authors' reply:

We interpret the comment from the Reviewer as they find two potential issues with the signal at the *MUC12* locus. The first (a) is that they find the signal at rs4556017 unreliable as it is imputed based on a sparsely genotyped array. The second issue (b) is that the signal might have arisen from being in LD with causal variants in the unresolved region downstream of *MUC12* covering *MUC3A*.

(a) We find the signal reliable based on the following line of evidence:

1. The rs4556017 signal for *Coprobacillus cateniformis* is confirmed with consistent effect direction and low p-values in two external materials:
HUNT, p-value 1.7×10^{-11}
FINRISK, p-value 2.7×10^{-7}
2. The rs4556017 variant has previously been reported as the lead variant associated with gastrointestinal traits:
Stool frequency: 167,875 Europeans (Belgium, Netherlands, Sweden, U.K., U.S.): p-value, 1×10^{-9} (Bonfiglio et al., 2021)
Hemorrhoidal disease: 944,133 Europeans (U.S., U.K., Estonia), p-value, 1×10^{-22} (Zheng et al., 2021)
Diverticular disease: 454,768 Europeans (U.S., Finland, U.K.), p-value, 5×10^{-9} (Wu et al., 2023)

! % % 23 % " % * ! ! ! " \$+ - * %
 ! + % , % %
 - ! * ! % " . * ! E + %
 ! * * ! * ! " * + ! 5 ! E % !! "
 * * ! * + !! + > > * \$+ >

(b) The second issue is that the signal might have arisen from being in LD with causal variants in the partially unresolved gap region downstream of *MUC12* covering *MUC3A*. We agree that the gap of known variation in this region covering approximately 60kbp limits the potential to better understand the association of genetic variation at this locus with the gut microbiome. The mentioned T2T-CHM13 assembly was published in Science in 2022 by the Telomere-to-Telomere (T2T) Consortium (Nurk et al., 2022) and managed to fill almost all gaps in the GRCh38 reference. The study was based on extensive long-read sequencing with complementary techniques of a specific single cell culture with no genetic variation (all genotypes in this cell line are homozygous) and did, therefore, by design, not report genetic variation. The T2T-CHM13 assembly is indeed a powerful resource but the identification of genetic variation in this gap region is still lacking. To characterize the genetic variation in this gap region in large-population-based studies like ours would be a major undertaking for any research group or consortium. Genotyping is not possible as we would not know what to genotype. Short-read sequencing has already failed in previous attempts. High-quality long-read sequencing might solve the issue, but in the current technological state, this would be unfeasible at this scale. To acknowledge this limitation, we have now added this as a limitation in the Discussion.

Discussion, page 17

Our large sample size and the use of deep shotgun metagenomic sequencing enabled these findings, providing robust and detailed profiling of the gut microbiome. However, all detected associations on the study-wide level were for species that were identified in 29% or more of the population, although the vast majority of detected species in these studies are less common than that. This study is still underpowered to detect associations with rare genetic variants or less prevalent microbial species. **Moreover, the incomplete genomic coverage of the region downstream *MUC12* on chromosome 7 in the reference genome underlying our genotyping arrays hindered us from fully exploring associations in that locus.**

(3) Co-variates: For the genetic data, I am happy with the reply, but I was more thinking of other factors that are known to shape the microbiome: BMI, alcohol consumption, smoking, ... and also stool composition is strongly associated to gut microbiome composition. Can the authors exclude that accounting for these factors won't change the association results of the main hits?

Authors' reply:

We have now included results from models adjusted for BMI, smoking, alcohol, and fiber intake. Bristol stool scale was unfortunately only available in a small subset (<2300).

Results, page 10

We did not observe evidence of genomic inflation (mean lambda = 1.03, SD = 0.02), and findings were consistent across studies (Supplementary Table 4, **Extended Data Fig. 5**). The genome-wide significant associations were similar in sensitivity analysis including models using centered log-ratio transformation for species analyzed using linear regression and Firth correction for species analyzed using logistic regression, in models not including age² as a covariate, in models where SCAPIS-Malmö and SCAPIS-Uppsala were analyzed separately, after excluding all but one individual per household and after excluding all but one individual from each related pair, when individuals who used antibiotics in the last six months, or who self-reported inflammatory bowel disease (IBD) were excluded, **or when including BMI, smoking, alcohol intake or fiber intake as covariates (Extended Data Fig. 6)**.

Methods, page 25

Body mass index (BMI) was defined as weight divided by height squared (kg/m²). Usual alcohol and fiber intakes were estimated from a food frequency questionnaire (g/day).⁵⁹ Smoking behavior was assessed using a questionnaire and defined as current, former, and never smoker.

Extended Data Fig. 6: (a) Sensitivity analysis of 149 genome-wide associations restricted to individuals without antibiotic use in the past 6 months ($n=14,171$) compared to using data from all participants ($n=16,017$). (b) Sensitivity analysis of 149 genome-wide associations restricted to individuals without inflammatory bowel disease ($n=15,260$) compared to using data from all participants ($n=16,017$). (c) Sensitivity analysis of 149 genome-wide associations restricted to unrelated participants ($n=14,229$) compared to using data from all participants ($n=16,017$). Related participants were identified based on kinship coefficients, and individuals were excluded until

there were no pairs remaining with 3rd degree relatedness or closer. (d) Sensitivity analysis of 149 genome-wide associations restricted to one participant from each household in SIMPLER and MOS (n=6,983) compared to using data from all participants in those cohorts (n=7,284). (e) Sensitivity analysis of 56 genome-wide linear regression associations in the full dataset (n=16,017) using centered log-ratio (CLR) transformation + rank-based inverse normal transformation of relative abundances compared to rank-based inverse normal transformation. (f) Sensitivity analysis of 93 genome-wide logistic regression associations in the full dataset (n=16,017) using Firth correction compared to not using Firth correction. (g) Sensitivity analysis of 149 genome-wide associations in the full dataset (n=16,017) excluding age² compared to including age² as a covariate. (h) Sensitivity analysis of 149 genome-wide associations in the full dataset (n=16,017) where SCAPIS-Malmö and SCAPIS-Uppsala were analyzed separately compared to models adjusted for the SCAPIS sites (original analysis). (i) **Sensitivity analysis of 149 genome-wide associations in the full dataset including BMI (n=16,011) as additional covariate compared to not adding BMI (n=16,017).** (j) **Sensitivity analysis of 149 genome-wide associations in SCAPIS including alcohol intake (n=8,707) as additional covariate compared to not adding alcohol intake (n=8,733).** (k) **Sensitivity analysis of 149 genome-wide associations in SCAPIS including smoking behavior (n= 8,452; current smokers: 12%, never smokers: 49%, missing: 3%) as additional covariate compared to not adding smoking behavior (n=8,733).** (l) **Sensitivity analysis of 149 genome-wide associations in SCAPIS including fiber intake (n= 8,624) as additional covariate compared to not adding fiber intake (n=8,733). Smoking behavior (3% missing) was categorized into current smokers (12%), former smokers (35%), never smokers (49%). Mean±SD for fiber intake and median (25th-75th percentile) for alcohol intake in SCAPIS were 12.0±4.2 g/d and 5.9 (2.0-10.6) g/d, respectively.**

In all plots the diagonal black line indicates where values of $y = x$ and the red line a slope from linear regressions of beta coefficients from the sensitivity analysis and the original analysis, and in the upper left corner the Pearson correlation coefficient r is shown.

Methods, page 34

Sensitivity analyses

Sensitivity analyses were performed for the 149 genome-wide locus-species associations by 1) excluding individuals with antibiotic use in the 6 months before sampling, 2) excluding individuals with self-reported inflammatory bowel disease, 3) retaining an unrelated subset where no individuals had 3rd degree relatedness or closer with any other individual using a KING-robust

kinship estimator threshold of 0.0442, 4) retaining one random spouse in SIMPLER and one random participant living at the same address in MOS to assess cohabitation (SCAPIS was removed for this analysis), 5) using centered log ratio plus rank inverse normal transformation for species analyzed using linear regression, 6) using Firth correction for species analyzed using logistic regression, 7) removing age² from the covariates, 8) analyzing SCAPIS-Uppsala and SCAPIS-Malmö as two separate cohorts in the meta-analysis, **and 9-12) adding BMI, alcohol intake, smoking, or fiber intake, respectively, as a covariate. The analyses with smoking, alcohol and fiber were performed in SCAPIS only, where data on these variables were nearly complete.**

Reviewer #2 (Remarks to the Author):

The authors have thoroughly taken into account the concerns from me and another reviewer. My concerns were mostly about:

The complex genetic architecture of studied populations. It seems that the adjustment for that was done properly.

GWAS on higher taxonomic ranks and functional pathways. This is now also done.

Mess in taxonomic names and taxonomic databases. This is now better.

Weak MR part, especially FDR and test for horizontal pleiotropy. Now is also done.

Overall, the authors did a good job in addressing my concerns. This paper is a great addition to a field of microbiome GWAS overall, and I was very glad for having the opportunity to review it.

Authors' reply:

We thank the Reviewer for the positive comments.

References

Bonfiglio, F. et al. GWAS of stool frequency provides insights into gastrointestinal motility and irritable bowel syndrome. *Cell Genom* 1, 100069 (2021).

Nurk, S. et al. The complete sequence of a human genome. *Science* 376, 44-53 (2022).

Wu, Y. et al. 150 risk variants for diverticular disease of intestine prioritize cell types and enable polygenic prediction of disease susceptibility. *Cell Genom* 3, 100326 (2023).

Zheng, T. et al. Genome-wide analysis of 944 133 individuals provides insights into the etiology of haemorrhoidal disease. *Gut* 70, 1538-49 (2021).